# Minimum Stein Discrepancy Estimators

**Alessandro Barp**
Department of Mathematics
Imperial College London
a.barp16@imperial.ac.uk

**François-Xavier Briol**
Department of Statistical Science
University College London
f.briol@ucl.ac.uk

**Andrew B. Duncan**
Department of Mathematics
Imperial College London
a.duncan@imperial.ac.uk

**Mark Girolami**
Department of Engineering
University of Cambridge
mag92@eng.cam.ac.uk

**Lester Mackey**
Microsoft Research
Cambridge, MA, USA
lmackey@microsoft.com

## Abstract

When maximum likelihood estimation is infeasible, one often turns to score matching, contrastive divergence, or minimum probability flow to obtain tractable parameter estimates. We provide a unifying perspective of these techniques as *minimum Stein discrepancy estimators*, and use this lens to design new diffusion kernel Stein discrepancy (DKSD) and diffusion score matching (DSM) estimators with complementary strengths. We establish the consistency, asymptotic normality, and robustness of DKSD and DSM estimators, then derive stochastic Riemannian gradient descent algorithms for their efficient optimisation. The main strength of our methodology is its flexibility, which allows us to design estimators with desirable properties for specific models at hand by carefully selecting a Stein discrepancy. We illustrate this advantage for several challenging problems for score matching, such as non-smooth, heavy-tailed or light-tailed densities.

## 1 Introduction

Maximum likelihood estimation [9] is a de facto standard for estimating the unknown parameters in a statistical model $\{\mathbb{P}_\theta : \theta \in \Theta\}$. However, the computation and optimization of a likelihood typically requires access to the normalizing constants of the model distributions. This poses difficulties for complex statistical models for which direct computation of the normalisation constant would entail prohibitive multidimensional integration of an unnormalised density. Examples of such models arise naturally in modelling images [27, 39], natural language [54], Markov random fields [61] and nonparametric density estimation [63, 69]. To by-pass this issue, various approaches have been proposed to address parametric inference for unnormalised models, including Monte Carlo maximum likelihood [22], contrastive divergence [28], minimum probability flow learning [62], noise-contrastive estimation [10, 26, 27] and score matching (SM) [34, 35].

The SM estimator is a minimum score estimator [16] based on the Hyvärinen scoring rule that avoids normalizing constants by depending on $\mathbb{P}_\theta$ only through the gradient of its log density $\nabla_x \log p_\theta$. SM estimators have proven to be a widely applicable method for estimation for models with unnormalised smooth positive densities, with generalisations to bounded domains [35] and compact Riemannian manifolds [51]. Despite the flexibility of this approach, SM has three important and distinct limitations. Firstly, as the Hyvärinen score depends on the Laplacian of the log-density, SM estimation will be expensive in high dimension and will break down for non-smooth models or for models in which the second derivative grows very rapidly. Secondly, as we shall demonstrate, SM estimators can behave poorly for models with heavy tailed distributions. Thirdly, the SM estimator is not robust to

outliers in many applications of interest. Each of these situations arise naturally for energy models, particularly product-of-experts models and ICA models [33].

In a separate strand of research, new approaches have been developed to measure discrepancy between an unnormalised distribution and a sample. In [23, 25, 50, 24], it was shown that Stein's method can be used to construct discrepancies that control weak convergence of an empirical measure to a target.

In this paper we consider minimum Stein discrepancy (SD) estimators and show that SM, minimum probability flow and contrastive divergence estimators are all special cases. Within this class we focus on SDs constructed from reproducing kernel Hilbert Spaces (RKHS), establishing the consistency, asymptotic normality and robustness of these estimators. We demonstrate that these SDs are appropriate for estimation of non-smooth distributions and heavy- or light- tailed distributions. The remainder of the paper is organized as follows. In Section 2 we introduce the class of minimum SD estimators, then investigate asymptotic properties of SD estimators based on kernels in Section 3, demonstrating consistency and asymptotic normality under general conditions, as well as conditions for robustness. Section 4 presents three toy problems where SM breaks down, but our new estimators are able to recover the truth. All proofs are in the supplementary materials.

## 2 Minimum Stein Discrepancy Estimators

Let $\mathcal{P}_\mathcal{X}$ the set of Borel probability measures on $\mathcal{X}$. Given identical and independent (IID) realisations from $\mathbb{Q} \in \mathcal{P}_\mathcal{X}$ on an open subset $\mathcal{X} \subset \mathbb{R}^d$, the objective is to find a sequence of measures $\mathbb{P}_n$ that approximate $\mathbb{Q}$ in an appropriate sense. More precisely we will consider a family $\mathcal{P}_\Theta = \{\mathbb{P}_\theta : \theta \in \Theta\} \subset \mathcal{P}_\mathcal{X}$ together with a function $D : \mathcal{P}_\mathcal{X} \times \mathcal{P}_\mathcal{X} \to \mathbb{R}_+$ which quantifies the discrepancy between any two measures in $\mathcal{P}_\mathcal{X}$, and wish to estimate an optimal parameter $\theta^*$ satisfying $\theta^* \in \arg\min_{\theta \in \Theta} D(\mathbb{Q}\|\mathbb{P}_\theta)$. In practice, it is often difficult to compute the discrepancy $D$ explicitly, and it is useful to consider a random approximation $\hat{D}(\{X_i\}_{i=1}^n\|\mathbb{P}_\theta)$ based on a IID sample $X_1, \ldots, X_n \sim \mathbb{Q}$, such that $\hat{D}(\{X_i\}_{i=1}^n\|\mathbb{P}_\theta) \xrightarrow{a.s.} D(\mathbb{Q}\|\mathbb{P}_\theta)$ as $n \to \infty$. We then consider the sequence of estimators

$$\hat{\theta}_n^D \in \operatorname{argmin}_{\theta \in \Theta} \hat{D}(\{X_i\}_{i=1}^n\|\mathbb{P}_\theta).$$

The choice of discrepancy will impact the consistency, efficiency and robustness of the estimators. Examples of such estimators include minimum distance estimators [4, 58] where the discrepancy will be a metric on probability measures, including minimum maximum mean discrepancy (MMD) estimation [18, 42, 8] and minimum Wasserstein estimation [19, 21, 6].

More generally, minimum scoring rule estimators [16] arise from proper scoring rules, for example Hyvärinen, Bregman and Tsallis scoring rules. These discrepancies are often statistical divergences, i.e., $D(\mathbb{Q}\|\mathbb{P}) = 0 \Leftrightarrow \mathbb{P} = \mathbb{Q}$ for all $\mathbb{P}, \mathbb{Q}$ in a subset of $\mathcal{P}_\mathcal{X}$. Suppose that $\mathbb{P}_\theta$ and $\mathbb{Q}$ are absolutely continuous with respect to a common measure $\lambda$ on $\mathcal{X}$, with respective positive densities $p_\theta$ and $q$. Then a well-known statistical divergence is the Kullback-Leibler (KL) divergence $\mathrm{KL}(\mathbb{Q}\|\mathbb{P}_\theta) \equiv \int_\mathcal{X} \log(\mathrm{d}\mathbb{Q}/\mathrm{d}\mathbb{P}_\theta)\mathrm{d}\mathbb{Q} = \int_\mathcal{X} \log q \mathrm{d}\mathbb{Q} - \int_\mathcal{X} \log p_\theta \mathrm{d}\mathbb{Q}$. Minimising $\mathrm{KL}(\mathbb{Q}\|\mathbb{P}_\theta)$ is equivalent to maximising $\int_\mathcal{X} \log p_\theta \mathrm{d}\mathbb{Q}$, which can be estimated using the likelihood $\widehat{\mathrm{KL}}(\{X_i\}_{i=1}^n\|\mathbb{P}_\theta) \equiv \frac{1}{n}\sum_{i=1}^n \log p_\theta(X_i)$. Informally, we see that minimising the KL-divergence is equivalent to performing maximum likelihood estimation.

For our purposes we are interested in discrepancies that can be evaluated when $\mathbb{P}_\theta$ is only known up to normalisation, precluding the use of KL divergence. We instead consider a related class of discrepancies based on integral probability pseudometric (IPM) [55] and Stein's method [3, 11, 65]. Let $\Gamma(\mathcal{Y}) \equiv \Gamma(\mathcal{X}, \mathcal{Y}) \equiv \{f : \mathcal{X} \to \mathcal{Y}\}$. A map $\mathcal{S}_\mathbb{P} : \mathcal{G} \subset \Gamma(\mathbb{R}^d) \to \Gamma(\mathbb{R})$ is a Stein operator over a Stein class $\mathcal{G}$ if $\int_\mathcal{X} \mathcal{S}_\mathbb{P}[f]\mathrm{d}\mathbb{P} = 0 \ \forall f \in \mathcal{G}$ for any $\mathbb{P}$. We can then define an associated *Stein discrepancy* (SD) [23] using an IPM with entry-dependent function space $\mathcal{F} \equiv \mathcal{S}_{\mathbb{P}_\theta}[\mathcal{G}]$

$$\mathrm{SD}_{\mathcal{S}_{\mathbb{P}_\theta}[\mathcal{G}]}(\mathbb{Q}\|\mathbb{P}_\theta) \equiv \sup_{f \in \mathcal{S}_{\mathbb{P}_\theta}[\mathcal{G}]}\left|\int_\mathcal{X} f \mathrm{d}\mathbb{P}_\theta - \int_\mathcal{X} f \mathrm{d}\mathbb{Q}\right| = \sup_{g \in \mathcal{G}}\left|\int_\mathcal{X} \mathcal{S}_{\mathbb{P}_\theta}[g]\mathrm{d}\mathbb{Q}\right|. \tag{1}$$

The Stein discrepancy depends on $\mathbb{Q}$ only through expectations, and does not require the existence of a density, therefore permitting $\mathbb{Q}$ to be an empirical measure. If $\mathbb{P}$ has a $C^1$ density $p$ on $\mathcal{X}$, one can consider the Langevin-Stein discrepancy arising from the Stein operator $\mathcal{T}_p[g] \equiv \langle\nabla \log p, g\rangle + \nabla \cdot g$ [23, 25]. In this case, the Stein discrepancy will not depend on the normalising constant of $p$.

In this paper, for an arbitrary $m \in \Gamma(\mathbb{R}^{d \times d})$ which we call *diffusion matrix*, we shall consider the more general *diffusion Stein operators* [25]: $\mathcal{S}_p^m[g] \equiv (1/p)\nabla \cdot (pmg)$, $\mathcal{S}_p^m[A] \equiv (1/p)\nabla \cdot (pmA)$, where $g \in \Gamma(\mathbb{R}^d)$, $A \in \Gamma(\mathbb{R}^{d \times d})$, and the associated *minimum Stein discrepancy estimators* which minimise (1). As we will only have access to a sample $\{X_i\}_{i=1}^n \sim \mathbb{Q}$, we will focus on the estimators minimising an approximation $\widehat{\mathrm{SD}}_{\mathcal{S}_{\mathbb{P}_\theta}[\mathcal{G}]}(\{X_i\}_{i=1}^n \| \mathbb{P}_\theta)$ based on a $U$-statistic of the $\mathbb{Q}$-integral:

$$\hat{\theta}_n^{\mathrm{Stein}} \equiv \mathrm{argmin}_{\theta \in \Theta} \widehat{\mathrm{SD}}_{\mathcal{S}_{\mathbb{P}_\theta}[\mathcal{G}]}(\{X_i\}_i^n \| \mathbb{P}_\theta).$$

Related and complementary approaches to inference using SDs include the nonparametric estimator of [41], the density ratio approach of [47] and the variational inference algorithms of [49, 60]. We now highlight several instances of SDs which will be studied in detail in this paper.

## 2.1 Example 1: Diffusion Kernel Stein Discrepancy Estimators

A convenient choice of Stein class is the unit ball of reproducing kernel Hilbert spaces (RKHS) [5] of a scalar kernel function $k$. For the Langevin Stein operator $\mathcal{T}_p$, the resulting *kernel Stein discrepancy* (KSD) first appeared in [57] and has since been considered extensively in the context of hypothesis testing, measuring sample quality and approximation of probability measures in [12–14, 17, 24, 44, 46, 43]. In this paper, we consider a more general class of discrepancies based on the diffusion Stein operator and matrix-valued kernels.

Consider an RKHS $\mathcal{H}^d$ of functions $f \in \Gamma(\mathbb{R}^d)$ with (matrix-valued) kernel $K \in \Gamma(\mathcal{X} \times \mathcal{X}, \mathbb{R}^{d \times d})$, $K_x \equiv K(x, \cdot)$ (see Appendix A.3 and A.4 for further details). The Stein operator $\mathcal{S}_p^m[f]$ induces an operator $\mathcal{S}_p^{m,2}\mathcal{S}_p^{m,1} : \Gamma(\mathcal{X} \times \mathcal{X}, \mathbb{R}^{d \times d}) \to \Gamma(\mathbb{R})$ which acts first on the first variable and then on the second one. We briefly mention two simple examples of matrix kernels constructed from scalar kernels. If we want the components of $f$ to be orthogonal, we can use the diagonal kernel (i) $K = \mathrm{diag}(\lambda_1 k^1, \ldots, \lambda_d k^d)$ where $\lambda_i > 0$ and $k^i$ is a $C^2$ kernel on $\mathcal{X}$, for $i = 1, \ldots, d$; else we can "correlate" the components by setting (ii) $K = Bk$ where $k$ is a (scalar) kernel on $\mathcal{X}$ and $B$ is a (constant) symmetric positive definite matrix.

We propose to study *diffusion kernel Stein discrepancies* indexed by $K$ and $m$ (see Appendix B):

**Theorem 1** (**Diffusion Kernel Stein Discrepancy**). *For any kernel $K$, we find that $\mathcal{S}_p^m[f](x) = \langle \mathcal{S}_p^{m,1}K_x, f \rangle_{\mathcal{H}^d}$ for any $f \in \mathcal{H}^d$. Moreover if $x \mapsto \|\mathcal{S}_p^{m,1}K_x\|_{\mathcal{H}^d} \in L^1(\mathbb{Q})$, we have*

$$\mathrm{DKSD}_{K,m}(\mathbb{Q}\|\mathbb{P})^2 \equiv \sup_{\substack{h \in \mathcal{H}^d \\ \|h\| \leq 1}} \left| \int_{\mathcal{X}} \mathcal{S}_p^m[h] \mathrm{d}\mathbb{Q} \right|^2 = \int_{\mathcal{X}} \int_{\mathcal{X}} k^0(x,y) \mathrm{d}\mathbb{Q}(x) \mathrm{d}\mathbb{Q}(y)$$

$$k^0(x,y) \equiv \mathcal{S}_p^{m,2}\mathcal{S}_p^{m,1}K(x,y) = \frac{1}{p(y)p(x)}\nabla_y \cdot \nabla_x \cdot \left(p(x)m(x)K(x,y)m(y)^\top p(y)\right). \quad (2)$$

In order to use these for minimum SD estimation, we propose the following $U$-statistic approximation:

$$\widehat{\mathrm{DKSD}}_{K,m}(\{X_i\}_{i=1}^n \| \mathbb{P}_\theta)^2 = \frac{2}{n(n-1)} \sum_{1 \leq i < j \leq n} k_\theta^0(X_i, X_j) = \frac{1}{n(n-1)} \sum_{i \neq j} k_\theta^0(X_i, X_j), \quad (3)$$

with associated estimators: $\hat{\theta}_n^{\mathrm{DKSD}} \in \mathrm{argmin}_{\theta \in \Theta} \widehat{\mathrm{DKSD}}_{K,m}(\{X_i\}_{i=1}^n \| \mathbb{P}_\theta)^2$.

As the proof shows, the Stein kernel $k^0$ is indeed a (scalar) kernel obtained from the feature map $\phi : \mathcal{X} \to \mathcal{H}^d$, $\phi(x) \equiv \mathcal{S}_p^{m,1}[K]|_x$. For $K = Ik$, $m = Ih$, DKSD is a KSD with scalar kernel $h(x)k(x,y)h(y)$, and if $h = 1$ our objective becomes the usual Langevin-based KSD of [14, 24, 46, 57] (see Appendix B.4). The work of [45] discussed the potential of optimizing the KSD with gradient descent but did not evaluate its merits. In the sections to follow, we will see the advantages conferred by introducing more flexible diffusion operators, matrix kernels, and Riemannian optimization.

Now that our DKSD estimators are defined, an important remaining question is under which conditions can DKSD discriminate distinct probability measures. To answer, we will need several definitions. We say a matrix kernel $K$ is in the Stein class of $\mathbb{Q}$ if $\int_{\mathcal{X}} \mathcal{S}_q^{m,1}[K] \mathrm{d}\mathbb{Q} = 0$, and that it is strictly integrally positive definite (IPD) if $\int_{\mathcal{X} \times \mathcal{X}} \mathrm{d}\mu^\top(x) K(x,y) \mathrm{d}\mu(y) > 0$ for any finite non-zero signed vector Borel measure $\mu$. From $\mathcal{S}_p^m[f](x) = \langle \mathcal{S}_p^{m,1}K_x, f \rangle_{\mathcal{H}^d}$ we have that $f \in \mathcal{H}^d$ is in the Stein class (i.e., $\int_{\mathcal{X}} \mathcal{S}_q^m[f] \mathrm{d}\mathbb{Q} = 0$) when $K$ is also in the class. Setting $s_p \equiv m^\top \nabla \log p \in \Gamma(\mathbb{R}^d)$:

**Proposition 1** (**DKSD as a Statistical Divergence**). *Suppose $K$ is IPD and in the Stein class of $\mathbb{Q}$, and $m(x)$ is invertible. If $s_p - s_q \in L^1(\mathbb{Q})$, then $\mathrm{DKSD}_{K,m}(\mathbb{Q}\|\mathbb{P})^2 = 0$ iff $\mathbb{Q} = \mathbb{P}$.*

See Appendix B.5 for the proof. Note that this proposition generalises Proposition 3.3 from [46] to a significantly larger class of SD. For the matrix kernels introduced above, the proposition below shows that $K$ is IPD when its associated scalar kernels are; a well-studied problem [64].

**Proposition 2** (**IPD Matrix Kernels**). *(i) Let $K = diag(k^1, \ldots, k^d)$. Then $K$ is IPD iff each kernel $k^i$ is IPD. (ii) Let $K = Bk$ for $B$ be symmetric positive definite. Then $K$ is IPD iff $k$ is IPD.*

## 2.2 Example 2: Diffusion Score Matching Estimators

A well-known family of estimators are the score matching (SM) estimators (based on the Fisher or Hyvarinen divergence) [34, 35]. As will be shown below, these can be seen as special cases of minimum SD estimators. The SM discrepancy is computable for sufficiently smooth densities:

$$\mathrm{SM}(\mathbb{Q}\|\mathbb{P}) \equiv \int_{\mathcal{X}} \|\nabla \log p - \nabla \log q\|_2^2 \, \mathrm{d}\mathbb{Q} = \int_{\mathcal{X}} \big(\|\nabla \log q\|_2^2 + \|\nabla \log p\|_2^2 + 2\Delta \log p\big)\mathrm{d}\mathbb{Q}$$

where $\Delta$ denotes the Laplacian and we have used the divergence theorem. If $\mathbb{P} = \mathbb{P}_\theta$, the first integral above does not depend on $\theta$, and the second one does not depend on the density of $\mathbb{Q}$, so we consider the approximation $\widehat{\mathrm{SM}}(\{X_i\}_{i=1}^n\|\mathbb{P}_\theta) \equiv \frac{1}{n}\sum_{i=1}^n \Delta \log p_\theta(X_i) + \frac{1}{2}\|\nabla \log p_\theta(X_i)\|_2^2$ based on an unbiased estimation for the minimiser of the SM divergence, and its estimators $\hat{\theta}_n^{\mathrm{SM}} \equiv \mathrm{argmin}_{\theta \in \Theta}\widehat{\mathrm{SM}}(\{X_i\}_{i=1}^n\|\mathbb{P}_\theta)$, for independent random vectors $X_i \sim \mathbb{Q}$.

The SM discrepancy can also be generalised to include higher-order derivatives of the log-likelihood [48] and does not require a normalised model. We will now introduce a further generalisation that we call *diffusion score matching (DSM)* which is a SD constructed from the diffusion Stein operator (see Appendix B.6):

**Theorem 2** (**Diffusion Score Matching**). *Let $\mathcal{X} = \mathbb{R}^d$ and consider some diffusion Stein operator $\mathcal{S}_p^m$ for some function $m \in \Gamma(\mathbb{R}^{d \times d})$ and the Stein class $\mathcal{G} \equiv \{g = (g_1, \ldots, g_d) \in C^1(\mathcal{X}, \mathbb{R}^d) \cap L^2(\mathcal{X}; \mathbb{Q}) : \|g\|_{L^2(\mathcal{X};\mathbb{Q})} \leq 1\}$. If $p, q > 0$ are differentiable and $s_p - s_q \in L^2(\mathbb{Q})$, then we define the diffusion score matching divergence as the Stein discrepancy,*

$$\mathrm{DSM}_m(\mathbb{Q}\|\mathbb{P}) \equiv \sup_{f \in \mathcal{S}_p[\mathcal{G}]}\big|\int_{\mathcal{X}} f\mathrm{d}\mathbb{Q} - \int_{\mathcal{X}} f\mathrm{d}\mathbb{P}\big|^2 = \int_{\mathcal{X}}\big\|m^\top(\nabla \log q - \nabla \log p)\big\|_2^2\mathrm{d}\mathbb{Q}.$$

*This satisfies $\mathrm{DSM}_m(\mathbb{Q}\|\mathbb{P}) = 0$ iff $\mathbb{Q} = \mathbb{P}$ when $m(x)$ is invertible. Moreover, if $p$ is twice-differentiable, and $qmm^\top\nabla \log p, \nabla \cdot (qmm^\top\nabla \log p) \in L^1(\mathbb{R}^d)$, then Stoke's theorem gives*

$$\mathrm{DSM}_m(\mathbb{Q}\|\mathbb{P}) = \int_{\mathcal{X}}\big(\|m^\top\nabla_x \log p\|_2^2 + \|m^\top\nabla \log q\|_2^2 + 2\nabla \cdot \big(mm^\top\nabla \log p\big)\big)\mathrm{d}\mathbb{Q}.$$

Notably, $\mathrm{DSM}_m$ recovers SM when $m(x)m(x)^\top = I$ and the (generalised) non-negative score matching estimator of [48] with the choice $m(x) \equiv diag(h_1(x_1)^{1/2}, \ldots, h_d(x_d)^{1/2})$. Like standard SM, DSM is only defined for distributions with sufficiently smooth densities. Since the $\theta$-dependent part of $\mathrm{DSM}_m(\mathbb{Q}\|\mathbb{P}_\theta)$ does not depend on the density of $\mathbb{Q}$, and can be estimated using an empirical mean, leading to the estimators $\hat{\theta}_n^{\mathrm{DSM}} \equiv \mathrm{argmin}_{\theta \in \Theta}\widehat{\mathrm{DSM}}_m(\{X_i\}_{i=1}^n\|\mathbb{P}_\theta)$ for

$$\widehat{\mathrm{DSM}}_m(\{X_i\}_{i=1}^n\|\mathbb{P}_\theta) \equiv \frac{1}{n}\sum_{i=1}^n\big(\|m^\top\nabla_x \log p_\theta\|_2^2 + 2\nabla \cdot \big(mm^\top\nabla \log p_\theta\big)\big)(X_i)$$

where $\{X_i\}_{i=1}^n$ is a sample from $\mathbb{Q}$. Note that this is only possible if $m$ is independent of $\theta$, in contrast to DKSD where $m$ can depend on $\mathcal{X} \times \Theta$, thus leading to a more flexible class of estimators.

An interesting remark is that the $\mathrm{DSM}_m$ discrepancy may in fact be obtained as a limit of DKSD over a sequence of target-dependent kernels: see Appendix B.6 for the complete result which corrects and significantly generalises previously established connections between the SM divergence and KSD (such as in Sec. 5 of [46]).

We conclude by commenting on the computational complexity. Evaluating the DKSD loss function requires $\mathcal{O}(n^2d^2)$ computation, due to the U-statistic and a matrix-matrix product. However, if $K = \mathrm{diag}(\lambda_1 k^1, \ldots, \lambda_d k^d)$ or $K = Bk$, and if $m$ is a diagonal matrix, then we can by-pass expensive matrix products and the cost is $\mathcal{O}(n^2d)$, making it comparable to that of KSD. Although we do not consider these in this paper, recent approximations to KSD could also be adapted to DKSD to reduce the computational cost to $\mathcal{O}(nd)$ [32, 36]. The DSM loss function has computational cost $\mathcal{O}(nd^2)$, which is comparable to the SM loss. From a computational viewpoint, DSM will hence be preferable to DKSD for large $n$, whilst DKSD will be preferable to DSM for large $d$.

## 2.3  Further Examples: Contrastive Divergence and Minimum Probability Flow

Before analysing DKSD and DSM estimators further, we show that the class of minimum SD estimators also includes other well-known estimators for unnormalised models. Let $X_\theta^n$, $n \in \mathbb{N}$ be a Markov process with unique invariant probality measure $\mathbb{P}_\theta$, for example a Metropolis-Hastings chain. Let $P_\theta^n$ be the associated transition semigroup, i.e. $(P_\theta^n f)(x) = \mathbb{E}[f(X_\theta^n)|X_\theta^0 = x]$. Choosing the Stein operator $\mathcal{S}_p = I - P_\theta^n$ and Stein class $\mathcal{G} = \{\log p_\theta + c \,:\, c \in \mathbb{R}\}$, leads to the following SD:

$$\mathrm{CD}(\mathbb{Q}\|\mathbb{P}_\theta) = \int_{\mathcal{X}} (\log p_\theta - P_\theta^n \log p_\theta) \mathrm{d}\mathbb{Q} = \mathrm{KL}(\mathbb{Q}\|\mathbb{P}_\theta) - \mathrm{KL}(\mathbb{Q}_\theta^n\|\mathbb{P}_\theta),$$

where $\mathbb{Q}_\theta^n$ is the law of $X_\theta^n|X_\theta^0 \sim \mathbb{Q}$ and assuming that $\mathbb{Q} \ll \mathbb{P}_\theta$ and $\mathbb{Q}_\theta^n \ll \mathbb{P}_\theta$, which is the loss function associated with contrastive divergence (CD) [28, 45]. Suppose now that $\mathcal{X}$ is a finite set. Given $\theta \in \Theta$ let $P_\theta$ be the transition matrix for a Markov process with unique invariant distribution $\mathbb{P}_\theta$. Suppose we observe data $\{x_i\}_{i=1}^n$ and let $q$ be the corresponding empirical distribution. Choosing the Stein operator $\mathcal{S}_p = I - P_\theta$ and the Stein set $\mathcal{G} = \{f \in \Gamma(\mathbb{R}) \,:\, \|f\|_\infty \leq 1\}$. Note that, $g \in \arg\sup_{g \in G} |\mathbb{Q}(\mathcal{S}_p[g])|$ will satisfy $g(i) = \mathrm{sgn}(q^\top(I - P_\theta)_i)$, and the resulting Stein discrepancy is the minimum probability flow loss objective function [62]:

$$\mathrm{MPFL}(\mathbb{Q}\|\mathbb{P}) = \sum_y |((I - P_\theta)^\top q)_y| = \sum_{y \notin \{x_i\}_{i=1}^n} \left| \frac{1}{n} \sum_{x \in \{x_i\}_{i=1}^n} (I - P_\theta)_{xy} \right|.$$

## 2.4  Implementing Minimum SD Estimators: Stochastic Riemannian Gradient Descent

In order to implement the minimum SD estimators, we propose to use a stochastic gradient descent (SGD) algorithm associated to the information geometry induced by the SD on the parameter space. More precisely, consider a parametric family $\mathcal{P}_\Theta$ of probability measures on $\mathcal{X}$ with $\Theta \subset \mathbb{R}^m$. Given a discrepancy $D : \mathcal{P}_\Theta \times \mathcal{P}_\Theta \to \mathbb{R}$ satisfying $D(\mathbb{P}_\alpha\|\mathbb{P}_\theta) = 0$ iff $\mathbb{P}_\alpha = \mathbb{P}_\theta$ (called a statistical divergence), its associated information matrix field on $\Theta$ is defined as the map $\theta \mapsto g(\theta)$, where $g(\theta)$ is the symmetric bilinear form $g(\theta)_{ij} = -\frac{1}{2}(\partial^2/\partial\alpha^i\partial\theta^j)D(\mathbb{P}_\alpha\|\mathbb{P}_\theta)|_{\alpha=\theta}$ [2]. When $g$ is positive definite, we can use it to perform (Riemannian) gradient descent on the parameter space $\Theta$. We provide below the information matrices of DKSD and DSM (and hence extends results of [37]):

**Proposition 3** (**Information Tensor DKSD**). *Assume the conditions of Proposition 1 hold. The information tensor associated to DKSD is positive semi-definite and has components*

$$g_{\mathrm{DKSD}}(\theta)_{ij} = \int_{\mathcal{X}} \int_{\mathcal{X}} (\nabla_x \partial_{\theta^j} \log p_\theta(x))^\top m_\theta(x) K(x,y) m_\theta^\top(y) \nabla_y \partial_{\theta^i} \log p_\theta(y) \mathrm{d}\mathbb{P}_\theta(x) \mathrm{d}\mathbb{P}_\theta(y).$$

**Proposition 4** (**Information Tensor DSM**). *Assume the conditions of Theorem 2 hold. The information tensor defined by DSM is positive semi-definite and has components*

$$g_{\mathrm{DSM}}(\theta)_{ij} = \int_{\mathcal{X}} \langle m^\top \nabla \partial_{\theta^i} \log p_\theta, m^\top \nabla \partial_{\theta^j} \log p_\theta \rangle \mathrm{d}\mathbb{P}_\theta.$$

See Appendix C for the proofs. Given an (information) Riemannian metric, recall the gradient flow of a curve $\theta$ on the Riemannian manifold $\Theta$ is the solution to $\dot\theta(t) = -\nabla_{\theta(t)} \mathrm{SD}(\mathbb{Q}\|\mathbb{P}_\theta)$, where $\nabla_\theta$ denotes the Riemannian gradient at $\theta$. It is the curve that follows the direction of steepest decrease (measured with respect to the Riemannian metric) of the function $\mathrm{SD}(\mathbb{Q}\|\mathbb{P}_\theta)$ (see Appendix A.5). The well-studied natural gradient descent [1, 2] corresponds to the case in which the Riemannian manifold is $\Theta = \mathbb{R}^m$ equipped with the Fisher metric and SD is replaced by KL. When $\Theta$ is a linear manifold with coordinates $(\theta^i)$ we have $\nabla_\theta \mathrm{SD}(\mathbb{Q}\|\mathbb{P}_\theta) = g(\theta)^{-1}\mathrm{d}_\theta \mathrm{SD}(\mathbb{Q}\|\mathbb{P}_\theta)$, where $\mathrm{d}_\theta f$ denotes the tuple $(\partial_{\theta^i} f)$. We will approximate this at step $t$ of the descent using the biased estimator $\hat{g}_{\theta_t}(\{X_i^t\}_i)^{-1}\mathrm{d}_{\theta_t}\widehat{\mathrm{SD}}(\{X_i^t\}_{i=1}^n\|\mathbb{P}_\theta)$, where $\hat{g}_{\theta_t}(\{X_i^t\}_{i=1}^n)$ is an unbiased estimator for the information matrix $g(\theta_t)$ and $\{X_i^t \sim \mathbb{Q}\}_i$ is a sample at step $t$. In general, we have no guarantee that $\hat{g}_{\theta_t}$ is invertible, and so we may need a further approximation step to obtain an invertible matrix. Given a sequence $(\gamma_t)$ of step sizes we will approximate the gradient flow with

$$\hat\theta_{t+1} = \hat\theta_t - \gamma_t \hat{g}_{\theta_t}(\{X_i^t\}_{i=1}^n)^{-1}\mathrm{d}_{\theta_t}\widehat{\mathrm{SD}}(\{X_i^t\}_{i=1}^n\|\mathbb{P}_\theta).$$

Minimum SD estimators hold additional appeal for exponential family models, since their densities have the form $p_\theta(x) \propto \exp(\langle\theta, T(x)\rangle_{\mathbb{R}^m})\exp(b(x))$ for natural parameters $\theta \in \mathbb{R}^m$, sufficient statistics $T \in \Gamma(\mathbb{R}^m)$, and base measure $\exp(b(x))$. For these models, the U-statistic approximations of DKSD and DSM are convex quadratics with closed form solutions whenever $K$ and $m$ are independent of $\theta$. Moreover, since the absolute value of an affine function is convex, and the supremum of convex functions is convex, any SD with a diffusion Stein operator is convex in $\theta$, provided $m$ and the Stein class $\mathcal{G}$ are independent of $\theta$.

# 3   Theoretical Properties for Minimum Stein Discrepancy Estimators

We now show that the DKSD and DSM estimators have many desirable properties such as consistency, asymptotic normality and bias-robustness. These results do not only provide us with reassuring theoretical guarantees on the performance of our algorithms, but can also be a practical tool for choosing a Stein operator and Stein class given an inference problem of interest.

We begin by establishing strong consistency and for DKSD; i.e. almost sure convergence: $\hat{\theta}_n^{\text{DKSD}} \xrightarrow{a.s.} \theta_*^{\text{DKSD}} \equiv \operatorname{argmin}_{\theta \in \Theta} \text{DKSD}_{K,m}(\mathbb{Q} \| \mathbb{P}_\theta)^2$. This will be followed by a proof of asymptotic normality. We will assume we are in the specified setting, so that $\mathbb{Q} = \mathbb{P}_{\theta^{\text{DKSD}}_*} \in \mathcal{P}_\Theta$. In the misspecified setting, we will need to also assume the existence of a unique minimiser.

**Theorem 3** (**Strong Consistency of DKSD**). *Let $\mathcal{X} = \mathbb{R}^d$, $\Theta \subset \mathbb{R}^m$. Suppose that $K$ is bounded with bounded derivatives up to order $2$, that $k^0(x, y)$ is continuously-differentiable on an $\mathbb{R}^m$-open neighbourhood of $\Theta$, and that for any compact subset $C \subset \Theta$ there exist functions $f_1, f_2, g_1, g_2$ such that for $\mathbb{Q}$-a.e. $x \in \mathcal{X}$,*

1. *$\left\| m^\top(x) \nabla \log p_\theta(x) \right\| \le f_1(x)$, where $f_1 \in L^1(\mathbb{Q})$ and continuous,*

2. *$\left\| \nabla_\theta \big( m(x)^\top \nabla \log p_\theta(x) \big) \right\| \le g_1(x)$, where $g_1 \in L^1(\mathbb{Q})$ is continuous,*

3. *$\| m(x) \| + \| \nabla_x m(x) \| \le f_2(x)$ where $f_2 \in L^1(\mathbb{Q})$ and continuous,*

4. *$\| \nabla_\theta m(x) \| + \| \nabla_\theta \nabla_x m(x) \| \le g_2(x)$ where $g_2 \in L^1(\mathbb{Q})$ is continuous.*

*Assume further that $\theta \mapsto \mathbb{P}_\theta$ is injective. Then we have a unique minimiser $\theta_*^{\text{DKSD}}$, and if either $\Theta$ is compact, or $\theta_*^{\text{DKSD}} \in int(\Theta)$ and $\Theta$ and $\theta \mapsto \widehat{\text{DKSD}}_{K,m}(\{X_i\}_{i=1}^n \| \mathbb{P}_\theta)^2$ are convex, then $\hat{\theta}_n^{\text{DKSD}}$ is strongly consistent.*

**Theorem 4** (**Central Limit Theorem for DKSD**). *Let $\mathcal{X}$ and $\Theta$ be open subsets of $\mathbb{R}^d$ and $\mathbb{R}^m$ respectively. Let $K$ be a bounded kernel with bounded derivatives up to order $2$ and suppose that $\hat{\theta}_n^{\text{DKSD}} \xrightarrow{p} \theta_*^{\text{DKSD}}$ and that there exists a compact neighbourhood $\mathcal{N} \subset \Theta$ of $\theta_*^{\text{DKSD}}$ such that $\theta \to \widehat{\text{DKSD}}_{K,m}(\{X_i\}_{i=1}^n, \mathbb{P}_\theta)^2$ is twice continuously differentiable for $\theta \in \mathcal{N}$ and, for $\mathbb{Q}$-a.e. $x \in \mathcal{X}$,*

1. *$\| m^\top(x) \nabla \log p_\theta(x) \| + \| \nabla_\theta \big( m(x)^\top \nabla \log p_\theta(x) \big) \| \le f_1(x)$,*

2. *$\| m(x) \| + \| \nabla_x m(x) \| + \| \nabla_\theta m(x) \| + \| \nabla_\theta \nabla_x m(x) \| \le f_2(x)$,*

3. *$\| \nabla_\theta \nabla_\theta \big( m(x)^\top \nabla \log p_\theta(x) \big) \| + \| \nabla_\theta \nabla_\theta \nabla_\theta \big( m(x)^\top \nabla \log p_\theta(x) \big) \| \le g_1(x)$,*

4. *$\| \nabla_\theta \nabla_\theta m(x) \| + \| \nabla_\theta \nabla_\theta \nabla_x m(x) \| + \| \nabla_\theta \nabla_\theta \nabla_\theta m(x) \| + \| \nabla_\theta \nabla_\theta \nabla_\theta \nabla_x m(x) \| \le g_2(x)$,*

*where $f_1, f_2 \in L^2(\mathbb{Q}), g_1, g_2 \in L^1(\mathbb{Q})$ are continuous. Suppose also that the information tensor $g$ is invertible at $\theta_*^{\text{DKSD}}$. Then*

$$\sqrt{n}\Big(\hat{\theta}_n^{\text{DKSD}} - \theta_*^{\text{DKSD}}\Big) \xrightarrow{d} \mathcal{N}\big(0, g_{\text{DKSD}}^{-1}(\theta_*^{\text{DKSD}}) \Sigma_{\text{DKSD}} g_{\text{DKSD}}^{-1}(\theta_*^{\text{DKSD}})\big),$$

*where $\Sigma_{\text{DKSD}} = \int_{\mathcal{X}} \Big( \int_{\mathcal{X}} \nabla_\theta k_{\theta_*^{\text{DKSD}}}^0(x, y) \mathrm{d}\mathbb{Q}(y) \Big) \otimes \Big( \int_{\mathcal{X}} \nabla_\theta k_{\theta_*^{\text{DKSD}}}^0(x, z) \mathrm{d}\mathbb{Q}(z) \Big) \mathrm{d}\mathbb{Q}(x)$.*

See Appendix D for proofs. For both results, the assumptions on the kernel are satisfied by most kernels common in the literature, such as Gaussian, inverse-multiquadric (IMQ) and any Matérn kernels with smoothness greater than 2. Similarly, the assumptions on the model are very weak given that the diffusion tensor $m$ can be adapted to guarantee consistency and asymptotic normality.

We now prove analogous results for DSM. This time we show weak consistency, i.e. convergence in probability: $\hat{\theta}_n^{\text{DSM}} \xrightarrow{p} \theta_*^{\text{DSM}} \equiv \operatorname{argmin}_{\theta \in \Theta} \text{DSM}_m(\mathbb{Q} \| \mathbb{P}_\theta) = \operatorname{argmin}_{\theta \in \Theta} \int_{\mathcal{X}} F_\theta(x) \mathrm{d}\mathbb{Q}(x)$. This will be a sufficient form of convergence for asymptotic normality.

**Theorem 5** (**Weak Consistency of DSM**). *Let $\mathcal{X}$ be an open subset of $\mathbb{R}^d$, and $\Theta \subset \mathbb{R}^m$. Suppose $\log p_\theta(\cdot) \in C^2(\mathcal{X})$ and $m \in C^1(\mathcal{X})$, and $\| \nabla_x \log p_\theta(x) \| \le f_1(x)$ for $\mathbb{Q}$-a.e. $x$. Suppose also that $\| \nabla_x \nabla_x \log p_\theta(x) | \le f_2(x)$ on any compact set $C \subset \Theta$ for $\mathbb{Q}$-a.e. $x$, where $\| m^\top \| f_1 \in L^2(\mathbb{Q})$, $\| \nabla \cdot (mm^\top) \| f_1 \in L^1(\mathbb{Q})$, $\| mm^\top \|_\infty f_2 \in L^1(\mathbb{Q})$. If either $\Theta$ is compact, or $\Theta$ and $\theta \mapsto F_\theta$ are convex and $\theta_*^{\text{DSM}} \in int(\Theta)$, then $\hat{\theta}_n^{\text{DSM}}$ is weakly consistent for $\theta_*^{\text{DSM}}$.*

**Theorem 6 (Central Limit Theorem for DSM).** *Let $\mathcal{X}, \Theta$ be open subsets of $\mathbb{R}^d$ and $\mathbb{R}^m$ respectively. Suppose $\hat{\theta}_n^{\mathrm{DSM}} \xrightarrow{p} \theta_*^{\mathrm{DSM}}$, $\theta \mapsto \log p_\theta(x)$ is twice continuously differentiable on a closed ball $\bar{B}(\epsilon, \theta_*^{\mathrm{DSM}}) \subset \Theta$, and that for $\mathbb{Q}$-a.e. $x \in \mathcal{X}$,*

*(i)* $\|m(x)m^\top(x)\| + \|\nabla_x \cdot (m(x)m^\top(x))\| \leq f_1(x)$, *and* $\|\nabla_x \log p_\theta(x)\| + \|\nabla_\theta \nabla_x \log p_\theta(x)\| + \|\nabla_\theta \nabla_x \nabla_x \log p_\theta(x)\| \leq f_2(x)$, *with* $f_1 f_2, f_1 f_2^2 \in L^2(\mathbb{Q})$

*(ii)* *for* $\theta \in \bar{B}(\epsilon, \theta^*)$, $\|\nabla_\theta \nabla_x \log p_\theta\|^2 + \|\nabla_x \log p_\theta\| \|\nabla_\theta \nabla_\theta \nabla_x \log p_\theta\| + \|\nabla_\theta \nabla_\theta \nabla_x \log p_\theta\| + \|\nabla_\theta \nabla_\theta \nabla_x \nabla_x \log p_\theta\| \leq g_1(x)$, *and* $f_1 g_1 \in L^1(\mathbb{Q})$.

*Then, if the information tensor is invertible at $\theta_*^{\mathrm{DSM}}$, we have*

$$\sqrt{n}\Big(\hat{\theta}_n^{\mathrm{DSM}} - \theta_*^{\mathrm{DSM}}\Big) \xrightarrow{d} \mathcal{N}\big(0, g_{\mathrm{DSM}}^{-1}(\theta^{\mathrm{DSM}}) \Sigma_{\mathrm{DSM}} g_{\mathrm{DSM}}^{-1}(\theta_*^{\mathrm{DSM}})\big).$$

*where* $\Sigma_{\mathrm{DSM}} = \int_{\mathcal{X}} \nabla_\theta F_{\theta_*^{\mathrm{DSM}}}(x) \otimes \nabla_\theta F_{\theta_*^{\mathrm{DSM}}}(x) \mathrm{d}\mathbb{Q}(x)$.

All of the proofs can be found in Appendix D.2. An important special case covered by our theory is that of natural exponential families, which admit densities of the form $\log p_\theta(x) \propto \langle \theta, T(x) \rangle_{\mathbb{R}^m} + b(x)$. If $K$ is IPD with bounded derivative up to order 2, $\nabla T$ has linearly independent rows, $m$ is invertible, and $\|\nabla Tm\|, \|\nabla_x b\| \|m\|, \|\nabla_x m\| + \|m\| \in L^2(\mathbb{Q})$, then the sequence of minimum DKSD and DSM estimators are strongly consistent and asymptotically normal (see Appendix D.3).

Before concluding this section, we turn to a concept of importance to practical inference: robustness when subjected to corrupted data [31]. We quantify the robustness of DKSD and DSM estimators in terms of their influence function, which can be interpreted as measuring the impact of an infinitesimal perturbation of a distribution $\mathbb{P}$ by a Dirac located at a point $z \in \mathcal{X}$ on the estimator. If $\theta_{\mathbb{Q}}$ denotes the unique minimum SD estimator for $\mathbb{Q}$, then the influence functions is given by $\mathrm{IF}(z, \mathbb{Q}) \equiv \partial_t \theta_{\mathbb{Q}_t}|_{t=0}$ if it exists, where $\mathbb{Q}_t = (1-t)\mathbb{Q} + t\delta_z$, for $t \in [0,1]$. An estimator is said to be bias robust if $\mathrm{IF}(z, \mathbb{Q})$ is bounded in $z$.

**Proposition 7 (Robustness of DKSD estimators).** *Suppose that the map $\theta \to \mathbb{P}_\theta$ over $\Theta$ is injective, then* $\mathrm{IF}(z, \mathbb{P}_\theta) = g_{\mathrm{DKSD}}(\theta)^{-1} \int_X \nabla_\theta k^0(z, y) \mathrm{d}\mathbb{P}_\theta(y)$. *Moreover, suppose that $y \mapsto F(x, y)$ is $\mathbb{Q}$-integrable for any $x$, where $F(x, y) = \|K(x,y)s_p(y)\|, \|K(x,y)\nabla_\theta s_p(y)\|, \|\nabla_x K(x,y)s_p(y)\|, \|\nabla_x K(x,y)\nabla_\theta s_p(y)\|, \|\nabla_y \nabla_x (K(x,y)m(y))\|, \|\nabla_y \nabla_x (K(x,y)\nabla_\theta m(y))\|$. Then if $x \mapsto (\|s_p(x)\| + \|\nabla_\theta s_p(x)\|) \int F(x,y)\mathbb{Q}(dy)|_{\theta_*^{\mathrm{DKSD}}}$ is bounded, the DKSD estimators are bias robust:* $\sup_{z \in \mathcal{X}} \|\mathrm{IF}(z, \mathbb{Q})\| < \infty$.

The analogous results for DSM estimators can be found in Appendix E. Consider a Gaussian location model, i.e. $p_\theta \propto \exp(-\|x - \theta\|_2^2)$, for $\theta \in \mathbb{R}^d$. The Gaussian kernel satisfies the assumptions of Proposition 7 so that $\sup_z \|\mathrm{IF}(z, \mathbb{Q})\| < \infty$, even when $m = I$. Indeed $\|\mathrm{IF}(z, \mathbb{P}_\theta)\| \leq C(\theta)e^{-\|z-\theta\|^2/4}\|z - \theta\|$, where $z \mapsto e^{-\|z-\theta\|^2/4}\|z - \theta\|$ is uniformly bounded over $\theta$. In contrast, the SM estimator has an influence function of the form $\mathrm{IF}(z, \mathbb{Q}) = z - \int_{\mathcal{X}} x \mathrm{d}\mathbb{Q}(x)$, which is unbounded with respect to $z$, and is thus not robust. This clearly demonstrates the importance of carefully selecting a Stein class for use in minimum SD estimators. An alternative way of inducing robustness is to introduce a spatially decaying diffusion matrix in DSM. To this end, consider the minimum DSM estimator with scalar diffusion coefficient $m$. Then $\theta_{\mathrm{DSM}} = (\int_{\mathcal{X}} m^2(x)\mathrm{d}\mathbb{Q}(x))^{-1}\big(\int_{\mathcal{X}} m^2(x)x\mathrm{d}\mathbb{Q}(x) + \int_{\mathcal{X}} \nabla m^2(x)\mathrm{d}\mathbb{Q}(x)\big)$. A straightforward calculation yields that the associated influence function will be bounded if both $m(x)$ and $\|\nabla m(x)\|$ decay as $\|x\| \to \infty$. This clearly demonstrates another significant advantage provided by the flexibility of our family of diffusion SD, where the Stein operator also plays an important role.

## 4 Numerical Experiments

In this section, we explore several examples which demonstrate worrying breakpoints for SM, and highlight how these can be straightforwardly handled using KSD, DKSD and DSM.

### 4.1 Rough densities: the symmetric Bessel distributions

A major drawback of SM is the smoothness requirement on the target density. However, this can be remedied by choosing alternative Stein classes, as will be demonstrated in the case of the symmetric

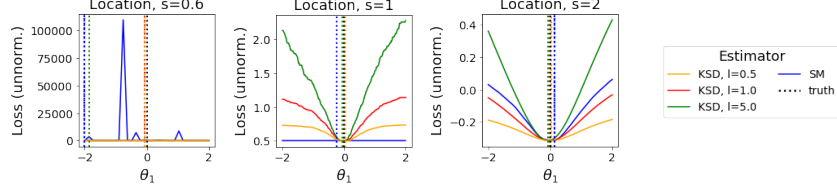

Figure 1: *Minimum SD Estimators for the Symmetric Bessel Distribution*. We consider the case where $\theta_1^* = 0$ and $\theta_2^* = 1$ and $n = 500$ for a range of smoothness parameter values $s$ in $d = 1$.

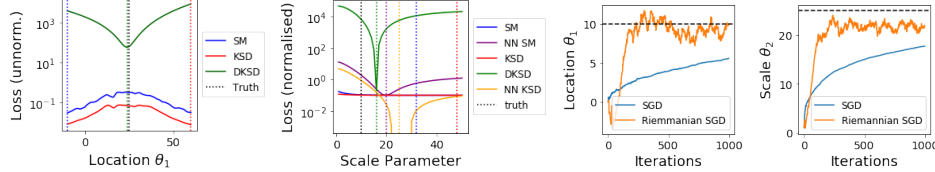

Figure 2: *Minimum SD Estimators for Non-standardised Student-t Distributions*. We consider a student-t problem with $\nu = 5, \theta_1^* = 25, \theta_2^* = 10$ and $n = 300$.

**Bessel distributions.** Let $K_{s-d/2}$ denote the modified Bessel function of the second kind with parameter $s - d/2$. This distribution generalises the Laplace distribution [40] and has log-density: $\log p_\theta(x) \propto (\|x - \theta_1\|_2/\theta_2)^{(s-d/2)} K_{s-d/2}(\|x - \theta_1\|_2/\theta_2)$ where $\theta_1 \in \mathbb{R}^d$ is a location parameter and $\theta_2 > 0$ a scale parameter. The parameter $s \geq d/2$ encodes smoothness.

We compared SM with KSD based on a Gaussian kernel and a range of lengthscale values in Fig. 1. These results are based on $n = 500$ IID realisations in $d = 1$. The case $s = 1$ corresponds to a Laplace distribution, and we notice that both SM and KSD are able to obtain a reasonable estimate of the location. For rougher values, for example $s = 0.6$, we notice that KSD outperforms SM for certain choices of lengthscales, whereas for $s = 2$, SM and KSD are both able to recover the parameter. Analogous results for scale can be found in Appendix F.1, and Appendix F.2 illustrates the trade-off between efficiency and robustness on this problem.

### 4.2 Heavy-tailed distributions: the non-standardised student-t distributions

A second drawback of standard SM is that it is inefficient for heavy-tailed distributions. To demonstrate this, we focus on non-standardised student-t distributions: $p_\theta(x) \propto (1/\theta_2)(1 + (1/\nu)\|x - \theta_1\|_2^2/\theta_2^2)^{-(\nu+1)/2}$ where $\theta_1 \in \mathbb{R}$ is a location parameter and $\theta_2 > 0$ a scale parameter. The parameter $\nu$ determines the degrees of freedom: when $\nu = 1$, we have a Cauchy distribution, whereas $\nu = \infty$ gives the Gaussian distribution. For small values of $\nu$, the student-t distribution is heavy-tailed.

We illustrate SM and KSD for $\nu = 5$ in Fig. 2, where we take an IMQ kernel $k(x, y; c, \beta) = (c^2 + \|x - y\|_2^2)^\beta$ with $c = 1.$ and $\beta = -0.5$. This choice of $\nu$ guarantees the first two moments exist, but the distribution is still heavy-tailed. In the left plot, both SM and KSD struggle to recover $\theta_1^*$ when $n = 300$, and the loss functions are far from convex. However, DKSD with $m_\theta(x) = 1 + \|x - \theta_1\|^2/\theta_2^2$ can estimate $\theta_1$ very accurately. In the middle left plot, we instead estimate $\theta_2$ with SM, KSD and their correponding non-negative version (NNSM & NNKSD, $m(x) = x$), which are particularly well suited for scale parameters. NNSM and NNKSD provide improvements on SM and KSD, but DKSD with $m_\theta(x) = ((x - \theta_1)/\theta_2)(1 + (1/\nu)\|x - \theta_1\|_2^2/\theta_2^2)$ provides significant further gains. On the right-hand side, we also consider the advantage of the Riemannian SGD algorithm over SGD by illustrating them on the KSD loss function with $n = 1000$. Both algorithms use constant stepsizes and minibatches of size 50. As demonstrated, Riemmannian SGD converges within a few dozen iterations, whereas SGD hasn't converged after 1000 iterations. Additional experiments on the robustness of these estimators is also available in Appendix F.2.

### 4.3 Robust estimators for light-tailed distributions: the generalised Gamma distributions

Our final example demonstrates a third failure mode for SM: its lack of robustness for light-tailed distributions. We consider generalised gamma location models with likelihoods $p_\theta(x) \propto \exp(-(x - \theta_1)^{\theta_2})$ where $\theta_1$ is a location parameter and $\theta_2$ determines how fast the tails decay. The larger $\theta_2$,

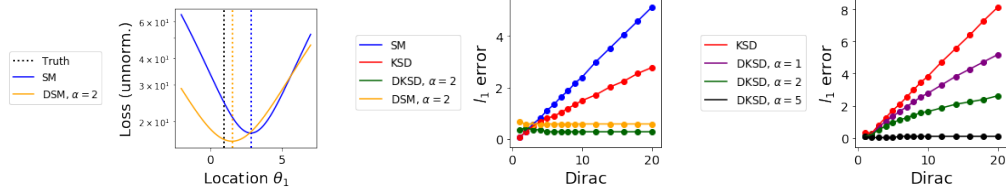

Figure 3: *Minimum SD Estimators for Generalised Gamma Distributions under Corruption*. We consider the case where $\theta_1^* = 0$ and $\theta_2^* = 2$ (left and middle) or $\theta_2^* = 5$ (right). Here $n = 300$.

the lighter the tails will be and vice-versa. We set $n = 300$ and corrupt 80 points by setting them to the value $x = 8$. A robust estimator should obtain a good approximation of $\theta^*$ even under this corruption. The left plot in Fig. 3 considers a Gaussian model (i.e. $\theta_2^* = 2$); we see that SM is not robust for this very simple model whereas DSM with $m(x) = 1/(1 + \|x\|^\alpha), \alpha = 2$ is robust. The middle plot shows that DKSD with this same $m$ is also robust, and confirms the analytical results of the previous section. Finally, the right plot considers the case $\theta_2^* = 5$ and we see that $\alpha$ can be chosen as a function of $\theta_2$ to guarantee robustness. In general, taking $\alpha \geq \theta_2^* - 1$ will guarantee a bounded influence function. Such a choice allows us to obtain robust estimators even for models with very light tails.

## 4.4 Efficient estimators for a simple unnormalised model

Finally we consider a simple intractable model from [47]: $p_\theta(x) \propto \exp(\eta(\theta)^\top \psi(x))$ where $\psi(x) = (\sum_{i=1}^d x_i^2, \sum_{i=3}^d x_1 x_i, \tanh(x))^\top$ and $\tanh$ is applied elementwise to $x$ and $\eta(\theta) = (-0.5, 0.2, 0.6, 0, 0, 0, \theta, 0)$. This model is intractable since we cannot easily compute its normalisation constant due to the difficulty of integrating the unnormalised part of the model. Our results based on $n = 200$ samples show that DKSD with $m(x) = \mathrm{diag}(1/(1 + x))$ is able to recover $\theta^* = -1$, whereas both SM and KSD provide less accurate estimates of the parameter. This illustrates yet again that a judicious choice of diffusion matrix can significantly improve the efficiency of our estimators.

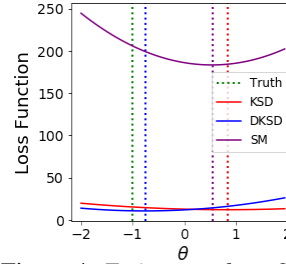

Figure 4: *Estimators for a Simple Intractable Model*

## 5 Conclusion

This paper introduced a general approach for constructing minimum distance estimators based on Stein's method, and demonstrated that many popular inference schemes can be recovered as special cases. This class of algorithms gives us additional flexibility through the choice of an operator and function space (the Stein operator and Stein class), which can be used to tailor the inference scheme to trade-off efficiency and robustness. However, this paper only scratches the surface of what is possible with minimum SD estimators. Looking ahead, it will be interesting to identify diffusion matrices which increase efficiency for important classes of problems in machine learning. One example on which we foresee progress are the product of student-t experts models [38, 66, 68], whose heavy tails render estimation challenging for SM. Advantages could also be found for other energy models, such as large graphical models where the kernel could be adapted to the graph [67].

### Acknowledgments

AB was supported by a Roth scholarship from the Department of Mathematics at Imperial College London. FXB was supported by the EPSRC grants [EP/L016710/1, EP/R018413/1]. AD and MG were supported by the Lloyds Register Foundation Programme on Data-Centric Engineering, the UKRI Strategic Priorities Fund under the EPSRC Grant [EP/T001569/1] and the Alan Turing Institute under the EPSRC grant [EP/N510129/1]. MG was supported by the EPSRC grants [EP/J016934/3, EP/K034154/1, EP/P020720/1, EP/R018413/1].

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
