[Supplementary Material]

# Supplementary Material

This document provides additional details for the paper "Minimum Stein Discrepancy Estimators". Appendix A contains background technical material required to understand the paper, Appendix B derives the minimum SD estimators from first principles and Appendix C derives the information metrics for DKSD and DSM. Appendix D contains proof of all asymptotic results including consistency and central limit theorems for DKSD and DSM, whilst Appendix E discusses their robustness.

Our derivations will use standard operators from vector calculus which we summarise in Appendix A.1. We will additionally introduce the following notation. We write $f \lesssim g$ if there is a constant $C > 0$ for which $f(x) \leq Cg(x)$ for all $x$. We set $\mathbb{Q}f \equiv \int f \mathrm{d}\mathbb{Q}$ and use $\Gamma(\mathcal{W}, \mathcal{Y})$ for the set of maps $\mathcal{W} \to \mathcal{Y}$ when $\mathcal{W} \neq \mathcal{X}$.

## A    Background Material

In this section, we provide background material which is necessary to follow the proofs in the following sections. This includes background in vector calculus, stochastic optimisation over manifolds and vector-valued reproducing kernel Hilbert spaces.

### A.1    Background on Vector Calculus

The following section contains background and important identities from vector calculus. For a function $g \in \Gamma(\mathcal{X}, \mathbb{R})$, $v \in \Gamma(\mathcal{X}, \mathbb{R}^d)$ and $A \in \Gamma(\mathcal{X}, \mathbb{R}^{d \times d})$ with components $A_{ij}, v_i, g$, we have $(\nabla g)_i = \partial_i g$, $(v \cdot A)_i = v_j A_{ji} = (v^\top A)_i$, $(\nabla \cdot A)_i = \partial_j A_{ji}$ which must be interpreted as the components of row-vectors; $(Av)_i = A_{ij}v_j$ which are the components of a column vector. Moreover $(\nabla v)_{ij} = \partial_j v_i$, $\nabla^2 f \equiv \nabla(\nabla f)$, $A : B \equiv \langle A, B \rangle = \mathrm{Tr}(A^\top B) = A_{ij}B_{ij}$. We have the following identities (where in the last equality we treat $\nabla \cdot A$ and $\nabla g$ as column vectors)

$$\nabla \cdot (gv) = \partial_i(gv_i) = v_i \partial_i g + g \partial_i v_i = (\nabla g)v + g\nabla \cdot v = \nabla g \cdot v + g\nabla \cdot v,$$
$$\nabla \cdot (gA) = \partial_i(gA_{ij})e_j = (A_{ij}\partial_i g + g\partial_i A_{ij})e_j = \nabla g \cdot A + g\nabla \cdot A = \nabla g^\top A + g\nabla \cdot A,$$
$$\nabla \cdot (Av) = \partial_i(A_{ij}v_j) = (\nabla \cdot A)v + \mathrm{Tr}[A\nabla v] = (\nabla \cdot A) \cdot v + \mathrm{Tr}[A\nabla v].$$

### A.2    Background on Norms

For $F \in \Gamma(\mathcal{X}, \mathbb{R}^{n_1 \times n_2})$ we set $\|F\|_p^p \equiv \int \|F(x)\|_p^p \mathrm{d}\mathbb{Q}(x)$, where $\|F(x)\|_p$ is the vector $p$-norm on $\mathbb{R}^{n_1 \times n_2}$ when $n_2 = 1$, else it is the induced operator norm. If $v \in \Gamma(\mathcal{X}, \mathbb{R}^{n_1})$, then $\|v\|_p^p = \int \|v(x)\|_p^p \mathrm{d}x = \int \sum_i |v_i(x)|^p \mathrm{d}x = \sum_i \|v_i\|_p^p$, hence $v \in L_p(\mathbb{Q})$ iff $v_i \in L_p(\mathbb{Q})$ for all $i$, and similarly $F \in L_p(\mathbb{Q})$ iff $F_{ij} \in L_p(\mathbb{Q})$ for all $i, j$ since the induced norm $\|F(x)\|_p$ and the vector norm $\|F\|_{vec}^p \equiv \sum_{ij} |F_{ij}(x)|^p$ are equivalent.

### A.3    Background on Vector-valued RKHS

A Hilbert space $\mathcal{H}$ of functions $\mathcal{X} \to \mathbb{R}^d$ is a RKHS if $\|f(x)\|_{\mathbb{R}^d} \leq C_x \|f\|_{\mathcal{H}}$. It follows that the evaluation "functional" $\delta_x : \mathcal{H} \to \mathbb{R}^d$ is continuous, for any $x$. Moreover for any $x \in \mathcal{X}, v \in \mathbb{R}^d$, the linear map $f \mapsto v \cdot f(x)$ is cts. By the Riesz representation theorem, there exists $K_x v \in \mathcal{H}$ s.t. $v \cdot f(x) = \langle K_x v, f \rangle$. From this we see that $K_x v$ is linear in $v$ (turns out linear combinations of $K_{x_i} v_i$ are dense in $\mathcal{H}$), and $K_x^* = \delta_x$. We define $K : \mathcal{X} \times \mathcal{X} \to \mathrm{End}(\mathbb{R}^d)$ by

$$K(x, y)v \equiv (K_y v)(x) = \delta_x \delta_y^* v.$$

It follows that $K(x, y) = K(y, x)^*$ and $u \cdot K(x, y)v = \langle K_y v, K_x u \rangle$. Denote by $e_i$ the $i$th vector in the standard basis of $\mathbb{R}^d$. From this we can get the components of the matrix:

$$(K(x, y))_{ij} = \langle K_x e_i, K_y e_j \rangle.$$

We have for any $v_i, x_j$, $\sum_{j,k} v_j \cdot K(x_j, x_k)v_k \geq 0$.

## A.4 Background on Separable Kernels

Consider the $d$ dimensional product space $\mathcal{H}^d$ of function $f : \mathcal{X} \to \mathbb{R}^d$ with components $f_i \in \mathcal{H}_i$ and $\mathcal{H}_i$ is a RKHS with kernel $C^2$ kernel $k^i : \mathcal{X} \times \mathcal{X} \to \mathbb{R}$. Let $K : \mathcal{X} \times \mathcal{X} \to \text{End}(\mathbb{R}^d) \cong \mathbb{R}^{d \times d}$ be the kernel of $\mathcal{H}^d$ (see Appendix A.3). Note if $K_x \equiv K(x, \cdot) : \mathcal{X} \to \text{End}(\mathbb{R}^d)$, and if $v \in \mathbb{R}^d$, then $K_x v \in \mathcal{H}^d$. The reproducing property then states that $\forall f \in \mathcal{H}^d$: $\langle f(x), v \rangle_{\mathbb{R}^d} = \langle f, K(\cdot, x)v \rangle_{\mathcal{H}^d}$. Moreover for the kernel $K = \text{diag}(\lambda_1 k^1, \dots, \lambda_d k^d)$ we will prove below that $\langle f, g \rangle_{\mathcal{H}^d} = \frac{1}{\lambda_i} \sum_i \langle f_i, g_i \rangle_{\mathcal{H}_i}$, whereas for $K = Bk$ where $B$ is symmetric and invertible we should have $\langle f, g \rangle_{\mathcal{H}^d} = \sum_{ij} B_{ij}^{-1} \langle f_i, g_j \rangle_{\mathcal{H}}$.

Given a real-valued kernel $k_i$ on $\mathcal{X}$, consider $K = \text{diag}(\lambda_1 k_1, \dots, \lambda_n k_n)$. Let $f = \sum_j \delta_{x_j}^* v_j$. Recall this is a dense subset of $\mathcal{H}^d$: we will derive the RKHS norm for this dense subset and by continuity this will hold for any function. Given the norm, the formula for the inner product will follow by the polarization identity. We have

$$f_i(x) = \delta_x(f) \cdot e_i = \delta_x \delta_{x_j}^* v_j \cdot e_i = K(x, x_j) v_j \cdot e_i$$
$$= \text{diag}(\lambda_1 k_1, \dots, \lambda_n k_n)(x, x_j) v_j \cdot e_i = \lambda_i k_i(x, x_j) v_j^i$$

$$\|f\|_{\mathcal{H}_K}^2 = \langle \delta_{x_j}^* v_j, \delta_{x_l}^* v_l \rangle_{\mathcal{H}_K} = v_j \cdot K(x_j, x_l) v_l = v_j^i \lambda_i k_i(x_j, x_l) v_l^i$$

On the other hand, $\sum_i \frac{1}{\lambda_i} \langle f_i, f_i \rangle_{k_i} = \sum_i \frac{1}{\lambda_i} \lambda_i^2 v_j^i v_l^i k_i(x_j, x_l)$. Thus $\|f\|_{\mathcal{H}_K}^2 = \frac{1}{\lambda_i} \sum_i \langle f_i, f_i \rangle_{k_i}$.

For a symmetric positive definite matrix $B$, consider the kernel on $\mathcal{H}$ $K(x, y) \equiv k(x, y)B$. Let $f = \sum_j \delta_{x_j}^* v_j$. We have:

$$f_i(x) = \delta_x(f) \cdot e_i = \delta_x \delta_{x_j}^* v_j \cdot e_i = K(x, x_j) v_j \cdot e_i = Bv_j \cdot e_i k_{x_j}(x)$$

This implies $f_i \in \mathcal{H}_k$. Then

$$\|f\|_{\mathcal{H}_K}^2 = \langle \delta_{x_j}^* v_j, \delta_{x_l}^* v_l \rangle_{\mathcal{H}_K} = v_j \cdot K(x_j, x_l) v_l = k(x_j, x_l) v_j \cdot Bv_l.$$

On the other hand $\langle f_i, f_j \rangle_k = e_i^\top Bv_r e_j^\top Bv_s k(x_s, x_r)$. Notice

$$B_{ij}^{-1} e_i^\top Bv_r = B_{ij}^{-1} B_{il} v_r^l = \delta_{lj} v_r^l = v_r^j.$$

So we have:

$$B_{ij}^{-1} \langle f_i, f_j \rangle_k = v_r^j e_j^\top Bv_s k(x_s, x_r) = v_r^j B_{ja} v_s^a k(x_s, x_r) = v_r \cdot Bv_s k(x_s, x_r)$$

## A.5 Background on Stochastic Optimisation on Riemmannian Manifolds

The gradient flow of a curve $\theta$ on a complete connected Riemannian manifold $\Theta$ (for example a Hilbert space) is the solution to $\dot{\theta}(t) = -\nabla_{\theta(t)} \text{SD}(\mathbb{Q}\|\mathbb{P}_\theta)$, where $\nabla_\theta$ is the Riemannian gradient at $\theta$. Typically [1] the gradient flow is approximated by the update equation $\theta(t+1) = \exp_{\theta(t)}(-\gamma_t H(Z_t, \theta))$ where $\exp$ is the Riemannian exponential map, $(\gamma_t)$ is a sequence of step sizes with $\sum \gamma_t^2 < \infty$, $\sum \gamma_t = +\infty$, and $H$ is an unbiased estimator of the loss gradient, $\mathbb{E}[H(Z_t, \theta)] = \nabla_\theta \text{SD}(\mathbb{Q}\|\mathbb{P}_\theta)$. When the Riemannian exponential is computationally expensive, it is convenient to replace it by a retraction $\mathcal{R}$, that is a first-order approximation which stays on the manifold. This leads to the update $\theta(t+1) = \mathcal{R}_{\theta(t)}(-\gamma_t H(Z_t, \theta))$ [7]. When $\Theta$ is a linear manifold it is common to take $\mathcal{R}_{\theta(t)}(-\gamma_t H(Z_t, \theta)) \equiv \theta(t) - \gamma_t H(Z_t, \theta)$. In local coordinates $(\theta^i)$ we have $\nabla_\theta \text{SD}(\mathbb{Q}\|\mathbb{P}_\theta) = g(\theta)^{-1} \mathrm{d}_\theta \text{SD}(\mathbb{Q}\|\mathbb{P}_\theta)$, where $\mathrm{d}_\theta f$ denotes the tuple $(\partial_{\theta^i} f)$, which we will approximate using the biased estimator $H(\{X_i^t\}_i, \theta) \equiv \hat{g}_{\theta(t)}(\{X_i^t\}_{i=1}^n)^{-1} \mathrm{d}_\theta \widehat{\text{SD}}(\{X_i^t\}_{i=1}^n \| \mathbb{P}_\theta)$, where $\hat{g}_{\theta(t)}(\{X_i^t\}_{i=1}^n)$ is an unbiased estimator for the information matrix $g(\theta(t))$ using a sample $\{X_i^t\}_{i=1}^n \sim \mathbb{Q}$. We thus obtain the following Riemannian gradient descent algorithm

$$\theta(t+1) = \theta(t) - \gamma_t \hat{g}_{\theta(t)}(\{X_i^t\}_{i=1}^n)^{-1} \mathrm{d}_{\theta(t)} \widehat{\text{SD}}(\{X_i^t\}_{i=1}^n \| \mathbb{P}_\theta).$$

When $\Theta = \mathbb{R}^m$, $\gamma_t = \frac{1}{t}$, $g$ is the Fisher metric and $\widehat{\text{SD}}(\{X_i^t\}_{i=1}^n \| \mathbb{P}_\theta)$ is replaced by $\widehat{\text{KL}}(\{X_i^t\}_{i=1}^n \| \mathbb{P}_\theta)$ this recovers the natural gradient descent algorithm [1].

# B  Derivation of Diffusion Stein Discrepancies

In this appendix, we carefully derive the diffusion SD studied in this paper. We begin by providing details on the diffusion Stein operator, then move on to the DKSD and DSM divergences and corresponding estimators.

For any matrix kernel we will show in Appendix B.1 that $\forall f \in \mathcal{H}^d$: $\mathcal{S}_p^m[f](x) = \langle \mathcal{S}_p^{m,1} K_x, f \rangle_{\mathcal{H}^d}$. In Appendix B.2 we prove that if $x \mapsto \|\mathcal{S}_p^{m,1} K_x\|_{\mathcal{H}^d} \in L^1(\mathbb{Q})$, then

$$\mathrm{DKSD}_{K,m}(\mathbb{Q}\|\mathbb{P})^2 \equiv \sup_{\substack{h \in \mathcal{H}^d \\ \|h\| \leq 1}} \left| \int_\mathcal{X} \mathcal{S}_p^m[h] \mathrm{d}\mathbb{Q} \right|^2 = \int_\mathcal{X} \int_\mathcal{X} \mathcal{S}_p^{m,2} \mathcal{S}_p^{m,1} K(x,y) \mathrm{d}\mathbb{Q}(x) \mathrm{d}\mathbb{Q}(y).$$

In Appendix B.3 we further show the Stein kernel satisfies

$$k^0(x,y) \equiv \mathcal{S}_p^{m,2} \mathcal{S}_p^{m,1} K(x,y) = \tfrac{1}{p(y)p(x)} \nabla_y \cdot \nabla_x \cdot \left( p(x) m(x) K(x,y) m(y)^\top p(y) \right).$$

## B.1  Stein Operator

By definition for $f \in \Gamma(\mathcal{X}, \mathbb{R}^d)$ and $A \in \Gamma(\mathcal{X}, \mathbb{R}^{d \times d})$

$$\mathcal{S}_p[f] = \tfrac{1}{p} \nabla \cdot (pmf) = m^\top \nabla \log p \cdot f + \nabla \cdot (mf),$$
$$\mathcal{S}_p[A] = \tfrac{1}{p} \nabla \cdot (pmA) = m^\top \nabla \log p \cdot A + \nabla \cdot (mA)$$

which are operators $\Gamma(\mathcal{X}, \mathbb{R}^d) \to \Gamma(\mathcal{X}, \mathbb{R})$ and $\Gamma(\mathcal{X}, \mathbb{R}^{d \times d}) \to \Gamma(\mathcal{X}, \mathbb{R}^d)$ respectively.

**Proposition 8.** *Let $\mathcal{X}$ be an open (connected) subset of $\mathbb{R}^d$, $m$ is continuously differentiable, and $K : \mathcal{X} \times \mathcal{X} \to \mathbb{R}^{d \times d}$ is the matrix kernel of $\mathcal{H}^d$. Suppose for any $j \in [1, d]$, $K, \partial_{1^j} \partial_{2^j} K$ are separately continuous and locally bounded. Then for any $f \in \mathcal{H}^d$*

$$\mathcal{S}_p[f](x) = \langle \mathcal{S}_p^1[K]|_x, f \rangle_{\mathcal{H}^d}$$

**Proof**

Note that technically the kernel $K$ of $\mathcal{H}^d$ takes value in the set of (bounded) linear operators on $\mathbb{R}^d$, and we view these linear operators as matrices by defining the components $(K(x,y))_{ji} \equiv e_j \cdot K(x,y) e_i$, where $(e_l)$ is the canonical basis of $\mathbb{R}^d$. For any $f \in \mathcal{H}^d$

$$
\begin{aligned}
\langle f(x), m(x)^\top \nabla \log p(x) \rangle_{\mathbb{R}^d} &= \langle f, K(\cdot, x) m(x)^\top \nabla \log p(x) \rangle_{\mathcal{H}^d} \\
&= \langle f, K_x^\top m(x)^\top \nabla \log p(x) \rangle_{\mathcal{H}^d} \\
&= \langle f, m(x)^\top \nabla \log p(x) \cdot K_x \rangle_{\mathcal{H}^d}.
\end{aligned}
$$

Moreover, under these assumptions the RKHS $\mathcal{H}^d$ is continuously embedded in the topological space $C^1(\mathcal{X}, \mathbb{R}^d)$, so its elements are continuously differentiable. Then for any $f \in \mathcal{H}^d$, by theorem 2.11 [53]

$$\langle f, \partial_{2^j} K(\cdot, x) e_r \rangle_{\mathcal{H}^d} = \langle e_r, \partial_j f|_x \rangle_{\mathbb{R}^d} = \partial_j f_r|_x.$$

Hence

$$
\begin{aligned}
\langle f, \nabla \cdot (mK)|_x \rangle_{\mathcal{H}^d} &= \langle f, \partial_{1^j}(m_{jr} K_{ri})|_x e_i \rangle_{\mathcal{H}^d} = \langle f, \partial_j m_{jr}|_x K_{ri}(x, \cdot) e_i + m_{jr}(x) \partial_{1^j} K_{ri}|_x e_i \rangle_{\mathcal{H}^d} \\
&= \partial_j m_{jr}|_x \langle f, K_{ir}(\cdot, x) e_i \rangle_{\mathcal{H}^d} + m_{jr}(x) \langle f, \partial_{1^j} K_{ri}(x, \cdot) e_i \rangle_{\mathcal{H}^d} \\
&= \partial_j m_{jr}|_x \langle f, K(\cdot, x) e_r \rangle_{\mathcal{H}^d} + m_{jr}(x) \langle f, \partial_{2^j} K_{ir}(\cdot, x) e_i \rangle_{\mathcal{H}^d} \\
&= \partial_j m_{jr}|_x \langle f, K(\cdot, x) e_r \rangle_{\mathcal{H}^d} + m_{jr}(x) \langle f, \partial_{2^j} K(\cdot, x) e_r \rangle_{\mathcal{H}^d} \\
&= \partial_j m_{jr}|_x f_r(x) + m_{jr}(x) \partial_j f_r|_x \\
&= \langle \nabla \cdot m, f(x) \rangle_{\mathbb{R}^d} + \mathrm{Tr}[m(x) \nabla_x f] \\
&= \nabla_x \cdot (mf).
\end{aligned}
$$

Therefore, we conclude that $\mathcal{S}_p[f](x) = \langle \mathcal{S}_p^1 K_x, f \rangle_{\mathcal{H}^d}$ where $\mathcal{S}_p^1 K_x \equiv \mathcal{S}_p^1[K]|_x$ means applying $\mathcal{S}_p$ to the first entry of $K$ and evaluate it $x$, so informally $\mathcal{S}_p^1[K]|_x : y \mapsto \tfrac{1}{p} \nabla_x \cdot (p(x) m(x) K(x,y))$.  ∎

## B.2 Diffusion Kernel Stein Discrepancies

**Proposition 9.** *Suppose $\mathcal{S}_p[f](x) = \langle \mathcal{S}_p^1[K]|_x, f \rangle_{\mathcal{H}^d}$ for any $f \in \mathcal{H}^d$. Let $m$ and $K$ be $C^2$, and $x \mapsto \mathcal{S}_p K_x$ be $\mathbb{Q}$-Bochner integrable. Then*

$$\mathrm{DKSD}_{K,m}(\mathbb{Q}, \mathbb{P})^2 = \int_{\mathcal{X}} \int_{\mathcal{X}} \mathcal{S}_p^2 \mathcal{S}_p^1 K(x,y) \mathrm{d}\mathbb{Q}(x) \mathrm{d}\mathbb{Q}(y).$$

**Proof**

Let us identify $\mathcal{H}_1 \otimes \mathcal{H}_2 \cong L(\mathcal{H}_1 \times \mathcal{H}_2, \mathbb{R}) \cong L(\mathcal{H}_2, \mathcal{H}_1)$ with $(v_1 \otimes v_2) \sim v_1 \langle v_2, \cdot \rangle_{\mathcal{H}_2}$ (since $\mathcal{H}_2 \cong \mathcal{H}_2^*$), so that $(v_1 \otimes v_2) u_2 \equiv v_1 \langle v_2, u_2 \rangle_{\mathcal{H}_2}$ (here $L(V,W)$ is the space of linear maps from $V$ to $W$). Then

$$\langle u_1 \otimes u_2, v_1 \otimes v_2 \rangle_{HS} \equiv \langle u_1, v_1 \rangle_{\mathcal{H}_1} \langle u_2, v_2 \rangle_{\mathcal{H}_2} = \langle u_1, (v_1 \otimes u_2) v_2 \rangle_{\mathcal{H}_1}.$$

For simplicity we will write $\mathcal{S}_p K_x \equiv \mathcal{S}_p^1[K]|_x$. Using the fact $x \mapsto \mathcal{S}_p K_x$ is $\mathbb{Q}$-Bochner integrable, then by Cauchy-Schwartz $x \mapsto \langle h, \mathcal{S}_p K_x \rangle_{\mathcal{H}^d}$ is $\mathbb{Q}$-integrable. Then

$$\begin{aligned}
\mathrm{DKSD}_{K,m}(\mathbb{Q}, \mathbb{P})^2 &= \sup_{\substack{h \in \mathcal{H}^d \\ \|h\| \leq 1}} \left\langle \int_{\mathcal{X}} \mathcal{S}_p[h](x) \mathrm{d}\mathbb{Q}(x), \int_{\mathcal{X}} \mathcal{S}_p[h](y) \mathrm{d}\mathbb{Q}(y) \right\rangle_{\mathbb{R}} \\
&= \sup_{\substack{h \in \mathcal{H}^d \\ \|h\| \leq 1}} \int_{\mathcal{X}} \langle h, \mathcal{S}_p K_x \rangle_{\mathcal{H}^d} \mathrm{d}\mathbb{Q}(x) \int_{\mathcal{X}} \langle h, \mathcal{S}_p K_y \rangle_{\mathcal{H}^d} \mathrm{d}\mathbb{Q}(y) \\
&= \sup_{\substack{h \in \mathcal{H}^d \\ \|h\| \leq 1}} \int_{\mathcal{X}} \int_{\mathcal{X}} \langle h, \mathcal{S}_p K_x \rangle_{\mathcal{H}^d} \langle h, \mathcal{S}_p K_y \rangle_{\mathcal{H}^d} \mathrm{d}\mathbb{Q}(x) \mathrm{d}\mathbb{Q}(y) \\
&= \sup_{\substack{h \in \mathcal{H}^d \\ \|h\| \leq 1}} \int_{\mathcal{X}} \int_{\mathcal{X}} \langle h, \mathcal{S}_p K_x \otimes \mathcal{S}_p K_y h \rangle_{\mathcal{H}^d} \mathrm{d}\mathbb{Q}(x) \mathrm{d}\mathbb{Q}(y) \\
&= \sup_{\substack{h \in \mathcal{H}^d \\ \|h\| \leq 1}} \int_{\mathcal{X}} \int_{\mathcal{X}} \langle h \otimes h, \mathcal{S}_p K_x \otimes \mathcal{S}_p K_y \rangle_{HS} \mathrm{d}\mathbb{Q}(x) \mathrm{d}\mathbb{Q}(y)
\end{aligned}$$

Moreover $\int_{\mathcal{X}} \|\mathcal{S}_p K_x \otimes \mathcal{S}_p K_y\|_{HS} \mathrm{d}\mathbb{Q}(x) \mathrm{d}\mathbb{Q}(y) < \infty$, since

$$\begin{aligned}
& \int_{\mathcal{X}} \|\mathcal{S}_p K_x \otimes \mathcal{S}_p K_y\|_{HS} \mathrm{d}\mathbb{Q}(x) \otimes \mathrm{d}\mathbb{Q}(y) \\
&= \int_{\mathcal{X}} \int_{\mathcal{X}} \sqrt{\langle \mathcal{S}_p K_x, \mathcal{S}_p K_x \rangle_{\mathcal{H}^d} \langle \mathcal{S}_p K_y, \mathcal{S}_p K_y \rangle_{\mathcal{H}^d}} \mathrm{d}\mathbb{Q}(x) \mathrm{d}\mathbb{Q}(y) \\
&= \left( \int_{\mathcal{X}} \sqrt{\langle \mathcal{S}_p K_x, \mathcal{S}_p K_x \rangle_{\mathcal{H}^d}} \mathrm{d}\mathbb{Q}(x) \right)^2 \\
&= \left( \int_{\mathcal{X}} \|\mathcal{S}_p K_x\|_{\mathcal{H}^d} \mathrm{d}\mathbb{Q}(x) \right)^2 < \infty
\end{aligned}$$

since by assumption $x \mapsto \mathcal{S}_p K_x$ is $\mathbb{Q}$-Bochner integrable. Thus

$$\begin{aligned}
\mathrm{DKSD}_{K,m}(\mathbb{Q}, \mathbb{P})^2 &= \sup_{\substack{h \in \mathcal{H}^d \\ \|h\| \leq 1}} \left\langle h \otimes h, \int_{\mathcal{X}} \int_{\mathcal{X}} \mathcal{S}_p K_x \otimes \mathcal{S}_p K_y \mathrm{d}\mathbb{Q}(x) \mathrm{d}\mathbb{Q}(y) \right\rangle_{HS} \\
&= \left\| \int_{\mathcal{X}} \int_{\mathcal{X}} \mathcal{S}_p K_x \otimes \mathcal{S}_p K_y \mathrm{d}\mathbb{Q}(x) \mathrm{d}\mathbb{Q}(y) \right\|_{HS} \\
&= \left\| \int_{\mathcal{X}} \mathcal{S}_p K_x \mathrm{d}\mathbb{Q}(x) \otimes \int_{\mathcal{X}} \mathcal{S}_p K_y \mathrm{d}\mathbb{Q}(y) \right\|_{HS} \\
&= \left\| \int_{\mathcal{X}} \mathcal{S}_p K_x \mathrm{d}\mathbb{Q}(x) \right\|_{\mathcal{H}^d}^2 \\
&= \left\langle \int_{\mathcal{X}} \mathcal{S}_p K_x \mathrm{d}\mathbb{Q}(x), \int_{\mathcal{X}} \mathcal{S}_p K_y \mathbb{Q}(\mathrm{d}y) \right\rangle_{\mathcal{H}^d} \\
&= \int_{\mathcal{X}} \int_{\mathcal{X}} \langle \mathcal{S}_p K_x, \mathcal{S}_p K_y \rangle_{\mathcal{H}^d} \mathrm{d}\mathbb{Q}(x) \mathrm{d}\mathbb{Q}(y) \\
&= \int_{\mathcal{X}} \int_{\mathcal{X}} \mathcal{S}_p^2 \mathcal{S}_p^1 K(x,y) \mathrm{d}\mathbb{Q}(x) \mathrm{d}\mathbb{Q}(y).
\end{aligned}$$

To show the penultimate equality (exchange integral and inner product), we use the fact $\mathcal{S}_p K_x$ is $\mathbb{Q}$-Bochner integrable, and that the operator $W : f \mapsto \langle f, \int_{\mathcal{X}} \mathcal{S}_p K_y \mathbb{Q}(\mathrm{d}y) \rangle_{\mathcal{H}^d}$ is bounded, from which it follows that

$$\begin{aligned}
\left\langle \int_{\mathcal{X}} \mathcal{S}_p K_x \mathrm{d}\mathbb{Q}(x), \int_{\mathcal{X}} \mathcal{S}_p K_y \mathbb{Q}(\mathrm{d}y) \right\rangle_{\mathcal{H}^d} &= W \left[ \int_{\mathcal{X}} \mathcal{S}_p K_x \mathrm{d}\mathbb{Q}(x) \right] = \int_{\mathcal{X}} W[\mathcal{S}_p K_x \mathrm{d}\mathbb{Q}(x)] \\
&= \int_{\mathcal{X}} \left\langle \mathcal{S}_p K_x, \int_{\mathcal{X}} \mathcal{S}_p K_y \mathrm{d}\mathbb{Q}(y) \right\rangle_{\mathcal{H}^d} \mathrm{d}\mathbb{Q}(x) \\
&= \int_{\mathcal{X}} \int_{\mathcal{X}} \langle \mathcal{S}_p K_x, \mathcal{S}_p K_y \rangle_{\mathcal{H}^d} \mathrm{d}\mathbb{Q}(x) \mathrm{d}\mathbb{Q}(y)
\end{aligned}$$

Hence $\mathrm{DKSD}_{K,m}(\mathbb{Q}, \mathbb{P})^2 = \int_{\mathcal{X}} \int_{\mathcal{X}} \mathcal{S}_p^2 \mathcal{S}_p^1 K(x,y) \mathrm{d}\mathbb{Q}(x) \mathrm{d}\mathbb{Q}(y)$.

Note that from this proof we have

$$k^0(x,y) \equiv \mathcal{S}_p^2 \mathcal{S}_p^1 K(x,y) = \langle \mathcal{S}_p K_x, \mathcal{S}_p K_y \rangle_{\mathcal{H}^d},$$

which shows the map $\phi : \mathcal{X} \to \mathcal{H}^d$, $\phi(x) \equiv \mathcal{S}_p^1[K]|_x$ is a feature map (more precisely it is dual to the feature map) for the scalar reproducing kernel $k^0$, and its RKHS consists of functions $g(\cdot) = \langle \phi(\cdot), f \rangle_{\mathcal{H}^d}$ for $f \in \mathcal{H}^d$ [52]. ∎

## B.3 The Stein Kernel Corresponding to the Diffusion Kernel Stein Discrepancy

Note the Stein kernel satisfies

$$k^0 = \tfrac{1}{p(y)p(x)} \nabla_y \cdot \nabla_x \cdot \left( p(x)m(x)Km(y)^\top p(y) \right)$$

since

$$
\begin{aligned}
k^0 = \mathcal{S}_p^2 \mathcal{S}_p^1 K(x,y) &= \tfrac{1}{p(y)p(x)} \nabla_y \cdot (p(y)m(y)\nabla_x \cdot (p(x)m(x)K)) \\
&= \tfrac{1}{p(y)p(x)} \nabla_y \cdot (p(y)m(y)\partial_{x^i}(p(x)m(x)_{ir}K_{rs})e_s) \\
&= \tfrac{1}{p(y)p(x)} \nabla_y \cdot (p(y)m(y)_{ls}\partial_{x^i}(p(x)m(x)_{ir}K_{rs})e_l) \\
&= \tfrac{1}{p(y)p(x)} \partial_{y^l}(p(y)m(y)_{ls}\partial_{x^i}(p(x)m(x)_{ir}K_{rs})) \\
&= \tfrac{1}{p(y)p(x)} \partial_{y^l}\partial_{x^i}\left(p(x)m(x)_{ir}K_{rs}m(y)_{sl}^\top p(y)\right) \\
&= \tfrac{1}{p(y)p(x)} \nabla_y \cdot \nabla_x \cdot \left(p(x)m(x)Km(y)^\top p(y)\right).
\end{aligned}
$$

Note it is also possible to view $m(x)Km(y)^\top$ as a new matrix kernel. That is the matrix field $m$ defines a new kernel $K_m : (x,y) \mapsto m(x)K(x,y)m^\top(y)$, since $K_m(y,x)^\top = m(x)K(y,x)m(y)^\top = K_m(x,y)$ and for any $v_j \in \mathbb{R}^d, x_i \in \mathcal{X}$,

$$v_j \cdot K_m(x_j,x_l)v_l = v_j \cdot m(x_j)K(x_j,x_l)m(x_l)^\top v_l = \left(m(x_j)^\top v_j\right) \cdot K(x_j,x_l)\left(m(x_l)^\top v_l\right) \geq 0$$

We can expand the Stein kernel using the following expressions:

$$
\begin{aligned}
&\nabla_y \cdot (p(y)m(y)\nabla_x \cdot (p(x)m(x)K)) \\
&= \nabla_y \cdot \left(p(y)m(y)\left(Km(x)^\top \nabla_x p + p(x)\nabla_x \cdot (m(x)K)\right)\right).
\end{aligned}
$$

$$
\begin{aligned}
&\nabla_y \cdot \left(p(y)m(y)Km(x)^\top \nabla_x p\right) \\
&= m^\top(x)\nabla_x p \cdot Km(y)^\top \nabla_y p + p(y)\nabla_y \cdot \left(m(y)Km(x)^\top \nabla_x p\right) \\
&= m^\top(x)\nabla_x p \cdot Km(y)^\top \nabla_y p + p(y)\nabla_y \cdot (m(y)K) \cdot m(x)^\top \nabla_x p,
\end{aligned}
$$

$$
\begin{aligned}
&\nabla_y \cdot (p(y)m(y)p(x)\nabla_x \cdot (m(x)K)) \\
&= p(x)(\nabla_y \cdot (p(y)m(y)) \cdot \nabla_x \cdot (m(x)K) + p(y)\mathrm{Tr}[m(y)\nabla_y\nabla_x \cdot (m(x)K)]) \\
&= p(x)p(y)\mathrm{Tr}[m(y)\nabla_y\nabla_x \cdot (m(x)K)] \\
&\quad + p(x)\nabla_x \cdot (m(x)K) \cdot \left(m(y)^\top \nabla_y p + p(y)\nabla_y \cdot m\right).
\end{aligned}
$$

Hence

$$
\begin{aligned}
k^0 &= m^\top(x)\nabla_x \log p \cdot Km(y)^\top \nabla_y \log p \\
&\quad + \nabla_y \cdot (m(y)K) \cdot m(x)^\top \nabla_x \log p + \nabla_x \cdot (m(x)K) \cdot m(y)^\top \nabla_y \log p \\
&\quad + \nabla_x \cdot (m(x)K) \cdot \nabla_y \cdot m + \mathrm{Tr}[m(y)\nabla_y\nabla_x \cdot (m(x)K)] \\
&= \langle s_p(x), Ks_p(y) \rangle + \langle \nabla_y \cdot (m(y)K), s_p(x) \rangle + \langle \nabla_x \cdot (m(x)K), s_p(y) \rangle \\
&\quad + \langle \nabla_x \cdot (m(x)K), \nabla_y \cdot m \rangle + \mathrm{Tr}[m(y)\nabla_y\nabla_x \cdot (m(x)K)]
\end{aligned}
$$

## B.4 Special Cases of Diffusion Kernel Stein Discrepancy

Consider

$$k^0 = \frac{1}{p(y)p(x)} \nabla_y \cdot \nabla_x \cdot \left(p(x)m(x)K(x,y)m(y)^\top p(y)\right)$$

and decompose $m(x)K(x,y)m(y)^\top \equiv gA$ where $g$ is scalar and $A$ is matrix-valued. Then we

$$k^0 = g\langle \nabla_y \log p, A\nabla_x \log p\rangle + \langle \nabla_y \log p, A\nabla_x g\rangle + \langle \nabla_y g, A\nabla_x \log p\rangle$$
$$+ \mathrm{Tr}[A\nabla_x \nabla_y g] + g\nabla_y \cdot \nabla_x \cdot A + \langle \nabla_x \cdot A, \nabla_y g\rangle + \langle \nabla_y \cdot A^\top, \nabla_x g\rangle$$
$$+ g\langle \nabla_y \cdot A^\top, \nabla_x \log p\rangle + g\langle \nabla_x \cdot A, \nabla_y \log p\rangle.$$

For the case, $K = \mathrm{diag}(k^1, \ldots, k^d)$, setting $\mathcal{T}_i^x \equiv \frac{1}{p(x)}\partial_{x^i}(p(x)\cdot)$ then

$$\mathcal{S}_p^2\mathcal{S}_p^1[\mathrm{diag}(k^1, \ldots, k^d)] = \mathcal{T}_l^y\left(m_{li}(y)\mathcal{T}_c^x\left(k^i(x,y)m_{ic}^\top(x)\right)\right) = \mathcal{T}_l^y\mathcal{T}_c^x\left(m_{li}(y)k^i(x,y)m_{ci}(x)\right).$$

If $K = Ik$ in components

$$\mathcal{S}_p^2\mathcal{S}_p^1[Ik] = (s_p(x))_i k(x,y)(s_p(y))_i + \partial_{y^i}(m_{ir}k)(s_p(x))_r + \partial_{x^i}(m(x)_{ir}k)(s_p(y))_r$$
$$+ \partial_{x^i}(m(x)_{ir}k)\partial_{y^l}(m_{lr}) + m(y)_{ir}\partial_{y^i}\partial_{x^s}(m(x)_{sr}k)$$

When $p = p_\theta$ we are often interested in the gradient $\nabla_\theta k_\theta^0$. Note $\nabla_y \cdot (m(y)K) = k\nabla_y \cdot m + \nabla_y k \cdot m(y)$, so [2]

$$\partial_{\theta^i}[k\langle \nabla_y \cdot m, s_p(x)\rangle] = k\partial_{\theta^i}\langle \nabla_y \cdot m, s_p(x)\rangle$$
$$\partial_{\theta^i}[\langle \nabla_y k \cdot m(y), s_p(x)\rangle] = \langle \nabla_y k, \partial_{\theta^i}[m(y)s_p(x)]\rangle$$
$$\mathrm{Tr}[m(y)\nabla_y\nabla_x \cdot (m(x)K)] = \nabla_y k^\top m(y)\nabla_x \cdot m + \mathrm{Tr}[m(y)m(x)^\top\nabla_y\nabla_x k]$$

and the terms in $\partial_{\theta^i}k^0$ reduce to

$$\partial_{\theta^i}\langle s_p(x), Ks_p(y)\rangle = k\partial_{\theta^i}\langle s_p(x), s_p(y)\rangle$$
$$\partial_{\theta^i}\langle \nabla_y \cdot (m(y)K), s_p(x)\rangle = k\partial_{\theta^i}\langle \nabla_y \cdot m, s_p(x)\rangle + \langle \nabla_y k, \partial_{\theta^i}[m(y)s_p(x)]\rangle$$
$$\partial_{\theta^i}\langle \nabla_x \cdot (m(x)K), s_p(y)\rangle = k\partial_{\theta^i}\langle \nabla_x \cdot m, s_p(y)\rangle + \langle \nabla_x k, \partial_{\theta^i}[m(x)s_p(y)]\rangle$$
$$\partial_{\theta^i}\langle \nabla_x \cdot (m(x)K), \nabla_y \cdot m\rangle = k\partial_{\theta^i}\langle \nabla_x \cdot m, \nabla_y \cdot m\rangle + \partial_{\theta^i}\langle \nabla_x k \cdot m(x), \nabla_y \cdot m\rangle.$$

When $K = kI$ and we further have a diagonal matrix $m = \mathrm{diag}(f_i)$, $m(y)m(x)^\top = \mathrm{diag}(f_i(y)f_i(x))$. If $u \odot v$ denotes the vector given by the pointwise product of vectors, i.e., $(u \odot v)_i = u_i v_i$, and $f$ is the vector, then $m(x)\nabla_x \log p = f(x) \odot \nabla_x \log p$ and $(\nabla_y \cdot m)_i = \partial_{y^i} f_i$, $(\nabla_x \cdot (mk))_i = \partial_{x^i}(f_i k)$,

$$s_p(x) \cdot Ks_p(y) = k(x,y)f_i(x)\partial_{x^i}\log p f_i(y)\partial_{y^i}\log p$$
$$\nabla_y \cdot (m(y)K) \cdot s_p(x) = \partial_{y^i}(f_i(y)k)f_i(x)\partial_{x^i}\log p$$
$$\nabla_x \cdot (m(x)K) \cdot \nabla_y \cdot m = \partial_{x^i}(f_i(x)k)\partial_{y^i}(f_i(y))$$
$$\mathrm{Tr}[m(y)\nabla_y\nabla_x \cdot (mk)] = f_i(y)\partial_{x^i}\left(f_i(x)\partial_{y^i}k\right)$$

and if $m \mapsto mI$ (is scalar), (this is just KSD with $k(x,y) \mapsto m(x)k(x,y)m(y)$):

$$k^0 = m(x)m(y)k(x,y)\nabla_x \log p \cdot \nabla_y \log p$$
$$+ m(x)\nabla_y(m(y)k) \cdot \nabla_x \log p + m(y)\nabla_x(m(x)k) \cdot \nabla_y \log p$$
$$+ \nabla_x(m(x)k) \cdot \nabla_y m + m(y)\nabla_x \cdot (m(x)\nabla_y k),$$

When $m = I$, we recover the usual definition of kernel-Stein discrepancy (KSD):

$$\mathrm{KSD}(\mathbb{Q}\|\mathbb{P})^2 = \int_{\mathcal{X}}\int_{\mathcal{X}}\frac{1}{p(y)p(x)}\nabla_y \cdot \nabla_x(p(x)k(x,y)p(y))\mathrm{d}\mathbb{Q}(x)\mathrm{d}\mathbb{Q}(y).$$

$$\partial_{\theta^i}[(\nabla_y \cdot m) \cdot Ks_p(x)] = kB_{sr}\partial_{\theta^i}((\nabla_y \cdot m)_s(s_p(x))_r) = k\,\mathrm{Tr}[B\partial_{\theta^i}(s_p(x) \otimes \nabla_y \cdot m)]$$
$$\partial_{\theta^i}\left[\nabla_y k^\top m(y)Bs_p(x)\right] = \partial_{y^s}kB_{jr}\partial_{\theta^i}[m_{sj}(y)(s_p(x))_r]$$

## B.5 Diffusion Kernel Stein Discrepancies as Statistical Divergences

In this section, we prove that DKSD is a statistical divergence and provide sufficent conditions on the matrix-valued kernel.

### B.5.1 Proof of Proposition 1: DKSD as statistical divergence

By Stoke's theorem $\int_{\mathcal{X}} \mathcal{S}_q[v] \mathrm{d}\mathbb{Q} = \int_{\mathcal{X}} \nabla \cdot (qmv) \mathrm{d}x = 0$, thus $\int_{\mathcal{X}} \mathcal{S}_p[v] \mathrm{d}\mathbb{Q} = \int_{\mathcal{X}} (\mathcal{S}_p[v] - \mathcal{S}_q[v]) \mathrm{d}\mathbb{Q} = \int_{\mathcal{X}} (s_p - s_q) \cdot v \mathrm{d}\mathbb{Q}$, and by assumption $\int_{\mathcal{X}} \mathcal{S}_q[K] \mathrm{d}\mathbb{Q} = \int_{\mathcal{X}} \nabla \cdot (qmK) \mathrm{d}x = 0$. Moreover, with $s_p = m^\top \nabla \log p$, and $\delta_{p,q} \equiv s_p - s_q$. Hence

$$
\begin{aligned}
\mathrm{DKSD}_{K,m}(\mathbb{Q}, \mathbb{P})^2 &= \int_{\mathcal{X}} \int_{\mathcal{X}} \mathcal{S}_p^2 \big[ \mathcal{S}_p^1 K(x,y) \big] \mathrm{d}\mathbb{Q}(y) \mathrm{d}\mathbb{Q}(x) \\
&= \int_{\mathcal{X}} \int_{\mathcal{X}} (s_p(y) - s_p(y)) \cdot \big[ \mathcal{S}_p^1 K(x,y) \big] \mathrm{d}\mathbb{Q}(y) \mathrm{d}\mathbb{Q}(x) \\
&= \int_{\mathcal{X}} (s_p(y) - s_p(y)) \mathrm{d}\mathbb{Q}(y) \cdot \int_{\mathcal{X}} \big[ \mathcal{S}_p^1 K(x,y) \big] \mathrm{d}\mathbb{Q}(x) \\
&= \int_{\mathcal{X}} (s_p(y) - s_p(y)) \mathrm{d}\mathbb{Q}(y) \cdot \int_{\mathcal{X}} \big[ \mathcal{S}_p^1 K(x,y) - \mathcal{S}_q^1 K(x,y) \big] \mathrm{d}\mathbb{Q}(x) \\
&= \int_{\mathcal{X}} (s_p(y) - s_p(y)) \mathrm{d}\mathbb{Q}(y) \cdot \int_{\mathcal{X}} [(s_p(x) - s_p(x)) \cdot K(x,y)] \mathrm{d}\mathbb{Q}(x) \\
&= \int_{\mathcal{X}} \int_{\mathcal{X}} q(x) \delta_{p,q}(x)^\top K(x,y) \delta_{p,q}(y) q(y) \mathrm{d}x \mathrm{d}y \\
&= \int_{\mathcal{X}} \int_{\mathcal{X}} \mathrm{d}\mu^\top(x) K(x,y) \mathrm{d}\mu(y).
\end{aligned}
$$

where $\mu(\mathrm{d}x) \equiv q(x) \delta_{p,q}(x) \mathrm{d}x$, which is a finite measure by assumption. If $\mathcal{S}(q,p) = 0$, then since $K$ is IPD we have $q\delta_{p,q} \equiv 0$, and since $q > 0$ and $m$ is invertible we must have $\nabla \log p = \nabla \log q$ and thus $q = p$.

### B.5.2 Proof of Proposition 2: IPD matrix kernels

Let $\mu$ be a finite signed vector measure. $(i)$ If each $k^i$ is IPD, then $\int \mathrm{d}\mu^\top K \mathrm{d}\mu = \int k^i(x,y) \mathrm{d}\mu_i(x) \mathrm{d}\mu_i(y) \geq 0$ with equality iff $\mu_i \equiv 0$ for all $i$. Conversely suppose $\int k^i(x,y) \mathrm{d}\mu_i(x) \mathrm{d}\mu_i(y) \geq 0$ with equality iff $\mu_i \equiv 0$ for all $i$. Suppose $k^j$ is not IPD for some $j$, then there exists a finite non-zero signed measure $\nu$ s.t., $\int k^j \mathrm{d}\nu \otimes \mathrm{d}\nu \leq 0$, so if we define the vector measure $\mu_i \equiv \delta_{ij} \nu$, which is non-zero and finite, then $\int k^i(x,y) \mathrm{d}\mu_i(x) d\mu_i(y) \leq 0$ which contradicts the assumption. For $(ii)$, we first diagonalise $B = R^\top D R$ where $R$ is orthogonal and $D$ diagonal with positive entries $\lambda_i > 0$. Then

$$
\int \mathrm{d}\mu^\top K \mathrm{d}\mu = \int k \mathrm{d}\mu^\top R^\top D R \mathrm{d}\mu = \int k (R\mathrm{d}\mu)^\top D(R\mathrm{d}\mu) = \int k(x,y) \lambda_i \mathrm{d}\nu_i(x) \mathrm{d}\nu_i(y),
$$

where $\nu \equiv R\mu$ is finite and non-zero, since $\mu$ is non-zero and $R$ is invertible, thus maps non-zero vectors to non-zero vectors. Clearly if $k$ is IPD then $\int \mathrm{d}\mu^\top K \mathrm{d}\mu \geq 0$ with equality iff $\nu_i \equiv 0$ for all $i$. Suppose $K$ is IPD but $k$ is not, then there exists finite non-zero signed measure $\nu$ for which $\int k \mathrm{d}\nu \otimes \mathrm{d}\nu \leq 0$, but then setting $\mu \equiv R^\top \xi$, with $\xi_i \equiv \delta_{ij} \nu$ which is finite and non-zero, implies $\int \mathrm{d}\mu^\top K \mathrm{d}\mu = \int k \mathrm{d}\xi^\top D \mathrm{d}\xi = \lambda_j \int k \mathrm{d}\nu \otimes \mathrm{d}\nu \leq 0$.

## B.6 Diffusion Score Matching

Another example of SD is the diffusion score matching (DSM) discrepancy, as introduced below:

### B.6.1 Proof of Theorem 2: Diffusion Score Matching

Note that the Stein operator satisfies

$$
\mathcal{S}_p[g] = \frac{\nabla \cdot (pmg)}{p} = \frac{\langle \nabla p, mg \rangle + p \nabla \cdot (mg)}{p} = \langle \nabla \log p, mg \rangle + \nabla \cdot (mg) = \langle m^\top \nabla \log p, g \rangle + \nabla \cdot (mg).
$$

Since $\int_{\mathcal{X}} \mathcal{S}_q[g]\mathrm{d}\mathbb{Q} = 0$, we have

$$D(\mathbb{Q}\|\mathbb{P}) = \sup_{g \in \mathcal{G}} \left| \int_{\mathcal{X}} \mathcal{S}_p[g](x)\mathbb{Q}(\mathrm{d}x) \right|^2 = \sup_{g \in \mathcal{G}} \left| \int_{\mathcal{X}} (\mathcal{S}_p[g](x) - \mathcal{S}_q[g](x))\mathbb{Q}(\mathrm{d}x) \right|^2$$

$$= \sup_{g \in \mathcal{G}} \left| \int_{\mathcal{X}} ((\nabla \log p - \nabla \log q) \cdot (mg))\mathrm{d}\mathbb{Q} \right|^2,$$

$$= \sup_{g \in \mathcal{G}} \left| \left\langle m^\top (\nabla \log p - \nabla \log q), g \right\rangle_{L^2(\mathbb{Q})} \right|^2$$

$$= \left\| m^\top (\nabla \log p - \nabla \log q) \right\|_{L^2(\mathbb{Q})}^2$$

$$= \int_{\mathcal{X}} \left\| m^\top (\nabla \log p - \nabla \log q) \right\|_2^2 \mathrm{d}\mathbb{Q},$$

where we have used the fact that $\mathcal{G}$ is dense in the unit ball of $L^2(\mathbb{Q})$ (since smooth functions with compact support are dense in $L^2(\mathbb{Q})$), and that the supremum over a dense subset of the continuous functional $F(\cdot) \equiv \left\langle m^\top (\nabla \log p - \nabla \log q), \cdot \right\rangle_{L^2(\mathbb{Q})}$ is equal to the supremum over the closure, $\sup_{\mathcal{G}} F = \sup_{\overline{\mathcal{G}}} F$. Suppose $D(\mathbb{Q}\|\mathbb{P}) = 0$. Then since $q > 0$ we must have $\left\| m^\top (\nabla \log p - \nabla \log q) \right\|_2^2 = 0$, i.e., $m^\top (\nabla \log p - \nabla \log q) = 0$, i.e., $\nabla(\log p - \log q) = 0$. Thus $\log(p/q) = c$, so $p = qe^c$ and integrating implies $c = 0$, so $D(\mathbb{Q}\|\mathbb{P}) = 0$ iff $\mathbb{Q} = \mathbb{P}$ a.e..

To obtain the estimator we will use the divergence theorem, which holds for example if $X, \nabla \cdot X \in L^1(\mathbb{R}^d)$ for $X = qmm^\top \nabla \log p$ (see theorem 2.36, 2.28 [59] or theorem 2.38 for weaker conditions). Note

$$\left\| m^\top (\nabla \log p - \nabla \log q) \right\|_2^2 = \|m^\top \nabla \log p\|_2^2 + \|m^\top \nabla \log q\|_2^2 - 2m^\top \nabla \log p \cdot m^\top \nabla \log q$$

thus we have

$$\int_{\mathcal{X}} \left\langle m^\top \nabla \log p, m^\top \nabla \log q \right\rangle \mathrm{d}\mathbb{Q} = \int_{\mathcal{X}} \left\langle \nabla \log q, mm^\top \nabla \log p \right\rangle \mathrm{d}Q$$

$$= \int_{\mathcal{X}} \left\langle \nabla q, mm^\top \nabla \log p \right\rangle \mathrm{d}x$$

$$= \int_{\mathcal{X}} \left( \nabla \cdot (qmm^\top \nabla \log p) - q\nabla \cdot (mm^\top \nabla \log p) \right) \mathrm{d}x$$

$$= - \int_{\mathcal{X}} q\nabla \cdot (mm^\top \nabla \log p) \mathrm{d}x$$

$$= - \int_{\mathcal{X}} \nabla \cdot (mm^\top \nabla \log p) \mathrm{d}\mathbb{Q}.$$

### B.6.2 Diffusion Score Matching Estimators

As for the standard SM estimator, the DSM is only defined for distributions with sufficiently smooth densities. However the $\theta$-dependent part of $\mathrm{DSM}_m(\mathbb{Q}, \mathbb{P}_\theta)$ [3]

$$\int_{\mathcal{X}} \left( \left\| m^\top \nabla_x \log p_\theta \right\|_2^2 + 2\nabla \cdot (mm^\top \nabla \log p_\theta) \right) \mathrm{d}\mathbb{Q}$$

$$= \int_{\mathcal{X}} \left( \left\| m^\top \nabla_x \log p_\theta \right\|_2^2 + 2 (\langle \nabla \cdot (mm^\top), \nabla \log p \rangle + \mathrm{Tr}[mm^\top \nabla^2 \log p]) \right) \mathrm{d}\mathbb{Q},$$

does not depend on the density of $\mathbb{Q}$. An unbiased estimator for this quantity follows by replacing $\mathbb{Q}$ with the empirical random measure $\mathbb{Q}_n \equiv \frac{1}{n} \sum_i \delta_{X_i}$ where $X_i \sim \mathbb{Q}$ are independent. Hence we consider the estimator

$$\hat{\theta}_n^{\mathrm{DSM}} \equiv \mathrm{argmin}_{\theta \in \Theta} \mathbb{Q}_n \left( \left\| m^\top \nabla_x \log p_\theta \right\|_2^2 + 2 (\langle \nabla \cdot (mm^\top), \nabla \log p_\theta \rangle + \mathrm{Tr}[mm^\top \nabla^2 \log p_\theta]) \right).$$

In components, this corresponds to:

$$\hat{\theta}_n^{\mathrm{DSM}} = \mathrm{argmin}_{\theta \in \Theta} \int_{\mathcal{X}} \mathrm{d}\mathbb{Q}(x) \|m(x)^\top \nabla_x \log p(x|\theta)\|_2^2 + 2\sum_{j,k,l=1}^d \partial_{x^j}\partial_{x^k} \log p(x|\theta)m_{kl}(x)m_{jl}(x)$$

$$+ 2\sum_{j,k,l=1}^d \partial_{x^k} \log p(x|\theta)(\partial_{x^j}m_{kl}(x)m_{jl}(x) + m_{kl}(x)\partial_{x^j}m_{jl}(x))$$

### B.6.3 Proof of Theorem 10: DSM as a limit of DKSD

We now consider the the limit in which DKSD converges to DSM:

**Theorem 10 (DSM as a limit of DKSD).** *Let $\mathbb{Q}$ be a distribution on $\mathbb{R}^d$ with $q > 0$ and suppose $s_p - s_q \in C(\mathbb{R}^d) \cap L^2(\mathbb{Q})$. Let $\Phi_\gamma(s) \equiv \gamma^{-d}\Phi(s/\gamma)$, $\gamma > 0$, $\Phi \in L^1(\mathbb{R}^d)$, $\Phi > 0$ and $\int_{\mathbb{R}^d} \Phi(s)\mathrm{d}s = 1$. Consider the reproducing kernel $k_\gamma^q(x,y) = k_\gamma(x,y)/\sqrt{q(x)q(y)} = \Phi_\gamma(x-y)/\sqrt{q(x)q(y)}$, and set $K_\gamma^q \equiv Bk_\gamma^q$. Then, $\mathrm{DKSD}_{K_\gamma^q, m}(\mathbb{Q}\|\mathbb{P})^2 \to \mathrm{DSM}_m(\mathbb{Q}\|\mathbb{P})$, as $\gamma \to 0$.*

We use the following lemma as a stepping stone.

**Lemma 1.** *Suppose $\Phi \in L^1(\mathbb{R}^d)$, $\Phi > 0$ and $\int \Phi(s)\,\mathrm{d}s = 1$. Let $f, g \in C(\mathbb{R}^d) \cap L^2(\mathbb{R}^d)$, then defining $K_\gamma \equiv B\Phi_\gamma$ where $\Phi_\gamma(s) \equiv \gamma^{-d}\Phi(s/\gamma)$ and $\gamma > 0$, we have*

$$\int \int f(x)^\top K_\gamma(x,y)g(y)\mathrm{d}x\,\mathrm{d}y \;\to\; \int f(x)^\top Bg(x)\,\mathrm{d}x, \quad as \quad \gamma \to 0.$$

**Proof**    We rewrite

$$\int_{\mathcal{X}} \int_{\mathcal{X}} f(x)^\top B\Phi_\gamma(x-y)g(y)\,\mathrm{d}x\,\mathrm{d}y = \int_{\mathcal{X}} \int_{\mathcal{X}} f(x)^\top Bg(x-s)\mathrm{d}x\,\Phi_\gamma(s)\,\mathrm{d}s = \int_{\mathcal{X}} H(s)\Phi_\gamma(s)\,\mathrm{d}s,$$

where $H : \mathcal{X} \to \mathbb{R}$ is defined by

$$H(s) \equiv \int_{\mathcal{X}} f(x)^\top Bg(x-s)\,\mathrm{d}x = \int_{\mathcal{X}}\langle f(x), Bg(x-s)\rangle_{\mathbb{R}^d}\mathrm{d}x \equiv \int_{\mathcal{X}}\langle f(x), g(x-s)\rangle_B\mathrm{d}x.$$

Since $f, g \in C(\mathbb{R}^d) \cap L^2(\mathbb{R}^d)$, the function $H(s)$ is continuous, bounded, $|H(s)| \leq A\|f\|_{L^2(\mathbb{R}^d)}\|g\|_{L^2(\mathbb{R}^d)}$ for a constant $A > 0$ depending only on $B$, and $H(0) = \int f(x)^\top Bg(x)\,\mathrm{d}x$. Given $\delta > 0$, we can split the integral as follows:

$$\int_{|s|<\delta} H(s)\Phi_\gamma(s)\,\mathrm{d}s \;+\; \int_{|s|>\delta} H(s)\Phi_\gamma(s)\,\mathrm{d}s \equiv I_1 + I_2.$$

By continuity, given $\epsilon \in (0,1)$ there exists $\delta > 0$ such that $|H(s) - H(0)| < \epsilon$ for all $|s| < \delta$. Let $I_{<\delta} \equiv \int_{|y|<\delta} \Phi_\gamma(y)\,\mathrm{d}y > 0$ since $\Phi > 0$. Consider

$$
\begin{aligned}
I_1 - H(0) &= \int_{|s|<\delta} \Phi_\gamma(s)H(s)\mathrm{d}s - H(0) = \int_{|s|<\delta}\Phi_\gamma(s)\Big(H(s) - \tfrac{H(0)}{I_{<\delta}}\Big)\mathrm{d}s \\
&= \int_{|s|<\delta} \tfrac{\Phi_\gamma(s)}{I_{<\delta}}(H(s)I_{<\delta} - H(0))\,\mathrm{d}s.
\end{aligned}
$$

Clearly $\int \Phi_\gamma(s)\mathrm{d}s = \int \gamma^{-d}\Phi(s/\gamma)\mathrm{d}s = \int \Phi(z)\mathrm{d}z = 1$, since $z \equiv s/\gamma$ implies $\mathrm{d}z = \gamma^{-d}\mathrm{d}s$, so

$$I_{<\delta} = 1 - I_{>\delta} = 1 - \int_{|y|>\delta/\gamma} \Phi(y)\,\mathrm{d}y.$$

Then since $\Phi$ is integrable, there exists $\gamma_0(\delta) > 0$ s.t. for $\gamma < \gamma_0(\delta)$ we have $\int_{|y|>\delta/\gamma} \Phi(y)\,\mathrm{d}y < \epsilon$ and thus $0 < 1 - \epsilon < I_{<\delta} < 1$. Therefore, for $\gamma < \gamma_0(\delta)$ :

$$
\begin{aligned}
|I_1 - H(0)| &= \Big|\int_{|s|<\delta}\tfrac{\Phi_\gamma(s)}{I_{<\delta}}(H(s)I_{<\delta} - H(0))\mathrm{d}s\Big| \\
&\leq \int_{|s|<\delta}\tfrac{\Phi_\gamma(s)}{I_{<\delta}}|((H(s)-H(0))I_{<\delta} + H(0)(I_{<\delta}-1))|\mathrm{d}s \\
&\leq \int_{|s|<\delta}\tfrac{\Phi_\gamma(s)}{I_{<\delta}}(|H(s)-H(0)|I_{<\delta} + |1-I_{<\delta}|H(0))\mathrm{d}s \\
&\leq \int_{|s|<\delta}\tfrac{\Phi_\gamma(s)}{I_{<\delta}}(\epsilon I_{<\delta} + \epsilon H(0))\mathrm{d}s \\
&\leq \epsilon \int_{|z|<\delta/\gamma}\Phi(z)\mathrm{d}z + H(0)\epsilon \leq (1 + H(0))\epsilon.
\end{aligned}
$$

For the second term, since $H$ is bounded we have

$$I_2 = \int_{|s|>\delta} H(s)\Phi_\gamma(s)\mathrm{d}s = \int_{|s|>\delta/\gamma} H(\gamma s)\Phi(s)\mathrm{d}s \leq \|H\|_\infty \int_{|s|>\delta/\gamma}\Phi(s)\mathrm{d}s,$$

so that, $|I_2| \leq \|H\|_\infty \epsilon$, for $\gamma < \gamma_0(\delta)$. It follows that

$$
\begin{aligned}
\Big|\int\int f(x)^\top K_\gamma(x,y)g(y)\mathrm{d}x\,\mathrm{d}y - \int f(x)^\top Bg(x)\,\mathrm{d}x\Big| &= \Big|\int H(s)\Phi_\gamma(s)\mathrm{d}s - H(0)\Big| \\
&= |I_1 + I_2 - H(0)| \\
&\leq |I_1 - H(0)| + |I_2| \;\to\; 0,
\end{aligned}
$$

as $\gamma \to 0$ as required. ∎

We note that $f \in L^2(\mathbb{Q})$ if and only if $f\sqrt{q} \in L^2(\mathbb{R}^d)$. Therefore applying the previous result, we have that

$$\int_{\mathcal{X}} \int_{\mathcal{X}} f(x)^\top K_\gamma^q(x,y) g(y) \,\mathrm{d}\mathbb{Q}(x) \,\mathrm{d}\mathbb{Q}(y) = \int_{\mathcal{X}} \int_{\mathcal{X}} \left(\sqrt{q(x)} f(x)\right)^\top K_\gamma(x,y) \left(g(y)\sqrt{q(y)}\right) \mathrm{d}x \mathrm{d}y$$
$$\to \int_{\mathcal{X}} f(x)^\top B g(x) \mathrm{d}\mathbb{Q}(x), \quad \text{as} \quad \gamma \to 0.$$

Note that if $k$ is a (scalar) kernel function, then $(x,y) \mapsto r(x)k(x,y)r(y)$ is a kernel for any function $r : \mathcal{X} \to \mathbb{R}$, and thus $k_\gamma^q$ defines a sequence of kernels parametrised by a scale parameter $\gamma > 0$. It follows that the sequence of DKSD paramaterised by $K_\gamma^q$

$$\mathrm{DKSD}_{K_\gamma^q, m}(\mathbb{Q}\|\mathbb{P})^2 = \int_{\mathcal{X}} \int_{\mathcal{X}} q(x)\delta_{p,q}(x)^\top K_\gamma^q(x,y)\delta_{p,q}(y)q(y)\mathrm{d}x\mathrm{d}y$$

converges to DSM with inner product $\langle \cdot, \cdot \rangle_B \equiv \langle \cdot, B \cdot \rangle_2$ on $\mathbb{R}^d$.

$$\mathrm{DSM}_m(\mathbb{Q}\|\mathbb{P}) = \int_{\mathcal{X}} \delta_{q,p}(x)^\top B \delta_{q,p}(x)\mathrm{d}\mathbb{Q} = \int_{\mathcal{X}} \|m^\top(\nabla \log p - \nabla \log q)\|_B^2 \mathrm{d}\mathbb{Q}$$

# C   Information Semi-Metrics of Minimum Stein Discrepancy Estimators

In this section, we derive expressions for the metric tensor of DKSD and DSM. Let $\mathcal{P}_\Theta$ be a parametric family of probability measures on $\mathcal{X}$. Given a map $D : \mathcal{P}_\Theta \times \mathcal{P}_\Theta \to \mathbb{R}$, for which $D(\mathbb{P}_1\|\mathbb{P}_2) = 0$ iff $\mathbb{P}_1 = \mathbb{P}_2$, its associated information semi-metric is defined as the map $\theta \mapsto g(\theta)$, where $g(\theta)$ is the symmetric bilinear form $g(\theta)_{ij} = -\frac{1}{2}\frac{\partial^2}{\partial \alpha^i \partial \theta^j} D(\mathbb{P}_\alpha\|\mathbb{P}_\theta)|_{\alpha=\theta}$. When $g$ is positive definite, we can use it to perform (Riemannian) gradient descent on $\mathcal{P}_\Theta \cong \Theta$.

## C.1   Proof of Proposition 3: Information Semi-Metric of Diffusion Kernel Stein Discrepancy

From Proposition 1 we have

$$\mathrm{DKSD}_{K,m}(\mathbb{P}_\alpha, \mathbb{P}_\theta)^2 = \int_{\mathcal{X}} \int_{\mathcal{X}} p_\alpha(x)\delta_{p_\theta, p_\alpha}(x)^\top K(x,y)\delta_{p_\theta, p_\alpha}(y)p_\alpha(y)\mathrm{d}x\mathrm{d}y$$

where $\delta_{p_\theta, p_\alpha} = m_\theta^\top(\nabla \log p_\theta - \nabla \log p_\alpha)$. Thus

$$\partial_{\alpha^i}\partial_{\theta^j} \mathrm{DKSD}_{K,m}(\mathbb{P}_\alpha, \mathbb{P}_\theta)^2 = \partial_{\alpha^i}\partial_{\theta^j} \int_{\mathcal{X}} \int_{\mathcal{X}} p_\alpha(x)\delta_{p_\theta, p_\alpha}(x)^\top K(x,y)\delta_{p_\theta, p_\alpha}(y)p_\alpha(y)\mathrm{d}x\mathrm{d}y$$
$$= \partial_{\alpha^i} \int_{\mathcal{X}} \int_{\mathcal{X}} p_\alpha(x)\partial_{\theta^j}\delta_{p_\theta, p_\alpha}(x)^\top K(x,y)\delta_{p_\theta, p_\alpha}(y)p_\alpha(y)\mathrm{d}x\mathrm{d}y$$
$$+ \partial_{\alpha^i} \int_{\mathcal{X}} \int_{\mathcal{X}} p_\alpha(x)\delta_{p_\theta, p_\alpha}(x)^\top K(x,y)\partial_{\theta^j}\delta_{p_\theta, p_\alpha}(y)p_\alpha(y)\mathrm{d}x\mathrm{d}y,$$

and using $\delta_{p_\theta, p_\theta} = 0$, we get:

$$\partial_{\alpha^i} \int_{\mathcal{X}} \int_{\mathcal{X}} p_\alpha(x)\partial_{\theta^j}\delta_{p_\theta, p_\alpha}(x)^\top K(x,y)\delta_{p_\theta, p_\alpha}(y)p_\alpha(y)\mathrm{d}x\mathrm{d}y\big|_{\alpha=\theta}$$
$$= \partial_{\alpha^i} \int_{\mathcal{X}} \int_{\mathcal{X}} p_\alpha(x)\left(\partial_{\theta^j}m_\theta^\top(\nabla \log p_\theta - \nabla \log p_\alpha) + m_\theta^\top \partial_{\theta^j}\nabla \log p_\theta\right)^\top K(x,y)\delta_{p_\theta, p_\alpha}(y)p_\alpha(y)\mathrm{d}x\mathrm{d}y\big|_{\alpha=\theta}$$
$$= \int_{\mathcal{X}} \int_{\mathcal{X}} p_\alpha(x)\left(m_\theta^\top \partial_{\theta^j}\nabla \log p_\theta\right)^\top K(x,y)\partial_{\alpha^i}\delta_{p_\theta, p_\alpha}(y)p_\alpha(y)\mathrm{d}x\mathrm{d}y\big|_{\alpha=\theta}$$
$$= -\int_{\mathcal{X}} \int_{\mathcal{X}} p_\alpha(x)\left(m_\theta^\top \partial_{\theta^j}\nabla \log p_\theta\right)^\top K(x,y)\left(m_\theta^\top \partial_{\alpha^i}\nabla \log p_\alpha\right)(y)p_\alpha(y)\mathrm{d}x\mathrm{d}y\big|_{\alpha=\theta}$$
$$= -\int_{\mathcal{X}} \int_{\mathcal{X}} \left(m_\theta^\top \partial_{\theta^j}\nabla \log p_\theta\right)^\top(x)K(x,y)\left(m_\theta^\top \partial_{\theta^i}\nabla \log p_\theta\right)(y)\mathrm{d}\mathbb{P}_\theta(x)\mathrm{d}\mathbb{P}_\theta(y).$$

Similarly, we also get:

$$\partial_{\alpha^i} \int_{\mathcal{X}} \int_{\mathcal{X}} p_\alpha(x)\delta_{p_\theta, p_\alpha}(x)^\top K(x,y)\partial_{\theta^j}\delta_{p_\theta, p_\alpha}(y)p_\alpha(y)\mathrm{d}x\mathrm{d}y\big|_{\alpha=\theta}$$
$$= -\int_{\mathcal{X}} \int_{\mathcal{X}} \left(m_\theta^\top \partial_{\theta^i}\nabla \log p_\theta\right)^\top(x)K(x,y)\left(m_\theta^\top \partial_{\theta^j}\nabla \log p_\theta\right)(y)\mathrm{d}\mathbb{P}_\theta(x)\mathrm{d}\mathbb{P}_\theta(y)$$
$$= -\int_{\mathcal{X}} \int_{\mathcal{X}} \left(m_\theta^\top \partial_{\theta^i}\nabla \log p_\theta\right)^\top(y)K(y,x)\left(m_\theta^\top \partial_{\theta^j}\nabla \log p_\theta\right)(x)\mathrm{d}\mathbb{P}_\theta(y)\mathrm{d}\mathbb{P}_\theta(x)$$
$$= -\int_{\mathcal{X}} \int_{\mathcal{X}} \left(m_\theta^\top \partial_{\theta^i}\nabla \log p_\theta\right)^\top(y)K(x,y)^\top\left(m_\theta^\top \partial_{\theta^j}\nabla \log p_\theta\right)(x)\mathrm{d}\mathbb{P}_\theta(y)\mathrm{d}\mathbb{P}_\theta(x)$$
$$= -\int_{\mathcal{X}} \int_{\mathcal{X}} \left(m_\theta^\top \partial_{\theta^j}\nabla \log p_\theta\right)(x)^\top K(x,y)\left(m_\theta^\top \partial_{\theta^i}\nabla \log p_\theta\right)(y)\mathrm{d}\mathbb{P}_\theta(y)\mathrm{d}\mathbb{P}_\theta(x).$$

Hence, we conclude that

$$\frac{1}{2}\partial_{\alpha^i}\partial_{\theta^j}\,\mathrm{DKSD}_{K,m}(\mathbb{P}_\alpha,\mathbb{P}_\theta)^2 = -\int_{\mathcal{X}}\int_{\mathcal{X}}\big(m_\theta^\top\partial_{\theta^j}\nabla\log p_\theta\big)(x)^\top K(x,y)\big(m_\theta^\top\partial_{\theta^i}\nabla\log p_\theta\big)(y)\mathrm{d}\mathbb{P}_\theta(y)\mathrm{d}\mathbb{P}_\theta(x)$$

The information tensor is positive semi-definite. Indeed writing $V_\theta(y) \equiv m_\theta^\top(y)\nabla_y\langle v, \nabla_\theta\log p_\theta\rangle$:

$$
\begin{aligned}
\langle v, g(\theta)v\rangle &= v^i g_{ij}(\theta)v^j\\
&= \int_{\mathcal{X}}\int_{\mathcal{X}}\big(m_\theta^\top(x)\nabla_x\langle v, \nabla_\theta\log p_\theta\rangle\big)^\top K(x,y)\big(m_\theta^\top(y)\nabla_y\langle v, \nabla_\theta\log p_\theta\rangle\big)\mathrm{d}\mathbb{P}_\theta(x)\mathrm{d}\mathbb{P}_\theta(y)\\
&= \int_{\mathcal{X}}\int_{\mathcal{X}}\langle m_\theta^\top(x)\nabla_x\langle v, \nabla_\theta\log p_\theta\rangle, K(x,y)m_\theta^\top(y)\nabla_y\langle v, \nabla_\theta\log p_\theta\rangle\rangle\mathrm{d}\mathbb{P}_\theta(x)\mathrm{d}\mathbb{P}_\theta(y)\\
&= \int_{\mathcal{X}}\int_{\mathcal{X}}\langle V_\theta(x), K(x,y)V_\theta(y)\rangle\mathrm{d}\mathbb{P}_\theta(x)\mathrm{d}\mathbb{P}_\theta(y) \geq 0
\end{aligned}
$$

since $K$ is IPD.

### C.2 Proof of Proposition 4: Information Semi-Metric of Diffusion Score Matching

**Proof** The information metric is given by $g(\theta)_{ij} = -\frac{1}{2}\frac{\partial^2}{\partial\alpha^i\partial\theta^j}\,\mathrm{DSM}(p_\alpha\|p_\theta)|_{\alpha=\theta}$. Recall

$$\mathrm{DSM}(p_\alpha\|p_\theta) = \int_{\mathcal{X}}\big\|m^\top(\nabla\log p_\theta - \nabla\log p_\alpha)\big\|_2^2 p_\alpha\mathrm{d}x.$$

Moreover

$$
\begin{aligned}
\frac{1}{2}\partial_{\alpha^i}\partial_{\theta^j}\,\mathrm{DSM}(p_\alpha\|p_\theta)\big|_{\alpha=\theta} &= \frac{1}{2}\partial_{\alpha^i}\partial_{\theta^j}\int_{\mathcal{X}}\big\|m^\top(\nabla\log p_\theta - \nabla\log p_\alpha)\big\|_2^2 p_\alpha\mathrm{d}x\big|_{\alpha=\theta}\\
&= \partial_{\alpha^i}\int_{\mathcal{X}}\big(m^\top(\nabla\log p_\theta - \nabla\log p_\alpha)\big)\cdot\big(m^\top\partial_{\theta^j}\nabla\log p_\theta\big)p_\alpha\mathrm{d}x\big|_{\alpha=\theta}\\
&= \int_{\mathcal{X}}\big(m^\top(\nabla\log p_\theta - \nabla\log p_\alpha)\big)\cdot\big(m^\top\partial_{\theta^j}\nabla\log p_\theta\big)\partial_{\alpha^i}p_\alpha\mathrm{d}x\big|_{\alpha=\theta}\\
&\quad - \int_{\mathcal{X}}\big(m^\top\partial_{\alpha^i}\nabla\log p_\alpha\big)\cdot\big(m^\top\partial_{\theta^j}\nabla\log p_\theta\big)p_\alpha\mathrm{d}x\big|_{\alpha=\theta}\\
&= -\int_{\mathcal{X}}\big(m^\top\partial_{\theta^i}\nabla\log p_\theta\big)\cdot\big(m^\top\partial_{\theta^j}\nabla\log p_\theta\big)\mathrm{d}\mathbb{P}_\theta.
\end{aligned}
$$

Finally $g$ is semi-positive definite,

$$
\begin{aligned}
\langle v, g(\theta)v\rangle = v^i g_{ij}(\theta)v^j &= \int_{\mathcal{X}} v^i m_{rs}^\top\partial_{x^s}\partial_{\theta^i}\log p_\theta m_{rl}^\top\partial_{x^l}\partial_{\theta^j}\log p_\theta v^j\mathrm{d}\mathbb{P}_\theta\\
&= \int_{\mathcal{X}} m_{rs}^\top\partial_{x^s}\langle v, \nabla_\theta\log p_\theta\rangle m_{rl}^\top\partial_{x^l}\langle v, \nabla_\theta\log p_\theta\rangle\mathrm{d}\mathbb{P}_\theta\\
&= \int_{\mathcal{X}}\big\langle m^\top\nabla_x\langle v, \nabla_\theta\log p_\theta\rangle, m^\top\nabla_x\langle v, \nabla_\theta\log p_\theta\rangle\big\rangle\mathrm{d}\mathbb{P}_\theta\\
&= \int_{\mathcal{X}}\|m^\top\nabla_x\langle v, \nabla_\theta\log p_\theta\rangle\|^2\mathrm{d}\mathbb{P}_\theta \geq 0
\end{aligned}
$$

∎

## D  Proofs of Consistency and Asymptotic Normality for minimum Stein Discrepancy Estimators

In this appendix, we prove several results concerning the consistency and asymptotic normality of DKSD and DSM estimators.

### D.1  Diffusion Kernel Stein Discrepancies

Given the Stein kernel (2) we want to estimate $\theta_*^{\mathrm{DKSD}} \equiv \mathrm{argmin}_{\theta\in\Theta}\,\mathrm{DKSD}_{K,m}(\mathbb{Q},\mathbb{P}_\theta)^2 = \mathrm{argmin}_{\theta\in\Theta}\int_{\mathcal{X}}\int_{\mathcal{X}}\underbrace{k_\theta^0(x,y)}\mathbb{Q}(\mathrm{d}x)\mathbb{Q}(\mathrm{d}y)$ using a sequence of estimators $\hat{\theta}_n^{\mathrm{DKSD}} \in \mathrm{argmin}_{\theta\in\Theta}\widehat{\mathrm{DKSD}}_{K,m}(\mathbb{Q},\mathbb{P}_\theta)^2$ that minimise the $U$-statistic approximation (3). We will assume we are in the specified setting $\mathbb{Q} = \mathbb{P}_{\theta^{\mathrm{DKSD}}_*} \in \mathcal{P}_\Theta$. In the misspecified setting it is necessary to further assume the existence of a unique minimiser.

#### D.1.1  Strong Consistency

We first prove a general strong consistency result based on an equicontinuity assumption:

**Lemma 2.** *Let $\mathcal{X} = \mathbb{R}^d$. Suppose $\{\theta \mapsto k_\theta^0(x,y)\}, \{\theta \mapsto \mathbb{Q}_z k_\theta^0(x,z)\}$ are equicontinuous on any compact subset $C \subset \Theta$ for $x, y$ in a sequence of sets whose union has full $\mathbb{Q}$-measure, and $\|s_{p_\theta}(x)\| \le f_1(x), \|\nabla_x \cdot m_\theta(x)\| \le f_2(x), \|\nabla_x \cdot (m_\theta(x) K(x,y))\| \le f_3(x,y), |\mathrm{Tr}[m(y) \nabla_y \nabla_x \cdot (m(x) K)]| \le f_4(x,y)$ hold on $C$, where $f_1(x)\sqrt{K(x,x)_{ii}} \in L^1(\mathbb{Q})$, and $f_4, f_3 f_2, f_1 f_3 \in L^1(\mathbb{Q} \otimes \mathbb{Q})$. Assume further that $\theta \mapsto \mathbb{P}_\theta$ is injective. Then we have a unique minimiser $\theta_*^{\mathrm{DKSD}}$, and if either $\Theta$ is compact, or $\theta_*^{\mathrm{DKSD}} \in int(\Theta)$ and $\Theta$ and $\theta \mapsto \widehat{\mathrm{DKSD}}_{K,m}(\{X_i\}_{i=1}^n, \mathbb{P}_\theta)^2$ are convex, then $\hat{\theta}_n^{\mathrm{DKSD}}$ is strongly consistent.*

**Proof**

Note $\mathrm{DKSD}_{K,m}(\mathbb{Q}, \mathbb{P}_\theta)^2 = 0$ iff $\mathbb{P}_\theta = \mathbb{P}_{\theta_*^{\mathrm{DKSD}}}$ by Proposition 1, which implies $\theta = \theta_*^{\mathrm{DKSD}}$ since $\theta \mapsto \mathbb{P}_\theta$ is injective. Thus we have a unique minimiser at $\theta_*^{\mathrm{DKSD}}$.

Suppose first $\Theta$ is compact and take $C = \Theta$. Note

$$
\begin{aligned}
|k^0(x,y)| \le & |\langle s_p(x), K s_p(y) \rangle| + |\langle \nabla_y \cdot (m(y) K), s_p(x) \rangle| + |\langle \nabla_x \cdot (m(x) K), s_p(y) \rangle| \\
& + |\langle \nabla_x \cdot (m(x) K), \nabla_y \cdot m \rangle| + |\mathrm{Tr}[m(y) \nabla_y \nabla_x \cdot (m(x) K)]| \\
\le & |\langle s_p(x), K s_p(y) \rangle| + f_3(y,x) f_1(x) + f_3(x,y) f_1(y) + f_3(x,y) f_2(y) + f_4(x,y),
\end{aligned}
$$

From the reproducing property $f(x) = \langle f, K(\cdot, x) v \rangle_{\mathcal{H}^d}$, for any $f \in \mathcal{H}^d$, $v \in \mathbb{R}^d$. Using $K(y,x) = K(x,y)^\top$ we have $K(\cdot, x)_{,i} = K(x, \cdot)_{i,}$, where $K(\cdot, x)_{,i}$ and $K(x, \cdot)_{i,}$ denote the $i^{\mathrm{th}}$ column and row respectively, which implies that $K(x, \cdot)_{i,}, K(\cdot, x)_{,i} \in \mathcal{H}^d$ and $f(x)_i = \langle f, K(\cdot, x)_{,i} \rangle_{\mathcal{H}^d}$. Choosing $f = K(\cdot, y)_{,j}$ implies

$$
\begin{aligned}
K(x,y)_{ij} = \langle K(\cdot, y)_{,j}, K(\cdot, x)_{,i} \rangle_{\mathcal{H}^d} & \le \|K(\cdot, y)_{,j}\|_{\mathcal{H}^d} \|K(\cdot, x)_{,i}\|_{\mathcal{H}^d} \\
& = \sqrt{\langle K(\cdot, y)_{,j}, K(\cdot, y)_{,j} \rangle_{\mathcal{H}^d}} \sqrt{\langle K(\cdot, x)_{,i}, K(\cdot, x)_{,i} \rangle_{\mathcal{H}^d}} \\
& = \sqrt{K(y,y)_{jj}} \sqrt{K(x,x)_{ii}}.
\end{aligned}
$$

It follows that

$$
\begin{aligned}
\langle s_p(x), K s_p(y) \rangle = (s_p)_i(x) K(x,y)_{ij} (s_p)_j(y) & \le (s_p)_i(x) \sqrt{K(x,x)_{ii}} \sqrt{K(y,y)_{jj}} (s_p)_j(y) \\
& \le \|s_p(x)\|_\infty \sqrt{K(x,x)_{ii}} \sqrt{K(y,y)_{jj}} \|s_p(y)\|_\infty \\
& \le C f_1(x) \sqrt{K(x,x)_{ii}} \sqrt{K(y,y)_{jj}} f_1(y),
\end{aligned}
$$

where the constant $C > 0$ arises from the norm-equivalence of $\|s_p(y)\|$ and $\|s_p(y)\|_\infty$. Hence $k^0$ is integrable. Thus by theorem 1 [70],

$$
\sup_\theta \left| \widehat{\mathrm{DKSD}}_{K,m}(\{X_i\}_{i=1}^n, \mathbb{P}_\theta)^2 - \mathrm{DKSD}_{K,m}(\mathbb{Q}, \mathbb{P}_\theta)^2 \right| \xrightarrow{a.s.} 0
$$

and $\theta \mapsto \mathrm{DKSD}_{K,m}(\mathbb{Q}, \mathbb{P}_\theta)^2$ are continuous. By theorem 2.1 [56] then $\hat{\theta}_n^{\mathrm{DKSD}} \xrightarrow{a.s.} \theta_*^{\mathrm{DKSD}}$.

On the other hand, if $\Theta$ is convex we follow a similar strategy to the proof of theorem 2.7 [56]. Since $\theta_*^{\mathrm{DKSD}} \in int(\Theta)$, we can find a $\epsilon > 0$ for which $C = \overline{B}(\theta_*^{\mathrm{DKSD}}, 2\epsilon) \subset \Theta$ is a closed ball containing $\theta_*^{\mathrm{DKSD}}$ (which is compact since $\Theta \subset \mathbb{R}^m$). Using the compact case, we know any sequence of estimators $\tilde{\theta}_n^{\mathrm{DKSD}} \in \mathrm{argmin}_{\theta \in C} \widehat{\mathrm{DKSD}}_{K,m}(\{X_i\}_{i=1}^n, \mathbb{P}_\theta)^2$ is strongly consistent for $\theta_*^{\mathrm{DKSD}}$. In particular, there exists $N_0$ a.s. s.t. for $n > N_0$, $\|\tilde{\theta}_n^{\mathrm{DKSD}} - \theta_*^{\mathrm{DKSD}}\| < \epsilon$. If $\theta \notin C$, there exists $\lambda \in [0,1)$ s.t. $\lambda \tilde{\theta}_n^{\mathrm{DKSD}} + (1-\lambda)\theta$ lies on the boundary of the closed ball $C$. Using convexity and the fact $\tilde{\theta}_n^{\mathrm{DKSD}}$ is a minimiser over $C$,

$$
\begin{aligned}
& \widehat{\mathrm{DKSD}}_{K,m}(\{X_i\}_{i=1}^n, \mathbb{P}_{\tilde{\theta}_n^{\mathrm{DKSD}}})^2 \\
& \le \widehat{\mathrm{DKSD}}_{K,m}(\{X_i\}_{i=1}^n, \mathbb{P}_{\lambda \tilde{\theta}_n^{\mathrm{DKSD}} + (1-\lambda)\theta})^2 \\
& \le \lambda \widehat{\mathrm{DKSD}}_{K,m}(\{X_i\}_{i=1}^n, \mathbb{P}_{\tilde{\theta}_n^{\mathrm{DKSD}}})^2 + (1-\lambda) \widehat{\mathrm{DKSD}}_{K,m}(\{X_i\}_{i=1}^n, \mathbb{P}_\theta)^2
\end{aligned}
$$

which implies $\widehat{\mathrm{DKSD}}_{K,m}(\{X_i\}_{i=1}^n, \mathbb{P}_{\tilde{\theta}_n^{\mathrm{DKSD}}})^2 \le \widehat{\mathrm{DKSD}}_{K,m}(\{X_i\}_{i=1}^n, \mathbb{P}_\theta)^2$ and $\tilde{\theta}_n^{\mathrm{DKSD}}$ is the global minimum of $\theta \mapsto \widehat{\mathrm{DKSD}}_{K,m}(\{X_i\}_{i=1}^n, \mathbb{P}_\theta)^2$ for $n > N_0$. ∎

When $k^0$ is Fréchet differentiable on $\Theta$ equicontinuity can be obtained using the Mean value theorem, which simplifies the assumptions under which strong consistency holds.

We now prove our main result for consistency of minimum DKSD estimators: Theorem 3:

**Proof** Let $\|K\| + \|\nabla_x K\| + \|\nabla_x \nabla_y K\| \leq K_\infty$. Note $\|\nabla_y \cdot (m(y)K)\| \leq 2f_2(y)K_\infty$ and $|\text{Tr}[m(y)\nabla_y \nabla_x \cdot (m(x)K)]| \leq 2f_2(y)f_2(x)K_\infty$ so

$$|k^0_\theta(x,y)| \leq f_1(x)K_\infty f_1(y) + 2f_2(x)K_\infty f_1(y) + 2f_2(y)K_\infty f_1(x) + 3K_\infty f_2(x)f_2(y)$$

which is symmetric and integrable by assumption. Let $S_m$, $m = 1, 2, \ldots$ be an increasing sequence of closed balls in $\mathbb{R}^d$, such that $\cup_{m=1}^\infty S_m = \mathbb{R}^d$. Moreover,

$$\|\nabla_\theta \langle s_p(x), K s_p(y)\rangle\| \leq g_1(x)f_1(y)K_\infty + g_1(y)f_1(x)K_\infty$$
$$\|\nabla_\theta \langle \nabla_y \cdot (m(y)K), s_p(x)\rangle\| \leq 2K_\infty g_2(y)f_1(x) + 2f_2(y)g_1(x)K_\infty$$
$$\|\nabla_\theta \langle \nabla_x \cdot (m(x)K), \nabla_y \cdot m\rangle\| \leq 2K_\infty g_2(x)f_2(y) + 2K_\infty f_2(x)g_2(y)$$
$$\|\nabla_\theta \text{Tr}[m(y)\nabla_y \nabla_x \cdot (m(x)K)]\| \leq 2K_\infty g_2(y)f_2(x) + 2K_\infty f_2(y)g_2(x)$$

thus $\|\nabla_\theta k^0_\theta(x,y)\|$ is bounded above by a continuous integrable symmetric function, $(x,y) \mapsto s(x,y)$, which attains a maximum on the compact spaces $S_m \times S_m$. By the MVT applied on the $\mathbb{R}^m$-open neighbourhood of $\Theta$, $|k^0_\theta(x,y) - k^0_\alpha(x,y)| \leq \|\nabla_\theta k^0_\theta(x,y)\|\|\theta - \alpha\| \leq s(x,y)\|\theta - \alpha\| \leq \max_{x,y \in S_m} s(x,y)\|\theta - \alpha\|$, and $k^0_\theta(x,y)$ is equicontinuous in $\theta \in C$ for $x,y \in S_m$. Similarly, since $s$ is integrable, $|\int_\mathcal{X} k^0_\theta(x,y)\mathbb{Q}(dy) - \int_\mathcal{X} k^0_\alpha(x,z)\mathbb{Q}(dz)| \leq \|\nabla_\theta \int_\mathcal{X} k^0_\theta(x,z)d\mathbb{Q}(z)\|\|\theta - \alpha\| \leq \int_\mathcal{X} \|\nabla_\theta k^0_\theta(x,z)\|d\mathbb{Q}(z)\|\theta - \alpha\| \leq \max_{x \in S_m} \mathbb{Q}_z s(x,z)\|\theta - \alpha\| \leq$ is equicontinuous in $\theta \in C$ for $x \in S_m$. The rest follows as in the previous proposition. ∎

### D.1.2 Proof of Theorem 4: Asymptotic Normality

**Proof** Note that $\nabla_\theta \widehat{\text{DKSD}}_{K,m}(\{X_i\}_{i=1}^n, \mathbb{P}_\theta)^2 = \frac{1}{N(N-1)} \sum_{i \neq j} \nabla_\theta k^0_\theta(X_i, X_j)$. Let $\mu(\theta) \equiv \mathbb{Q} \otimes \mathbb{Q}[\nabla_\theta k^0_\theta]$. Assumptions 1 and 2 imply that $\mathbb{Q} \otimes \mathbb{Q}[\|\nabla_\theta k^0_\theta\|^2] < \infty$. By [29, Theorem 7.1 ] it follows that

$$\sqrt{n}\Big(\nabla_\theta \widehat{\text{DKSD}}_{K,m}(\{X_i\}_{i=1}^n, \mathbb{P}_\theta)^2 - \mu(\theta)\Big) \xrightarrow{d} \mathcal{N}(0, 4\Sigma(\theta))$$

where

$$\Sigma = \mathbb{Q}\big[\mathbb{Q}_2\big[\nabla_\theta k^0_\theta - \mu(\theta)\big] \otimes \mathbb{Q}_2\big[\nabla_\theta k^0_\theta - \mu(\theta)\big]\big]$$
$$= \int_\mathcal{X}\big(\int_\mathcal{X} \nabla_\theta k^0_\theta(x,y)d\mathbb{Q}(y) - \mu(\theta)\big) \otimes \big(\int_\mathcal{X} \nabla_\theta k^0_\theta(x,z)d\mathbb{Q}(z) - \mu(\theta)\big)d\mathbb{Q}(x)$$

Note that $\mu(\theta_*^{\text{DKSD}}) = \mathbb{Q} \otimes \mathbb{Q}[\nabla_\theta k^0_\theta|_{\theta_*^{\text{DKSD}}}] = \nabla_\theta\big(\mathbb{Q} \otimes \mathbb{Q}[k^0_\theta]\big)|_{\theta=\theta_*^{\text{DKSD}}}$, and if $\mathbb{Q} \otimes \mathbb{Q}[k^0_\theta]$ is differentiable around $\theta_*^{\text{DKSD}}$, then the first order optimality condition implies $\mu(\theta_*^{\text{DKSD}}) = 0$.

Consider now $\nabla_\theta \nabla_\theta \widehat{\text{DKSD}}_{K,m}(\{X_i\}, \mathbb{P}_\theta)^2 = \frac{1}{n(n-1)} \sum_{i \neq j} \nabla_\theta \nabla_\theta k^0_\theta(X_i, X_j)$. Note

$$\|\nabla_\theta \nabla_\theta \nabla_\theta \langle s_p(x), K s_p(y)\rangle\| \lesssim g_1(x)K_\infty f_1(y) + f_1(x)K_\infty g_1(y)$$
$$\|\nabla_\theta \nabla_\theta \nabla_\theta \langle \nabla_y \cdot (m(y)K), s_p(x)\rangle\| \lesssim g_2(y)K_\infty f_1(x) + f_2(y)K_\infty g_1(x)$$
$$\|\nabla_\theta \nabla_\theta \nabla_\theta \langle \nabla_x \cdot (m(x)K), \nabla_y \cdot m\rangle\| \lesssim f_2(y)K_\infty g_2(x) + g_2(y)K_\infty f_2(x)$$
$$\|\nabla_\theta \nabla_\theta \nabla_\theta \text{Tr}[m(y)\nabla_y \nabla_x \cdot (m(x)K)]\| \lesssim g_2(y)K_\infty f_2(x) + f_2(y)K_\infty g_2(x)$$

Hence by Assumptions 1-4 $\|\nabla_\theta \nabla_\theta \nabla_\theta k^0_\theta\|$ is bounded above by a continuous integrable symmetric function and we can apply the MVT to show equicontinuity as in the proof above. Moreover the conditions of [70, Theorem 1] hold for the components of $\nabla_\theta \nabla_\theta k^0_\theta$, so that $\sup_{\theta \in \mathcal{N}}\big|\frac{1}{n(n-1)} \sum_{i \neq j} \partial_{\theta^a}\partial_{\theta^b} k^0_\theta(X_i, X_j) - \mathbb{Q} \otimes \mathbb{Q}\partial_{\theta^a}\partial_{\theta^b} k^0_\theta\big| \xrightarrow{a.s.} 0$ as $n \to \infty$, for all $a$ and $b$.

Finally we observe that $\mathbb{Q} \otimes \mathbb{Q} \partial_{\theta^a} \partial_{\theta^b} k_\theta^0 \big|_{\theta = \theta_*^{\mathrm{DKSD}}} = g_{ab}(\theta_*^{\mathrm{DKSD}})$, where $g$ is the information metric associated with $\mathrm{DKSD}_{K,m}$. Indeed using $\delta_{p,q} = 0$ if $p = q$

$$
\begin{aligned}
&\mathbb{Q} \otimes \mathbb{Q} \partial_{\theta^a} \partial_{\theta^b} k_\theta^0 \big|_{\theta = \theta_*^{\mathrm{DKSD}}} \\
&= \partial_{\theta^a} \partial_{\theta^b} \int_\mathcal{X} \int_\mathcal{X} p_{\theta_*^{\mathrm{DKSD}}}(x) \delta_{p_\theta, p_{\theta_*^{\mathrm{DKSD}}}}(x)^\top K(x,y) \delta_{p_\theta, p_{\theta_*^{\mathrm{DKSD}}}}(y) p_{\theta_*^{\mathrm{DKSD}}}(y) \mathrm{d}x \mathrm{d}y \big|_{\theta = \theta_*^{\mathrm{DKSD}}} \\
&= \partial_{\theta^a} \int_\mathcal{X} \int_\mathcal{X} p_{\theta_*^{\mathrm{DKSD}}}(x) \partial_{\theta^b} \delta_{p_\theta, p_{\theta_*^{\mathrm{DKSD}}}}(x)^\top K(x,y) \delta_{p_\theta, p_{\theta_*^{\mathrm{DKSD}}}}(y) p_{\theta_*^{\mathrm{DKSD}}}(y) \mathrm{d}x \mathrm{d}y \big|_{\theta = \theta_*^{\mathrm{DKSD}}} \\
&\quad + \partial_{\theta^a} \int_\mathcal{X} \int_\mathcal{X} p_{\theta_*^{\mathrm{DKSD}}}(x) \delta_{p_\theta, p_{\theta_*^{\mathrm{DKSD}}}}(x)^\top K(x,y) \partial_{\theta^b} \delta_{p_\theta, p_{\theta_*^{\mathrm{DKSD}}}}(y) p_{\theta_*^{\mathrm{DKSD}}}(y) \mathrm{d}x \mathrm{d}y \big|_{\theta = \theta_*^{\mathrm{DKSD}}} \\
&= \int_\mathcal{X} \int_\mathcal{X} p_{\theta_*^{\mathrm{DKSD}}}(x) \partial_{\theta^b} \delta_{p_\theta, p_{\theta_*^{\mathrm{DKSD}}}}(x)^\top K(x,y) \partial_{\theta^a} \delta_{p_\theta, p_{\theta_*^{\mathrm{DKSD}}}}(y) p_{\theta_*^{\mathrm{DKSD}}}(y) \mathrm{d}x \mathrm{d}y \big|_{\theta = \theta_*^{\mathrm{DKSD}}} \\
&\quad + \int_\mathcal{X} \int_\mathcal{X} p_{\theta_*^{\mathrm{DKSD}}}(x) \partial_{\theta^a} \delta_{p_\theta, p_{\theta_*^{\mathrm{DKSD}}}}(x)^\top K(x,y) \partial_{\theta^b} \delta_{p_\theta, p_{\theta_*^{\mathrm{DKSD}}}}(y) p_{\theta_*^{\mathrm{DKSD}}}(y) \mathrm{d}x \mathrm{d}y \big|_{\theta = \theta_*^{\mathrm{DKSD}}} \\
&= 2 \int_\mathcal{X} \int_\mathcal{X} \left( m_{\theta_*^{\mathrm{DKSD}}}^\top(x) \nabla_x \partial_{\theta^j \mathrm{DKSD}} \log p_{\theta_*^{\mathrm{DKSD}}} \right)^\top K(x,y) \\
&\quad \left( m_{\theta_*^{\mathrm{DKSD}}}^\top(y) \nabla_y \partial_{\theta^i \mathrm{DKSD}}^* \log p_{\theta_*^{\mathrm{DKSD}}} \right) \mathrm{d}\mathbb{P}_{\theta_*^{\mathrm{DKSD}}}(x) \mathrm{d}\mathbb{P}_{\theta_*^{\mathrm{DKSD}}}(y),
\end{aligned}
$$

so $\mathbb{Q} \otimes \mathbb{Q} \partial_{\theta^a} \partial_{\theta^b} k_\theta^0 \big|_{\theta = \theta_*^{\mathrm{DKSD}}} = g_{ab}(\theta_*^{\mathrm{DKSD}})$. The conditions of [56, Theorem 3.1] hold, from which the advertised result follows. ∎

## D.2 Diffusion Score Matching

Recall that the DSM is given by:

$$
\mathrm{DSM}(\mathbb{Q} \| \mathbb{P}_\theta) = \int_\mathcal{X} \left( \| m^\top \nabla_x \log p_\theta \|_2^2 + \| m^\top \nabla \log q \|_2^2 + 2 \nabla \cdot \left( m m^\top \nabla \log p_\theta \right) \right) \mathrm{d}\mathbb{Q}
$$

and we wish to estimate

$$
\theta_*^{\mathrm{DSM}} = \mathrm{argmin}_{\theta \in \Theta} \int_\mathcal{X} \left( \| m^\top \nabla_x \log p_\theta \|_2^2 + 2 \nabla \cdot \left( m m^\top \nabla \log p_\theta \right) \right) \mathrm{d}\mathbb{Q} \equiv \mathrm{argmin}_{\theta \in \Theta} \int_\mathcal{X} F_\theta \mathrm{d}\mathbb{Q}
$$

with a sequence of $M$-estimators $\hat{\theta}_n^{\mathrm{DSM}} = \mathrm{argmin}_{\theta \in \Theta} \frac{1}{n} \sum_i^n F_\theta(X_i)$. Recall also we have

$$
F_\theta(x) = \| m^\top \nabla_x \log p_\theta \|_2^2 + 2 \langle \nabla \cdot (m m^\top), \nabla \log p_\theta \rangle + 2 \mathrm{Tr} \left[ m m^\top \nabla^2 \log p_\theta \right].
$$

We will have a unique minimiser $\theta_*^{\mathrm{DSM}}$ whenever the map $\theta \mapsto \mathbb{P}_\theta$ is injective.

### D.2.1 Weak Consistency of DSM

**Theorem 11** (Weak Consistency of DSM). *Suppose $\mathcal{X}$ be open subset of $\mathbb{R}^d$, and $\Theta \subset \mathbb{R}^m$. Suppose $\log p_\theta(\cdot)$ is $C^2(\mathcal{X})$ and $m \in C^1(\mathcal{X})$, and $\| \nabla_x \log p_\theta(x) \| \leq f_1(x)$. Suppose also that $\| \nabla_x \nabla_x \log p_\theta(x) \| \leq f_2(x)$ on any compact set $C \subset \Theta$, where $\| m^\top \| f_1 \in L^2(\mathbb{Q})$, $\| \nabla \cdot (m m^\top) \| f_1 \in L^1(\mathbb{Q})$, $\| m m^\top \|_\infty f_2 \in L^1(\mathbb{Q})$. If either $\Theta$ is compact, or $\Theta$ and $\theta \mapsto F_\theta$ are convex and $\theta^* \in int(\Theta)$, then $\hat{\theta}_n^{\mathrm{DSM}}$ is weakly consistent for $\theta^*$.*

**Proof**    By assumption $\theta \mapsto F_\theta(x)$ is continuous. Suppose $\Theta$ is compact, taking $C = \Theta$, note

$$
\begin{aligned}
|F_\theta| &= \left| \| m^\top \nabla_x \log p_\theta \|_2^2 + 2 \nabla \cdot \left( m m^\top \nabla \log p_\theta \right) \right| \\
&= \left| \| m^\top \nabla_x \log p_\theta \|_2^2 + 2 \left( \nabla \cdot (m m^\top) \cdot \nabla \log p_\theta + \mathrm{Tr} \left[ m m^\top \nabla^2 \log p_\theta \right] \right) \right| \\
&\lesssim \| m^\top \|^2 f_1^2 + 2 \| \nabla \cdot (m m^\top) \| f_1 + 2 \| m m^\top \|_\infty f_2
\end{aligned}
$$

which is integrable, so the conditions of Lemma 2.4 [56] are satisfied so $\theta \mapsto \mathbb{Q} F_\theta$ is continuous, and $\sup_\Theta |\frac{1}{n} \sum_i^n F_\theta(X_i) - \mathbb{Q} F_\theta| \xrightarrow{p} 0$, and thus from theorem 2.1 [56] $\hat{\theta}_n^{\mathrm{DSM}} \xrightarrow{p} \theta_*^{\mathrm{DSM}}$. If $\Theta$ is convex, note that the sum of convex functions is convex, so $\theta \mapsto \frac{1}{n} \sum_i^n F_\theta(X_i)$ is convex, and we can follow a derivation analogous to the one in Theorem 3. ∎

### D.2.2 Asymptotic Normality of DSM

**Theorem 12** (**Asymptotic Normality of DSM**). *Suppose $\mathcal{X}, \Theta$ be open subsets of $\mathbb{R}^d$ and $\mathbb{R}^m$ respectively. If (i) $\hat{\theta}_n^{\mathrm{DSM}} \xrightarrow{p} \theta^*$, (ii) $\theta \mapsto \log p_\theta(x)$ is twice continuously differentiable on a closed ball $\bar{B}(\epsilon, \theta^*) \subset \Theta$, and*

*(iii)* $\|mm^\top\| + \|\nabla_x \cdot (mm^\top)\| \leq f_1(x)$, *and* $\|\nabla_x \log p\| + \|\nabla_{\theta^*}\nabla_x \log p\| + \|\nabla_{\theta^*}\nabla_x\nabla_x \log p\| \leq f_2(x)$, *with* $f_1 f_2, f_1 f_2^2 \in L^2(\mathbb{Q})$

*(iv)* *for* $\theta \in \bar{B}(\epsilon, \theta^*)$ $\|\nabla_\theta\nabla_x \log p\|^2 + \|\nabla_x \log p\|\|\nabla_\theta\nabla_\theta\nabla_x \log p\| + \|\nabla_\theta\nabla_\theta\nabla_x \log p\| + \|\nabla_\theta\nabla_\theta\nabla_x\nabla_x \log p\| \leq g_1(x)$, *and* $f_1 g_1 \in L^1(\mathbb{Q})$,

*and (v) and the information tensor is invertible at $\theta^*$. Then*

$$\sqrt{n}\left(\hat{\theta}_n^{\mathrm{DSM}} - \theta^*\right) \xrightarrow{d} \mathcal{N}\left(0, g^{-1}(\theta^*)\mathbb{Q}[\nabla_{\theta^*}F_\theta \otimes \nabla_{\theta^*}F_\theta]g^{-1}(\theta^*)\right)$$

**Proof**  From (ii) $\theta \mapsto F_\theta$ is twice continuously differentiable on a ball $B(\epsilon, \theta^*) \subset \Theta$. Note $\nabla_\theta \frac{1}{N}\sum_i^N F_\theta(X_i) = \frac{1}{N}\sum_i^N \nabla_\theta F_\theta(X_i)$, then $\mathbb{Q}[\nabla_\theta F_{\theta_*^{\mathrm{DSM}}}(X_i)] = \nabla_\theta \mathbb{Q}[F_{\theta_*^{\mathrm{DSM}}}(X_i)] = 0$. Note

$$\|\nabla_\theta F_{\theta_*^{\mathrm{DSM}}}(x)\| \lesssim \|mm^\top\|\|\nabla_x \log p\|\|\nabla_\theta\nabla_x \log p\| + \|\nabla_x \cdot (mm^\top)\|\|\nabla_\theta\nabla_x \log p\|$$
$$+ \|mm^\top\|\|\nabla_\theta\nabla_x\nabla_x \log p\|$$
$$\lesssim f_1(x)f_2(x)[f_2(x) + 2].$$

Hence $\nabla_\theta F_{\theta_*^{\mathrm{DSM}}} \in L^2(\mathbb{Q})$, so by the CLT

$$\sqrt{n}\nabla_\theta \frac{1}{n}\sum_i^n F_{\theta_*^{\mathrm{DSM}}}(X_i) \xrightarrow{d} \mathcal{N}\left(0, \mathbb{Q}[\nabla_\theta F_{\theta_*^{\mathrm{DSM}}} \otimes \nabla_\theta F_{\theta_*^{\mathrm{DSM}}}]\right).$$

Now $\theta \mapsto \nabla_\theta\nabla_\theta F_\theta(x)$ is continuous on $\overline{B}(\epsilon, \theta^*)$ so we have:

$$\|\nabla_\theta\nabla_\theta F_\theta(x)\| \lesssim \|mm^\top\|\left(\|\nabla_\theta\nabla_x \log p\|^2 + \|\nabla_x \log p\|\|\nabla_\theta\nabla_\theta\nabla_x \log p\|\right)$$
$$+ \|\nabla \cdot (mm^\top)\|\|\nabla_\theta\nabla_\theta\nabla_x \log p\| + \|mm^\top\|\|\nabla_\theta\nabla_\theta\nabla_x\nabla_x \log p\|$$
$$\lesssim f_1(x)g_1(x)$$

Combining the above, we have that the assumptions of Lemma 2.4 [56] applied to $\overline{B}(\epsilon, \theta^*)$ hold, and $\sup_{\overline{B}(\epsilon,\theta^*)}\left|\frac{1}{n}\sum_i^n \partial_{\theta^a}\partial_{\theta^b}F_\theta|_{\theta^*}(X_i) - \mathbb{Q}\partial_{\theta^a}\partial_{\theta^b}F_\theta|_{\theta^*}\right| \xrightarrow{p} 0$. As in Theorem 4 $\mathbb{Q}\partial_{\theta^a}\partial_{\theta^b}F_\theta|_{\theta^*} = g_{ab}(\theta^*)$ is the information tensor, which is continuous at $\theta^*$ by Lemma 2.4. The result follows by theorem 3.1 [56]. ∎

### D.3 Strong Consistency and Central Limit Theorems for Exponential Families

Let $\mathcal{X}$ be an open subset of $\mathbb{R}^d$, $\Theta \subset \mathbb{R}^m$. Consider the case when the density $p$ lies in an exponential family, i.e. $p_\theta(x) \propto \exp(\langle\theta, T(x)\rangle_{\mathbb{R}^m} - c(\theta))\exp(b(x))$, where $\theta \in \mathbb{R}^m$ and sufficient statistic $T = (T_1, \ldots, T_m) : \mathcal{X} \to \mathbb{R}^m$. Then $\nabla T \in \Gamma(\mathcal{X}, \mathbb{R}^{m \times d})$ and $\nabla_x \log p_\theta = \nabla_x b + \theta \cdot \nabla_x T$, $\nabla_\theta \nabla_x \log p_\theta = \nabla_x T^\top$.

### D.3.1 Strong Consistency of the Minimum Diffusion Kernel Stein Discrepancy Estimator

We consider a RKHS $\mathcal{H}^d$ of functions $f : \mathcal{X} \to \mathbb{R}^d$ with matrix kernel $K$. Recall the Stein kernel is

$$k^0 = \nabla_x \log p \cdot m(x)Km(y)^\top\nabla_y \log p + \nabla_x \cdot (m(x)K) \cdot \nabla_y \cdot m + \mathrm{Tr}[m(y)\nabla_y\nabla_x \cdot (m(x)K)]$$
$$+ \nabla_y \cdot (m(y)K) \cdot m(x)^\top\nabla_x \log p + \nabla_x \cdot (m(x)K) \cdot m(y)^\top\nabla_y \log p$$

Given a (i.i.d.) sample $X_i \sim \mathbb{Q}$, we can define an estimator using the $U$-statistic

$$\widehat{\mathrm{DKSD}}_{K,m}(\{X_i\}_{i=1}^n, \mathbb{P}_\theta)^2 = \frac{2}{n(n-1)}\sum_{1 \leq i < j \leq n} k^0(X_i, X_j).$$

For the case where the density $p$ lies in an exponential family, then $k^0 = \theta^\top A\theta + v^\top\theta + c$ where $A \in \Gamma(\mathcal{X} \times \mathcal{X}, \mathbb{R}^{m \times m})$, $v \in \Gamma(\mathcal{X} \times \mathcal{X}, \mathbb{R}^m)$ are given by (we set $\phi \equiv m^\top \nabla T^\top \in \Gamma(\mathcal{X}, \mathbb{R}^{d \times m})$)

$$A = \phi(x)^\top K(x,y)\phi(y)$$
$$v^\top = \nabla_y b \cdot m(y)K(y,x)\phi(x) + \nabla_x b \cdot m(x)K(x,y)\phi(y)$$
$$+ \nabla_x \cdot (m(x)K) \cdot \phi(y) + \nabla_y \cdot (m(y)K) \cdot \phi(x)$$
$$c = \nabla_x b \cdot m(x)K(x,y)m(y)^\top \nabla_y b + \nabla_x \cdot (m(x)K) \cdot \nabla_y \cdot m + \text{Tr}[m(y)\nabla_y \nabla_x \cdot (m(x)K)]$$
$$+ \nabla_y \cdot (m(y)K) \cdot m(x)^\top \nabla_x b + \nabla_x \cdot (m(x)K) \cdot m(y)^\top \nabla_y b$$

**Lemma 3.** *Suppose $K$ is IPD, that $\nabla T$ has linearly independent rows, that $m$ is invertible, and $\|\phi\|_{L^1(\mathbb{Q})} < \infty$. Then the matrix $\int_{\mathcal{X}} A\mathbb{Q} \otimes \mathbb{Q}$ is symmetric positive definite.*

**Proof** The matrix $B = \int_{\mathcal{X}} A\mathbb{Q} \otimes \mathbb{Q}$ is symmetric

$$(\textstyle\int_{\mathcal{X}} A\mathbb{Q} \otimes \mathbb{Q})^\top = \int_{\mathcal{X}} A(x,y)^\top \mathbb{Q}(\mathrm{d}x) \otimes \mathbb{Q}(\mathrm{d}y) = \int_{\mathcal{X}} \nabla_y Tm(y)K(x,y)^\top m(x)^\top \nabla_x T^\top \mathbb{Q}(\mathrm{d}x) \otimes \mathbb{Q}(\mathrm{d}y)$$
$$= \int_{\mathcal{X}} \nabla_y Tm(y)K(y,x)m(x)^\top \nabla_x T^\top \mathbb{Q}(\mathrm{d}y) \otimes \mathbb{Q}(\mathrm{d}x) = \int_{\mathcal{X}} A\mathbb{Q} \otimes \mathbb{Q}.$$

Moreover, set $\phi \equiv m^\top \nabla T^\top$, so $A(x,y) = \phi(x)^\top K(x,y)\phi(y)$. If $v \neq 0$, then $u \equiv \phi v \neq 0$ as $\nabla T^\top$ has full column rank (i.e., the vectors $\{\nabla T_i\}$ are linearly independent) and $m$ is invertible, and $\|\phi v\|_{L^1(\mathbb{Q})} = \int_{\mathcal{X}} \|\phi(x)v\|_1 \mathrm{d}x \leq \|v\|_1 \int_{\mathcal{X}} \|\phi(x)\|_1 \mathrm{d}x < \infty$ implies $\mathrm{d}\mu_i \equiv u_i\mathrm{d}\mathbb{Q}$ is a finite signed Borel measure for each $i$. Clearly

$$v^\top(\textstyle\int_{\mathcal{X}} A\mathbb{Q} \otimes \mathbb{Q})v = \int_{\mathcal{X}} u(x)^\top K(x,y)u(y)\mathbb{Q}(\mathrm{d}x)\mathbb{Q}(\mathrm{d}y)$$
$$= \int_{\mathcal{X}} K(x,y)_{ij}u_i(x)u_j(y)\mathbb{Q}(\mathrm{d}x)\mathbb{Q}(\mathrm{d}y)$$
$$= \int_{\mathcal{X}} K(x,y)_{ij}\mu_i(\mathrm{d}x)\mu_j(\mathrm{d}y) \geq 0.$$

Moreover since the kernel is IPD, if this equals zero then for all $i$: $0 = \mu_i(C) = u_i\mathbb{Q}(C) = \phi_{ij}v_j\mathbb{Q}(C)$ for all measurable sets $C$, which implies $\phi v = 0$ and thus $v = 0$. ∎

**Theorem 1.** *Suppose $K$ is IPD with bounded derivative up to order $2$, that $\nabla T$ has linearly independent rows, and $m$ is invertible. Suppose $\|\phi\|, \|\nabla_x b\|\|m\|, \|\nabla_x m\| + \|m\| \in L^1(\mathbb{Q})$. The minimiser $\hat{\theta}_n^{\mathrm{DKSD}}$ of $\widehat{\mathrm{DKSD}}_{K,m}(\{X_i\}_{i=1}^n, \mathbb{P}_\theta)$ exists eventually, and converges almost surely to the minimiser $\theta^*$ of $\mathrm{DKSD}_{K,m}(\mathbb{Q}, \mathbb{P}_\theta)$.*

**Proof**

Let $X_i : \Omega \to \mathcal{X} \subset \mathbb{R}^d$ be independent $\mathbb{Q}$-distributed random vectors. The $U$-statistic $A_n \equiv \frac{2}{n(n-1)} \sum_{1 \leq i < j \leq n} A(X_i, X_j)$ is symmetric semi-definite. Since $\int_{\mathcal{X}} \|A\|\mathrm{d}\mathbb{Q} \otimes \mathbb{Q} < \infty$, by theorem 1 [30] the components of $A_n$ converge to the components of $B$ almost surely, and since the matrix inverse is a continuous map, by the continuous mapping theorem the components of $A_n^{-1}$ (the inverse exists eventually) converge almost surely to $B^{-1}$. Hence the minimiser of $\widehat{\mathrm{DKSD}}_{K,m}(\{X_i\}_{i=1}^n, \mathbb{P}_\theta)^2 = \theta^\top A_n\theta + v_n^\top\theta + c$ where $v_n \equiv \frac{2}{n(n-1)} \sum_{1 \leq i < j \leq n} v(X_i, X_j)$ exists eventually.

$$|A(x,y)| \lesssim K_\infty \|\phi(x)\|\|\phi(y)\|$$
$$\|v\| \lesssim K_\infty \|\nabla_y b\|\|m(y)\|\|\phi(x)\| + K_\infty \|\nabla_x b\|\|m(x)\|\|\phi(y)\|$$
$$+ (\|\nabla_x m\| + \|m(x)\|)K_\infty\|\phi(y)\| + (\|\nabla_y m\| + \|m(y)\|)K_\infty\|\phi(x)\|$$
$$|c| \lesssim K_\infty \|\nabla_x b\|\|m(x)\|\|m(y)\|\|\nabla_y b\| + K_\infty(\|\nabla_x m\| + \|m(x)\|)\|\nabla_y m\| +$$
$$+ K_\infty \|m(y)\|(1 + \|m(x)\| + \|\nabla_x m\|)$$
$$+ K_\infty(\|\nabla_y m\| + \|m(y)\|)\|\nabla_x m\|\|\nabla_x b\| + K_\infty(\|\nabla_x m\| + \|m(x)\|)\|\nabla_y m\|\|\nabla_y b\|$$

and it follows from the integrability assumptions that $\mathbb{Q} \otimes \mathbb{Q}|k_\theta^0| < \infty$. Since the product and sum of random variables that converge a.s. converge a.s., we have that $\hat{\theta}_n^{\mathrm{DKSD}} \to \theta^*$ a.s.,

$$\hat{\theta}_n^{\mathrm{DKSD}} = -\tfrac{1}{2}A_n^{-1}v_n \xrightarrow{a.s.} -\tfrac{1}{2}B^{-1}v = \theta^*.$$

∎

### D.3.2 Asymptotic Normality of the DKSD Estimator

We now consider the distribution of $\sqrt{n}(\hat{\theta}_n^{\mathrm{DKSD}} - \theta^*)$. Recall that $A \in \Gamma(\mathcal{X}, \mathbb{R}^{m \times m}), v \in \Gamma(\mathcal{X}, \mathbb{R}^m)$, and for $n$ large enough $A_n^{-1}$ exists a.s., and $\hat{\theta}_n^{\mathrm{DKSD}} = -\frac{1}{2} A_n^{-1} v_n$.

**Theorem 2.** *Suppose* $\|\phi\|, \|\nabla_x b\| \|m\|, \|\nabla_x m\| + \|m\| \in L^2(\mathbb{Q})$. *Then the* DKSD *estimator is asymptotically normal.*

**Proof** From the integrability assumptions, it follows that $v, A \in L^2(\mathbb{Q} \otimes \mathbb{Q})$, and since $\mathcal{X}$ has finite $\mathbb{Q} \otimes \mathbb{Q}$-measure, $v, A \in L^1(\mathbb{Q} \otimes \mathbb{Q})$. Assume first that $m = 1$. Hence the tuple $U_n \equiv (v_n, A_n) : \Omega \to \mathbb{R}^2$, with $\mathbb{E}[U_n] = (\int_{\mathcal{X}} v \mathbb{Q} \otimes \mathbb{Q}, \int_{\mathcal{X}} A \mathbb{Q} \otimes \mathbb{Q}) \equiv (U_1, U_2)$, is asymptotically normal

$$\sqrt{n}(U_n - \mathbb{E}[U_n]) \xrightarrow{d} \mathcal{N}(0, 4\Sigma)$$

where, setting $v^0 = v - U_1$ and $A^0 = A - U_2$

$$\Sigma = \mathbb{E}\big[\big(\int_{\mathcal{X}} v^0(X, y) \mathrm{d}\mathbb{Q}(y), \int_{\mathcal{X}} A^0(X, y) \mathrm{d}\mathbb{Q}(y)\big) \otimes \big(\int_{\mathcal{X}} v^0(X, y) \mathrm{d}\mathbb{Q}(y), \int_{\mathcal{X}} A^0(X, y) \mathrm{d}\mathbb{Q}(y)\big)\big]$$

$$= \begin{pmatrix} \int_{\mathcal{X}} v^0(x,y) \mathrm{d}\mathbb{Q}(y) \int_{\mathcal{X}} v^0(x,z) \mathrm{d}\mathbb{Q}(z) \mathrm{d}\mathbb{Q}(x) & \int_{\mathcal{X}} v^0(x,y) \mathrm{d}\mathbb{Q}(y) \int_{\mathcal{X}} A^0(x,z) \mathrm{d}\mathbb{Q}(z) \mathrm{d}\mathbb{Q}(x) \\ \int_{\mathcal{X}} v^0(x,y) \mathrm{d}\mathbb{Q}(y) \int_{\mathcal{X}} v^0(x,z) \mathrm{d}\mathbb{Q}(z) \mathrm{d}\mathbb{Q}(x) & \int_{\mathcal{X}} A^0(x,y) \mathrm{d}\mathbb{Q}(y) \int_{\mathcal{X}} A^0(x,z) \mathrm{d}\mathbb{Q}(z) \mathrm{d}\mathbb{Q}(x) \end{pmatrix}$$

Since $\hat{\theta}_n^{\mathrm{DKSD}} = g(U_n), \theta^* = g(U)$ where $g(x, y) \equiv -\frac{1}{2} x/y$, we can apply the delta method which states

$$\sqrt{n}(\hat{\theta}_n^{\mathrm{DKSD}} - \theta^*) = \sqrt{n}(g(U_n) - g(U)) \xrightarrow{d} \mathcal{N}\big(0, 4\nabla g(U) \Sigma \nabla g(U)^\top\big)$$

and $\nabla g(U) = \big(-1/2U_2, U_1/2U_2^2\big)$. Now let $m$ be arbitrary. Since $A \in L^2(\mathbb{Q})$ then setting $A^0 \equiv A - \int_{\mathcal{X}} A\mathbb{Q} \otimes \mathbb{Q}$ we find

$$\sqrt{n}(A_n - \mathbb{E}[A_n]) \xrightarrow{d} \mathcal{N}(0, 4\Sigma_1), \qquad \Sigma_1 \equiv \int_{\mathcal{X}} \big[\int_{\mathcal{X}} A^0(x,y) \mathrm{d}\mathbb{Q}(y) \otimes \int_{\mathcal{X}} A^0(x,y) \mathrm{d}\mathbb{Q}(y)\big] \mathrm{d}\mathbb{Q}(x)$$

and similarly, with $v^0 \equiv v - \int v \mathrm{d}\mathbb{Q} \otimes \mathrm{d}\mathbb{Q}$

$$\sqrt{n}(v_n - \mathbb{E}[v_n]) \xrightarrow{d} \mathcal{N}(0, 4\Sigma_2), \qquad \Sigma_2 \equiv \int_{\mathcal{X}} \big[\int_{\mathcal{X}} v^0(x,y) \mathrm{d}\mathbb{Q}(y) \otimes \int_{\mathcal{X}} v^0(x,y) \mathrm{d}\mathbb{Q}(y)\big] \mathrm{d}\mathbb{Q}(x).$$

and

$$\sqrt{n}((v_n, A_n) - \mathbb{E}[(v_n, A_n)]) \xrightarrow{d} \mathcal{N}(0, 4\Sigma)$$

where

$$\Sigma = \int_{\mathcal{X}} \big[\big(\int_{\mathcal{X}} v^0(x,y) \mathrm{d}\mathbb{Q}(y), \int_{\mathcal{X}} A^0(x,y) \mathrm{d}\mathbb{Q}(y)\big) \otimes \big(\int_{\mathcal{X}} v^0(x,y) \mathrm{d}\mathbb{Q}(y), \int_{\mathcal{X}} A^0(x,y) \mathrm{d}\mathbb{Q}(y)\big)\big] \mathrm{d}\mathbb{Q}(x).$$

Let $\mathcal{D} \equiv \mathbb{R}^m \times \mathbb{R}^{m \times m}$, which we equip with coordinates $z_{ijk} = (x_i, y_{jk})$. Consider the function $g : \mathcal{D} \to \mathbb{R}^m, (x, y) \mapsto -\frac{1}{2} y^{-1} x$, so $g(v_n, A_n) = \theta_n^{\mathrm{DKSD}}$. Note $\Sigma \in \mathcal{D} \times \mathcal{D}$ and $\nabla g : \mathcal{D} \to \mathrm{End}(\mathcal{D}, \mathbb{R}^m) \cong \mathbb{R}^m \times \mathcal{D}$, so that $\nabla g(U) \Sigma \nabla g(U)^\top \in \mathbb{R}^{m \times m}$. First consider the matrix inversion $h(y) = y^{-1}$, so $\nabla h(y) \in \mathbb{R}^{(m \times m) \times (m \times m)}$, and $\nabla h(y)_{(ij)(kr)} = \partial_{y^{kr}} h_{ij}$. Since $h(y)_{ij} y_{jl} = \delta_{il}$ we have $0 = \partial_{kr}(h(y)_{ij} y_{jl}) = \partial_{kr}(h(y)_{ij}) y_{jl} + h(y)_{ij} \delta_{jk} \delta_{rl} = \partial_{kr}(h(y)_{ij}) y_{jl} + h(y)_{ik} \delta_{rl}$ and

$$\nabla h(y)_{(is)(kr)} = \partial_{kr}(h(y)_{ij}) y_{jl} h(y)_{ls} = -h_{ik} \delta_{rl} h(y)_{ls} = -h(y)_{ik} h(y)_{rs}$$

and clearly $f : x \mapsto x$, then $\nabla f(x) = 1_{m \times m}$. Moreover

$$\partial_{y^{ab}} g_i(z) = \partial_{y^{ab}}(h(y)_{ij} f(x)_j) = \partial_{y^{ab}}(h(y)_{ij}) x_j = -h(y)_{ia} h(y)_{bj} x_j, \quad \partial_{x^l} g_i(z) = h(y)_{il}$$

Then

$$(\nabla g(z) \Sigma)_{ir} = \partial_v g_i \Sigma_{vr} = g_{i,x^l} \Sigma_{x^l r} + g_{i,y^{ab}} \Sigma_{y^{ab} r} = h(y)_{il} \Sigma_{x^l r} + \partial_{y^{ab}}(h(y)_{is}) x_s \Sigma_{y^{ab} r}$$
$$= h(y)_{il} \Sigma_{x^l r} - h(y)_{ia} h(y)_{bs} x_s \Sigma_{y^{ab} r},$$

so

$$(\nabla g(z) \Sigma \nabla g(z)^\top)_{ic} = (\nabla g(z) \Sigma)_{ir} (\nabla g(z))_{cr} = (\nabla g(z) \Sigma)_{ir} \partial_r g_c$$
$$= h(y)_{il} \Sigma_{x^l r} \partial_r g_c - h(y)_{ia} h(y)_{bs} x_s \Sigma_{y^{ab} r} \partial_r g_c$$

with

$$h(y)_{il}\Sigma_{x^l r}\partial_r g_c = h(y)_{il}\Sigma_{x^l x^b}\partial_{x^b} g_c + h(y)_{il}\Sigma_{x^l y^{as}}\partial_{y^{as}} g_c$$
$$= h(y)_{il}\Sigma_{x^l x^b}h(y)_{cb} - h(y)_{il}\Sigma_{x^l y^{as}}h_{ca}(y)h(y)_{sj}x_j$$

and

$$-h(y)_{ia}h(y)_{bs}x_s\Sigma_{y^{ab} r}\partial_r g_c = -h(y)_{ia}h(y)_{bs}x_s\big(\Sigma_{y^{ab}x^k}\partial_{x^k} g_c + \Sigma_{y^{ab}y^{ld}}\partial_{y^{ld}} g_c\big)$$
$$= -h(y)_{ia}h(y)_{bs}x_s\big(\Sigma_{y^{ab}x^k}h(y)_{ck} - \Sigma_{y^{ab}y^{ld}}h(y)_{cl}h(y)_{dj}x_j\big).$$

Note we have

$$\Sigma_{xx} = \int_{\mathcal{X}}\int_{\mathcal{X}} v^0(x,y)\mathrm{d}\mathbb{Q}(y) \otimes \int_{\mathcal{X}} v^0(x,z)\mathrm{d}\mathbb{Q}(z)\mathrm{d}\mathbb{Q}(x) \equiv \int_{\mathcal{X}} T(x) \otimes T(x)\mathrm{d}\mathbb{Q}(x)$$
$$\Sigma_{xy} = \int_{\mathcal{X}}\int_{\mathcal{X}} v^0(x,y)\mathrm{d}\mathbb{Q}(y) \otimes \int_{\mathcal{X}} A^0(x,z)\mathrm{d}\mathbb{Q}(z)\mathrm{d}\mathbb{Q}(x) \equiv \int_{\mathcal{X}} T(x) \otimes L(x)\mathrm{d}\mathbb{Q}(x)$$
$$\Sigma_{yy} = \int_{\mathcal{X}}\int_{\mathcal{X}} A^0(x,y)\mathrm{d}\mathbb{Q}(y) \otimes \int_{\mathcal{X}} A^0(x,z)\mathrm{d}\mathbb{Q}(z)\mathrm{d}\mathbb{Q}(x) \equiv \int_{\mathcal{X}} L(x) \otimes L(x)\mathrm{d}\mathbb{Q}(x)$$

then

$$4\nabla g(U_1,U_2)\Sigma\nabla g(U_1,U_2)^\top = \int_{\mathcal{X}}(U_2^{-1}T) \otimes (TU_2^{-1})\mathrm{d}\mathbb{Q}$$
$$- 2\int_{\mathcal{X}}\big(U_2^{-1}LU_2^{-1}U_1\big) \otimes \big(TU_2^{-1}\big)\mathrm{d}\mathbb{Q}$$
$$+ \int_{\mathcal{X}}\big(U_2^{-1}LU_2^{-1}U_1\big) \otimes \big(U_2^{-1}LU_2^{-1}U_1\big)\mathrm{d}\mathbb{Q}$$

∎

### D.3.3  Diffusion Score Matching Asymptotics

Consider the loss function

$$L(x,\theta) = \big\langle \nabla \log p_\theta, mm^\top \nabla \log p_\theta\big\rangle + 2\big(\nabla \cdot (mm^\top) \cdot \nabla \log p_\theta + \mathrm{Tr}\big[mm^\top \nabla^2 \log p_\theta\big]\big).$$

For the exponential family $L(x,\theta) = \theta^\top A\theta + v^\top\theta + c$, where (we set $S = mm^\top$)

$$A = \nabla T S \nabla T^\top$$
$$v^\top = 2\nabla b \cdot S\nabla T^\top + 2\nabla \cdot S \cdot \nabla T^\top + 2\mathrm{Tr}\big[S\nabla^2 T_i\big]e_i$$
$$c = \nabla b \cdot S\nabla b + 2\nabla \cdot S \cdot \nabla b + 2\mathrm{Tr}[S\nabla\nabla b].$$

**Theorem 13.** *Suppose $m$ is invertible and $\{\nabla T_i\}$ are linearly independent. Then if $A, v \in L^1(\mathbb{Q})$, $\hat{\theta}_n^{\mathrm{DSM}}$ eventually exists and is strongly consistent. If we also have $A, v \in L^2(\mathbb{Q})$, then $\hat{\theta}_n^{\mathrm{DSM}}$ is asymptotically normal.*

**Proof**    Let $\mathcal{M} \equiv \int A\mathrm{d}\mathbb{Q}$, $H \equiv \int v\mathrm{d}\mathbb{Q}$. If $A = \nabla T mm^\top \nabla T^\top = \nabla Tm(\nabla Tm)^\top$ so $\mathrm{rank}(A) = \mathrm{rank}(\nabla Tm(\nabla Tm)^\top) = \mathrm{rank}(\nabla Tm) = \mathrm{rank}(\nabla T) = \mathrm{rank}(\nabla T^\top)$ if $m$ is invertible. So if the vectors $\{\nabla T_i\}$ are linearly independent, then $\nabla T^\top$ has full column rank. Then $A$ it is symmetric positive (strictly) definite and the minimum of $L(\theta) \equiv \int L(x,\theta)\mathrm{d}\mathbb{Q}(x)$ is $\theta^* = -\frac{1}{2}\mathcal{M}^{-1}H$ which for sufficiently large $n$ can be estimated by the random variable $\hat{\theta}_n^{\mathrm{DSM}} \equiv -\frac{1}{2}\mathcal{M}_n^{-1}H_n$ which converges a.s. to $\theta$.

We consider the tuple $U_n \equiv (H_n, \mathcal{M}_n)$, so $\mathbb{E}[U_n] = (H, \mathcal{M})$. Since $A, v \in L^2(\mathbb{Q})$, then

$$\sqrt{n}(U_n - (H, \mathcal{M})) \xrightarrow{d} \mathcal{N}(0, \Gamma)$$

where, setting $v^0 = v - H$, $A^0 = A - \mathcal{M}$

$$\Gamma = \mathbb{E}\big[(v^0, A^0) \otimes (v^0, A^0)\big].$$

Let $\mathcal{D} \equiv \mathbb{R}^m \times \mathbb{R}^{m \times m}$, and consider $g : \mathcal{D} \to \mathbb{R}^m$, defined by $g(x,y) = -\frac{1}{2}y^{-1}x$. Using the Delta method

$$\sqrt{n}(\hat{\theta}_n^{\mathrm{DSM}} - \theta^*) \xrightarrow{d} \mathcal{N}\big(0, 4\nabla g(H, \mathcal{M})\Gamma\nabla g(H, \mathcal{M})^\top\big)$$

where, proceeding as in Appendix D.3.2, we find

$$4\nabla g(H, \mathcal{M})\Gamma\nabla g(H, \mathcal{M})^\top = \int_{\mathcal{X}}(\mathcal{M}^{-1}v^0) \otimes (v^0\mathcal{M}^{-1})\mathrm{d}\mathbb{Q}$$
$$- 2\int_{\mathcal{X}}\big(\mathcal{M}^{-1}A^0\mathcal{M}^{-1}H\big) \otimes \big(v^0\mathcal{M}^{-1}\big)\mathrm{d}\mathbb{Q}$$
$$+ \int_{\mathcal{X}}\big(\mathcal{M}^{-1}A^0\mathcal{M}^{-1}H\big) \otimes \big(\mathcal{M}^{-1}A^0\mathcal{M}^{-1}H\big)\mathrm{d}\mathbb{Q}$$

∎

# E Proofs of Robustness of Minimum Stein Discrepancy Estimators

In this section, we provide conditions on the Stein operator (and Stein class) to obtain robust estimators in the context of DKSD and DSM. In particular we prove Proposition 7 and derive the influence function of DSM.

## E.1 Robustness of Diffusion Kernel Stein Discrepancy

Let $T : \mathcal{P}_\Theta \to \Theta$ with $T(\mathbb{P}) = \operatorname{argmin}_\Theta \operatorname{DKSD}_{K,m}(\mathbb{P}\|\mathbb{P}_\theta)$ be defined by $\operatorname{IF}(z, \mathbb{Q}) \equiv \lim_{t\to 0}(T(\mathbb{Q} + t(\delta_z - \mathbb{Q})) - T(\mathbb{Q}))/t$. Denote $\mathbb{Q}_t = \mathbb{Q} + t(\delta_z - \mathbb{Q})$, $\theta_t = T(\mathbb{Q}_t)$, $\theta_0 = T(\mathbb{Q})$. Note that by the first order optimality condition:

$$\nabla_\theta \int_\mathcal{X} \int_\mathcal{X} k^0 \mathbb{Q}_t \otimes \mathbb{Q}_t|_{\theta_t} = \nabla_{\theta_t} \operatorname{DKSD}_{K,m}(\mathbb{Q}_t\|\mathbb{P}_\theta) = 0.$$

By the MVT, there exists $\bar\theta$ on the line joining $\theta_0$ and $\theta_t$ for which

$$0 = \int_\mathcal{X} \int_\mathcal{X} \nabla_\theta k^0|_{\theta_0} \mathbb{Q}_t \otimes \mathbb{Q}_t + \int_\mathcal{X} \int_\mathcal{X} \nabla_\theta \nabla_\theta k^0|_{\bar\theta} \mathbb{Q}_t \otimes \mathbb{Q}_t (\theta_t - \theta_0).$$

Expanding

$$\mathbb{Q}_t \otimes \mathbb{Q}_t \nabla_\theta k^0|_{\theta_0} = t^2(\delta_z - \mathbb{Q}) \otimes (\delta_z - \mathbb{Q})\nabla_\theta k^0|_{\theta_0} + 2t\mathbb{Q}_y \nabla_\theta k^0|_{\theta_0}(z,y)$$

where we have used the optimality condition. On the other hand

$$\mathbb{Q}_t \otimes \mathbb{Q}_t \nabla_\theta \nabla_\theta k^0|_{\bar\theta} = (1 - 2t)\mathbb{Q} \otimes \mathbb{Q}\nabla_\theta \nabla_\theta k^0|_{\bar\theta} + t^2(\delta_z - \mathbb{Q}) \otimes (\delta_z - \mathbb{Q})\nabla_\theta \nabla_\theta k^0|_{\bar\theta} + 2t\mathbb{Q}_y \nabla_\theta \nabla_\theta k^0|_{\bar\theta}(z,y).$$

Hence

$$\mathbb{Q}_y \nabla_\theta k^0|_{\theta_0}(z,y) = \tfrac{1}{2}\big((1 - 2t)\mathbb{Q} \otimes \mathbb{Q}\nabla_\theta \nabla_\theta k^0|_{\bar\theta} + 2t\mathbb{Q}_y \nabla_\theta \nabla_\theta k^0|_{\bar\theta}(z,y)\big)\tfrac{\theta_t - \theta_0}{t} + O(t),$$

and taking the limit $t \to 0$, $\bar\theta \to \theta_0$ and using a derivation as in the proof of Theorem 4

$$\mathbb{Q}_y \nabla_\theta k^0|_{\theta_0}(z,y) = \tfrac{1}{2}\int_\mathcal{X}\int_\mathcal{X} \nabla_\theta \nabla_\theta k^0|_{\theta_0} \mathrm{d}\mathbb{Q} \otimes \mathrm{d}\mathbb{Q}\,\operatorname{IF}(z,\mathbb{Q}) = g(\theta_0)\operatorname{IF}(z,\mathbb{Q})$$

hence the influence function is given by

$$\operatorname{IF}(z,\mathbb{Q}) = g(\theta_0)^{-1}\int_\mathcal{X} \nabla_\theta k^0|_{\theta_0}(z,y)\mathrm{d}\mathbb{Q}(y).$$

We aim to show the estimator is $B$-robust, that is $z \mapsto \|\operatorname{IF}(z,\mathbb{Q})\|$ is bounded. Suppose that the additional assumptions hold. Then there exists a function $c$ such that $\int \langle s_p(x), K(x,y)\nabla_{\theta_0} s_p(y)\rangle \mathbb{Q}(dy) \le \|s_p(x)\|c(x)$ which is bounded in $x \in \mathcal{X}$. Following a similar argument, and using the assumptions, a similar limit will hold for all terms in $\int \nabla_{\theta_0} k^0(z,y)\mathrm{d}\mathbb{Q}(y)$. It follows that $\sup_{z\in\mathcal{X}}\|\operatorname{IF}(z,\mathbb{Q})\| < \infty$.

## E.2 Robustness of Diffusion Score Matching

The scoring rule $S : \mathcal{X} \times \mathcal{P}_\mathcal{X} \to \mathbb{R}$ of DSM is

$$S(x,\mathbb{P}_\theta) \equiv \tfrac{1}{2}\big\|m^\top \nabla_x \log p_\theta\big\|_2^2 + \nabla \cdot \big(mm^\top \nabla \log p_\theta\big)(x)$$

Indeed the proof of Theorem 2 we have

$$\int_\mathcal{X}\big\|m^\top \nabla \log q\big\|^2 \mathrm{d}\mathbb{Q} = -\int_\mathcal{X} \nabla \cdot \big(mm^\top \nabla \log q\big)\mathrm{d}\mathbb{Q}.$$

which implies $\mathbb{Q}S(\cdot,\mathbb{Q}) = -\tfrac{1}{2}\int_\mathcal{X}\big\|m^\top \nabla \log q\big\|^2 \mathrm{d}\mathbb{Q}$, so

$$\mathbb{Q}S(\cdot,\mathbb{P}_\theta) - \mathbb{Q}S(\cdot,\mathbb{Q}) = \int_\mathcal{X}\Big(\tfrac{1}{2}\big\|m^\top \nabla_x \log p_\theta\big\|_2^2 + \tfrac{1}{2}\big\|m^\top \nabla \log q\big\|^2 + \nabla \cdot \big(mm^\top \nabla \log p_\theta\big)\Big)\mathrm{d}\mathbb{Q}$$
$$= \operatorname{DSM}_m(\mathbb{Q}\|\mathbb{P}_\theta).$$

From 4.2 [15] the influence function is then $\operatorname{IF}(x,\mathbb{P}_\theta) = g_{\mathrm{DSM}}(\theta)^{-1}s(x,\theta)$, where

$$s(x,\theta) \equiv \nabla_\theta S(x,\theta) = \tfrac{1}{2}\nabla_\theta\|m^\top \nabla_x \log p_\theta\|_2^2 + \nabla_\theta \nabla_x \cdot \big(mm^\top \nabla_x \log p_\theta\big)$$
$$= \tfrac{1}{2}\nabla_\theta\|m^\top \nabla_x \log p_\theta\|_2^2 + \nabla_\theta\big(\langle\nabla_x \cdot (mm^\top), \nabla \log p_\theta\rangle + \operatorname{Tr}\big[mm^\top \nabla_x^2 \log p_\theta\big]\big)$$
$$= \nabla_x \nabla_\theta \log p_\theta mm^\top \nabla_x \log p_\theta + (\nabla_x \nabla_\theta \log p_\theta)\nabla_x \cdot (mm^\top) + \operatorname{Tr}[mm^\top \nabla_x \nabla_x]\nabla_\theta \log p_\theta$$

and where $g_{\mathrm{DSM}}(\theta) \equiv \mathbb{P}_\theta \nabla_\theta \nabla_\theta S(\cdot,\theta)$ is the information metric associated with DSM. Hence the estimator is bias-robust iff $x \mapsto s(x,\theta_*^{\mathrm{DSM}})$ is bounded.

# F   Additional Numerical Experiments

In this section, we provide further details and expand on the numerical experiments in the main paper.

## F.1   Efficiency of Minimum SD Estimators for Scale Parameters of Symmetric Bessel distributions

In this section, we extend the results from the main text and compares SM with KSD based on a Gaussian kernel and a range of lengthscale values for the scale parameter of the symmetric Bessel distribution. The results, given in Fig. 1, are also based on $n = 500$ IID realisations in $d = 1$. Similar results to those for the location parameter are obtained: KSD can deal with rougher densities, as illustrated when $s = 0.6$.

Figure 5: *Minimum SD Estimators for the Scale of a Symmetric Bessel Distribution*. We consider the case where $\theta_1^* = 0$ and $\theta_2^* = 1$ and $n = 500$ for a range of smoothness parameter values $s$ in $d = 1$.

## F.2   Bias Robustness of Minimum SD Estimators for the Symmetric Bessel and Non-standardised Student-t Distributions

In this section, we explore the robustness of minimum SD estimators for the two other examples in the main paper: the symmetric Bessel distribution ($\nu = 1000$) and the non-standardised student-t distribution. We once again select a diffusion matrix of the form $m(x) = 1/(1 + \|x\|^\alpha)$, and fix $\alpha = 1$ in both cases. This choice is refered to as "robust DKSD". On the other hand, we call "efficient DKSD" the DKSDs with choices of $m$ as highlighted in the main text (and which were chosen to improve efficiency in both cases). The results are provided in Fig. 6. In each case, we used $n = 500$ data points, 80 of which were corrupted by a Dirac at some value of given on the x-axis. Both in the student-t and symmetric Bessel distribution, we notice that the "efficient DKSD" has an $l_1$ error which grows with the value of the Dirac, whereas the "robust DKSD" is bounded as a function of this Dirac.

Figure 6: *The Robustness of Minimum SD Estimators for the Symmetric Bessel and Student-t Distributions*. *Left:* Student-t distribution. *Right:* Symmetric Bessel distribution.

## Footnotes

[1] See sec 4.4 [20] for Riemannian Newton method

[2]More generally $\nabla_y \cdot (m(y)K) = (\nabla_y \cdot m) \cdot K + \mathrm{Tr}[\nabla_y K \otimes m(y)]$ where $\mathrm{Tr}[\nabla_y K \otimes m]_r = \partial_{y^i}K_{jr}m_{ij}$ and if $K = Bk$

[3] Here we use $\nabla \cdot (mm^\top \nabla \log p) = \langle \nabla \cdot (mm^\top), \nabla \log p \rangle + \mathrm{Tr}[mm^\top \nabla^2 \log p]$