[Reviews · NeurIPS 2019]

Reviewer 1



In this paper the authors present a kernel based frame work using stein estimators to fit normalized densities. This framework is quite general and covers several existing estimators. They also prove nice theorems on the finite/asymptotic behavior of the estimator as well as how it approximate other estimators. Finally they present evidence that this framework estimates better than the standard score matching on some nastier toy datasets, heavy tailed and nonsmooth. Overall the contribution of the paper seems very much worth publishing although this topic is very much outside my area of expertise, so I cannot be sure how it fits into existing literature, or how novel it is. My main issue with the paper is that it is very technically dense, perhaps unnecessarily so. As someone outside the field I felt like the mathematical exposition could be significantly decompressed for the introduction of their framework (Section 2.0) with some additional intuition and leading provided for the reader. To compensate one could omit, shorten, make more vague, some of the content in Sections 2.1-3.2. For example Theorem 1 would definitely benefit from hiding more of the details under the hood, i.e. in the supplement. It would be better if the paper was more readable and made it clear what additional details can be found in the supplemental section for the very interested reader. If this were improved I could see giving the paper a 7 or 8.

Reviewer 2



Originality: The papers build on existing literature to provide a unified framework for stein discrepancies based on a generalized diffusion operator. Although the proposed generalization is expected, it is clearly beneficial to have such complete and rigorous treatment of stein discrepancies in the context of learning. Significance: The paper exhibits and characterizes cases when score matching would have undesirable behaviors compared to KSD. Such contribution is very valuable to the community. Also, the new generalization provides a whole new class of discrepancies that are more suited for heavy-tailed and non-smooth distributions. Quality: the proofs are sound, the experiments are simple but convincing which makes it a complete and insightful paper. Clarity: The structure is very clear and the paper is pleasant to read. I have two small questions: - In section 4.2, it seems that m was picked to match somehow the expression of the parametric model. What guides such choice? Can one think of a general guideline to pick such function to improve the convergence properties of the loss (convexify the loss)? - The theory predicts that DSM is more robust than SM, could this be illustrated in a simple example? ================================ After reading the other reviews and the authors' response, I still think this is a good submission and should be accepted. It would be great to incorporate the explanations provided in the response to the final version of paper.

Reviewer 3



Update: Thanks for your feedback and additional experimental results. I still suggest accept. ============ As far as I know, it seems original that the work focuses on leveraging a flexible metric (e.g. information geometry) into probability discrepancies. I did not go through the proof so I do not know the validity and the difficulty in adapting existing results (e.g. [21]), but the results seem reasonable. The provided results also cover a wide range of considerations on the proposed discrepancies. Issues: * On generalizing the metric in discrepancies, the authors are recommended to discuss the relation between DKSD and the Riemannian kernel Stein discrepancy proposed in Liu & Zhu (2018; arXiv:1711.11216). * In Eq. (2), what does it mean that m is a diffusion matrix? Does it have to be symmetric positive definite? Referring to the operator in Theorem 2 of [21], if m is taken as the sum of the covariance and stream coefficients, it could be any matrix. Does the considered operator have this generality? * For DKSD and DSM to be discrepancies constructed by Stein's method, the corresponding operators need to satisfy Stein's identity. I noted that the identity for DKSD is provided in Line 120, but did not find the one for DSM. * Why the matrix-valued kernel considered in DKSD has to be in the two forms? Is it possible to extend the results to a general kernel? The corresponding vector-valued RKHS theory is known, e.g., [C. Micchelli & M. Pontil, 2003, On Learning Vector-Valued Functions]. * For the experiments, is it possible to make the experiments more related to the theoretical analysis? For example, analyse the failure of SM in estimating symmetric Bessel distributions, or compare DSM with SM to demonstrate the benefit. The presentation of the paper is basically clear, and I appreciate the notation system. Here are some additional concerns. * In Line 76, "Stein's identity" is referred before its definition in Line 80. * Unexplained abbreviation "SDE" in Line 90. * The authors could use another symbol for the dimensionality of \theta instead of m. * Possible typos: Line 177: "probality" -> "probability" Line 214: no subject in the sentence Line 283: "his" -> "This"

[Author Response · NeurIPS 2019]

**Author Response for Paper #7112: Minimum Stein Discrepancy Estimators**

We would like to start by thanking the three reviewers for their careful consideration of our paper, as well as their insightful comments which will certainly help further strengthen it. Our response below first addresses comments shared across reviewers then addresses more specific questions from individual reviewers.

**Shared Comments**

1. **R2:** *"...the paper should be more focused on giving a clear idea of what is being demonstrated by expanding certain sections, and perhaps omitting or making more vague the particularly dense parts"*, **R4:** *"The structure is very clear and the paper is pleasant to read."*, **R6:** *"The presentation of the paper is basically clear"*.

   While there is some disagreement regarding clarity (with the more confident reviewers expressing satisfaction), we agree that some aspects of the paper stand to be more accessible. We propose the following changes:

   - We will make the statement of Thm 1 more concise, as requested. Furthermore, we will shorten Thm 3, and unify Thm 4 & 5 under simpler assumptions. In each case, implications of the theorems will be discussed at greater length, and the full technical details will be relegated to the supplementary information.
   - We will expand Sec. 2.1, adding a number of clarifications on DSM and DKSD which will help the reader develop intuition. This will include greater discussion of how to make choices of $m$ and $K$ to improve performance.

2. **R2:** *"More extensive experiments would have been nice"*, **R4:** *"An illustrative example of the failure of the robustness of SM and how DSM can improve it"*., **R6:** *"compare DSM with SM to demonstrate the benefit."*

   We propose to add an experiment demonstrating the robustness of DSM & DKSD for generalised gamma location models: $p(x|\theta) \propto \exp(-(x - \theta)^c)$. We set $n = 300$ and corrupt $80$ points with the value of $x = 8$; a robust estimator should obtain a good approximation of $\theta^*$ even under corruption. *Left plot:* The model is Gaussian for $c = 2$; we see that SM is not robust for this very simple model whereas DSM with $m(x) = 1/(1 + \|x\|^\alpha), \alpha = 2$ is robust. *Middle plot:* The same $m$ also leads to a robust DKSD. *Right plot:* we take $c = 5$ and see that $\alpha$ can be chosen as a function of $c$ to guarantee robustness (our theory requires $\alpha \geq c$).

   We also propose to add an experiment showing the benefits of DKSDs for product-of-expert models. This will demonstrate the advantages of our methodology for applications where SM is typically applied.

**Reviewer 4**

- *"... m was picked to match somehow the expression of the parametric model. What guides such choice? Can one think of a general guideline to pick such function to improve the convergence properties of the loss [...]?"*

  Thank you for the opportunity to elaborate on this central point. The novelty of this approach is that $m$ can be chosen so that DKSD satisfies various desirable properties: sample efficiency (e.g., for a location parameter, select $m$ so that $m(x)\nabla_x \log p_\theta(x) \approx -(x - \theta)$), robustness (ensure $\|K(x,y)m(y)\nabla_y \log p_\theta(y)\| \to 0$ as $\|y\| \to \infty$) and convexity, with different choices of $m$ in each case. We will elaborate on general guidelines in the paper.

**Reviewer 6**

- *"In Eq. (2), what does it mean that m is a diffusion matrix? Does it have to be symmetric positive definite?"*

  The terminology "diffusion matrix" comes from the fact that the Stein operator arises from the generator of a diffusion process, and is in general an arbitrary matrix function. It needs to be invertible for both DKSD and DSM to be statistical divergences (see Prop 1 and Thm 2). It therefore does not necessarily need to be positive definite.

- *"...discuss the relation between DKSD and the Riemannian kernel Stein discrepancy..."*

  Thank you for pointing this out; we will elaborate on the connection in the paper.

- *"I noted that the identity for DKSD is provided in Line 120, but did not find the one for DSM"*

  The identity holds under the assumptions on $p$ and $q$ stated in Thm 2, where DSM is defined. This can be seen from the proof of the theorem, but we will also clarify it in the main text.

- *"Why the matrix-valued kernel considered in DKSD has to be in the two forms?"*

  Outside of Thm. 1 and Prop. 2, our stated results all hold for general matrix-valued kernels. We restricted the kernel choice for Thm. 1 to make the derivation easier to follow, but this result also extends to all matrix valued kernels, and we will include the general form in the revision.

[Meta-Review · NeurIPS 2019]

The authors introduce a general framework using kernels stein discrepancy (KSD) for estimation, which includes many existing estimators as a special case, and clearly demonstrates that such generalized discrepancies based on KSD are better at estimating non-smooth distributions and heavy-tailed distributions compared to the classical score matching (SM). This improvement is valuable to the community. There is a consensus among the reviewers that the paper should be accepted. Hence, I recommend this paper for publication at NeurIPS2019. Below are some important points that would improve the paper further. I urge the authors to take them into account for the camera-ready version. - provide more extensive experiments. - one of the reviewer complains that the paper is (unnecessarily) technically dense, which I concur. Please improve this aspect of the paper. - Please also incorporate the explanations provided in the rebuttal to the camera-ready version.